# Filling gaps in PM2.5 time series: A broad evaluation from statistical to advanced neural network models

Ruslan Safarov[1,2], Zhanat Shomanova[3*], Yuriy Nossenko[3*], Eldar Kopishev[1], Zhuldyz Bexeitova[4], Ruslan Kamatov[5]

1 Department of Chemistry, Faculty of Natural Sciences, L.N. Gumilyov Eurasian National University, Astana, Kazakhstan, 2 Department of Chemistry and Chemical Technology, Kh. Dosmukhamedov Atyrau University, Atyrau, Kazakhstan, 3 Higher School of Natural Science, Margulan University, Pavlodar, Kazakhstan, 4 Association of legal entities «Petrochemical Products Producers and Consumers Association (Petrochemical Association)», Astana, Kazakhstan, 5 Department of Science, Faculty of Natural Sciences, L.N. Gumilyov Eurasian National University, Astana, Kazakhstan

* zshoman@yandex.ru (ZS); nosenko1980@yandex.ru (YN)

## Abstract

This study addressed the critical challenge of filling gaps in PM2.5 time series data from Pavlodar, Kazakhstan. We developed and evaluated a comprehensive hierarchy of 46 gap-filling methods across five representative gap lengths (5–72 hours), introducing dynamic models capable of adapting to gaps of variable duration. Tree-based models with bidirectional sequence-to-sequence architectures delivered superior performance, with XGB Seq2Seq achieving a mean absolute error of $5.231 \pm 0.292$ µg/m$^3$ for 12-hour gaps, representing a 63% improvement over basic statistical methods. The advantage of multivariate models incorporating meteorological variables increased substantially with gap length, from modest improvements of 2–3% for 5-hour gaps to significant enhancements of 16–18% for 48–72 hour gaps. Dynamic multivariate models demonstrated remarkable operational flexibility by successfully processing real-world gaps ranging from 1 to 191 hours despite being trained on maximum lengths of 72 hours. Analysis of the reconstructed complete time series revealed that 61.2% of monitored hours exceeded the WHO daily threshold of 15 µg/m$^3$, with strong seasonal patterns and pronounced diurnal cycles. This research advances environmental monitoring capabilities by providing robust methodological tools for addressing data continuity challenges that currently limit the utility of PM2.5 measurements for public health applications and scientific analysis.

**Data availability statement:** All relevant data are within the manuscript and its Supporting Information files. Database of initial measurements of PM2.5 concentrations is available from the Zenodo database (https://doi.org/10.5281/zenodo.15305392). All code underlying the findings in this study is publicly available at: https://github.com/ruslan-saf/PM25-Gap-Filling and archived at Zenodo (https://doi.org/10.5281/zenodo.15714135).

**Funding:** This research was funded by the Science Committee of the Ministry of Science and Higher Education of the Republic of Kazakhstan within the framework of the grant AP19677560 "Monitoring and mapping of the ecological state of the Pavlodar air environment using machine learning methods".

**Competing interests:** The authors have declared that no competing interests exist.

## Introduction

### Air quality challenges and PM2.5 monitoring

Fine particulate matter (PM2.5) represents one of the most significant air pollutants affecting human health and environmental quality globally. With aerodynamic diameters smaller than 2.5 μm, these particles can penetrate deep into the respiratory system, cross the blood-air barrier, and cause widespread systemic effects [1,2]. Continuous monitoring of PM2.5 concentrations has become essential for environmental management, epidemiological research, and public health protection [3]. Such monitoring provides critical data for understanding temporal patterns, identifying pollution sources, evaluating regulatory compliance, and developing effective mitigation strategies [4].

The health implications of PM2.5 exposure are substantial and well-documented. Short-term exposure is associated with increased respiratory infections, exacerbation of asthma and chronic obstructive pulmonary disease, and cardiovascular events including stroke and heart attacks [5]. Long-term exposure contributes to reduced lung function development, chronic cardiovascular diseases, increased cancer risk, and premature mortality [6]. Recent studies have also linked PM2.5 exposure to adverse pregnancy outcomes, neurodevelopmental disorders, and accelerated cognitive decline [7,8]. The World Health Organization has progressively tightened its air quality guidelines for PM2.5, most recently to a daily mean concentration of 15 μg/$m^3$ and annual mean of 5 μg/$m^3$, reflecting mounting evidence that even low levels of exposure can harm health [9].

Beyond health impacts, PM2.5 contributes significantly to environmental degradation. These particles can transport toxic compounds over long distances, deposit in sensitive ecosystems, reduce visibility, and influence regional climate patterns through their effects on radiative forcing and cloud formation [10]. In urban areas, PM2.5 concentrations are influenced by a complex interplay of emission sources, meteorological conditions, and local topography, creating substantial spatial and temporal variability that requires dense monitoring networks [11].

In Central Asia, including Kazakhstan where our study is centered, air quality challenges are compounded by factors such as rapid industrialization, coal-dependent energy systems, aging infrastructure, continental climate extremes, and transitioning regulatory frameworks [12–14]. These conditions create both unique air pollution patterns and specific challenges for maintaining continuous monitoring operations [15].

### Missing data challenges in environmental monitoring

Despite the crucial importance of continuous air quality monitoring, measurement systems frequently encounter operational challenges resulting in data gaps. These discontinuities arise from various factors including sensor malfunctions, power outages, routine maintenance, calibration issues, and extreme weather events [16,17]. Studies have reported data completeness rates ranging from 60% to 85% in typical

monitoring networks, with some stations experiencing significantly worse coverage [18,19]. Such gaps undermine data analysis efforts, potentially leading to biased estimates of pollution levels, mischaracterization of temporal trends, and reduced effectiveness of public health warning systems [20].

Data collection disruptions can be broadly categorized as technical, operational, and environmental factors [21]. Technical causes include sensor malfunctions (component failures, measurement drift, calibration errors) [22,23], while operational interruptions stem from routine maintenance procedures like filter replacements and software updates [24]. Infrastructure challenges, particularly power outages and Internet connectivity issues in remote locations or regions with unstable grids, create additional discontinuities [25,26].

Environmental conditions present further complications, especially in regions with continental climates like Central Asia, where temperature extremes (--40°C to +40°C) can exceed equipment specifications and cause systematic seasonal gaps [27–30]. Compounding these issues are anthropogenic factors such as vandalism and resource limitations that prevent prompt repairs, often extending short gaps into prolonged missing periods [31,32].

These data gaps significantly impact the reliability and utility of air quality information across multiple applications. From an analytical perspective, missing values compromise statistical analysis by reducing sample sizes, introducing potential bias, and limiting the applicability of standard time series methods that assume continuous data [33]. When gaps coincide with important pollution events, such as severe episodes or seasonal transitions, critical information may be lost, leading to underestimation of pollution severity and population exposure [25]. Temporal pattern identification becomes particularly challenging when gaps occur non-randomly, potentially masking diurnal, weekly, or seasonal variations that are essential for understanding pollution dynamics [20].

For regulatory compliance and policy development, incomplete datasets may yield unreliable annual statistics, affecting attainment status determinations and policy effectiveness assessments [34,35]. Public health applications suffer similarly, as gaps in real-time monitoring can delay or prevent timely health advisories during pollution episodes when protection is most needed [36]. Research applications face even broader impacts, with missing data limiting the development of accurate forecasting models, exposure assessments, and source apportionment studies [37].

The importance of developing effective gap-filling methodologies extends beyond simply achieving dataset completeness. The quality of imputed values significantly affects downstream analyses and decisions. Methods that merely insert statistically convenient values (such as means or medians) without accounting for temporal patterns may satisfy basic continuity requirements but can introduce artificial patterns or suppress real variability [38]. Conversely, sophisticated approaches that leverage underlying data structures, correlations with meteorological factors, and known temporal patterns have the potential to reconstruct missing segments with high fidelity to actual conditions [39].

The problem of missing data is particularly acute in developing regions and transition economies where monitoring infrastructure may be less robust, maintenance resources more limited, and operational challenges more prevalent [40]. Addressing these data gaps is essential for ensuring the continuity and reliability of air quality information that underpins environmental policy, public health interventions, and scientific research. As monitoring networks expand globally and generate increasingly high-resolution temporal data, effective methodologies for handling missing values have become a critical component of environmental data management systems [41].

As environmental monitoring networks generate increasingly high-resolution data and support more complex applications, the development of advanced gap-filling approaches has become a critical research area. Effective imputation methods must balance competing demands: preserving temporal patterns while avoiding introducing artificial structures, maintaining statistical properties of the original data, accommodating gaps of varying lengths, functioning reliably with limited computational resources, and adapting to diverse pollutants and monitoring contexts. Meeting these challenges requires innovative methodological approaches that combine statistical rigor with practical deployability in operational monitoring systems.

**Existing approaches for gap-filling in PM2.5 time series**

Researchers have employed a wide spectrum of methods to fill gaps in PM2.5 time series, from simple statistical imputations to sophisticated machine learning models.

**Traditional statistical methods** offer straightforward solutions. For example, replacing missing values with summary statistics (mean or median) or carrying the last observed value forward are common practices [42]. Simple interpolation (linear or spline) across a gap is also frequently used. These techniques are easy to implement but often inadequate for complex environmental data–they tend to smooth out or miss important variability. Notably, basic interpolation cannot recover sharp pollution peaks or low troughs, leading to biased daily averages when data are missing [25]. Such limitations motivate more advanced approaches.

**Time-series modeling and classical machine learning** provide more dynamic gap-filling strategies. Autoregressive models like ARIMA (Auto-Regressive Integrated Moving Average) leverage temporal correlations in the PM2.5 series to predict missing values. ARIMA and its seasonal extensions (e.g., SARIMA/SARIMAX) have been widely applied in air quality time series analysis and serve as strong benchmarks for gap-filling [43]. In some evaluations, ARIMA-based methods achieved comparable or better accuracy than modern nonlinear models for short gaps [44]. Their strength lies in statistical rigor and the ability to model seasonality; however, they require assumptions of stationarity and can propagate errors when used to impute long consecutive gaps (by iteratively forecasting each missing point). To incorporate additional predictors and nonlinear relationships, researchers have turned to ensemble learning methods. **Tree-based models** like Random Forests and gradient boosting (e.g. XGBoost) have been used to predict missing PM2.5 readings from correlated variables and historical data [44]. For instance, Xiao et al. [45] developed an ensemble model that combined decision-tree learners (including Random Forest and XGBoost) to reconstruct historical PM2.5 concentrations [44]. These machine learning models can capture complex, non-linear interactions and often handle multivariate inputs (such as meteorological factors or neighboring station data) seamlessly. Studies report that such models provide more accurate imputations than simple univariate interpolation, especially when pollutant levels depend on external factors (weather, traffic, etc.) [42]. A key advantage of tree ensembles is their robustness and relatively low risk of overfitting for moderate dataset sizes, but they do not inherently account for time dependencies unless temporal features (e.g., time lags or timestamps) are included. Consequently, their performance can degrade for long gaps where the model has to extrapolate far beyond the last known observation.

**Deep learning approaches** have gained traction for gap-filling in recent years, owing to their ability to learn sequential patterns. Recurrent neural networks (RNNs)–particularly Long Short-Term Memory (LSTM) and Gated Recurrent Unit (GRU) networks–are well-suited to time series imputation because they maintain an internal "memory" of past values [46]. LSTM-based models have been applied to PM2.5 datasets to predict missing intervals by learning from historical sequences [47]. Researchers have found LSTM can outperform simpler methods like mean fill or moving averages, achieving lower error when sufficient training data are available [48]. GRU networks, which are a streamlined variant of LSTMs, have also shown promise; in one study a GRU model achieved a mean absolute percentage error of ~11%, outperforming conventional LSTM and even other machine learning methods for imputation of hourly PM2.5 data [44]. Deep learning models excel at capturing complex temporal dynamics and can naturally handle multivariate inputs. More-over, advanced architectures (e.g., sequence-to-sequence models, denoising autoencoders, and attention-based net-works) have been proposed to directly reconstruct long missing segments by learning from the context before and after the gap [49]. These methods often yield improvements in accuracy, for example by better preserving diurnal patterns or seasonal cycles that simpler models might miss. However, neural networks require large datasets for training and care-ful tuning–otherwise they may underperform simpler models. The most commonly adopted neural network architectures for time series applications include Long Short-Term Memory (LSTM) networks, Gated Recurrent Unit (GRU) networks, Convolutional Neural Networks (CNN), and Echo State Networks (ESN) [50]. LSTM represents a specialized type of recurrent neural network designed to capture long-term dependencies in sequential data through gating mechanisms that

selectively retain or discard information, effectively addressing vanishing and exploding gradient problems that limit traditional RNNs [51]. GRU networks offer a streamlined alternative to LSTM with fewer parameters while maintaining similar capabilities for temporal modeling. CNN architectures apply convolution operations to identify local patterns and temporal structures, while ESN networks leverage reservoir computing principles for efficient training of recurrent architectures.

Indeed, a recent comparative study showed that a basic imputer like *k*-nearest neighbors or a well-tuned SARIMAX model can outperform vanilla deep networks for extended gaps, highlighting that deep learning is not guaranteed to excel without optimization [52]. Another drawback is the computational cost: training an LSTM/GRU model can be resource-intensive [25], which may be impractical for real-time gap-filling applications.

**Limitations of current methods.** Despite the variety of available techniques, gap-filling in PM2.5 time series remains challenging, especially for long-duration gaps. Many traditional methods perform adequately when the missing span is short (e.g., a few timestamps) and surrounded by known data, but their accuracy degrades for longer gaps [25,42]. Autoregressive models like ARIMA, when used to fill consecutive missing hours or days, suffer from cumulative forecast uncertainty–errors compound with each step, often leading to divergence from true values over long gaps. Machine learning models, while more flexible, face a data availability problem: for a prolonged gap, there may be limited recent information to inform the model. Unless supplemented by additional context (nearby station readings, meteorology, or seasonal profiles), even advanced models effectively operate semi-blind over long voids in the data. Researchers have noted that most gap-filling techniques partially fail when confronted with prominent (extended) gaps or anomalous events [53]. For instance, methods that ignore the local daily cycle can badly miss peak pollution hours when filling a multi-day gap. Moreover, many studies historically focused on short gaps, and consecutive missing periods spanning 12–24 + hours received comparatively little attention until recently [44]. As a result, existing approaches often struggle to reliably impute long stretches of missing PM2.5 data. In summary, while traditional statistical imputation, classical time-series models (ARIMA), tree-based regressors, and deep learning techniques each have demonstrated successes in gap-filling, they all exhibit important limitations. Simpler methods lack fidelity to complex pollution dynamics, and more advanced models can be data-hungry or prone to error propagation. These gaps in capability are especially pronounced for long missing intervals, underscoring the need for continued research into more robust gap-filling methodologies.

Despite significant advances in gap-filling methodologies, the practical implementation of these approaches faces substantial limitations in air quality monitoring systems. For continental climate conditions characteristic of cities like Pavlodar (Kazakhstan), the challenge of selecting an optimal model remains critical, particularly when dealing with extended gaps (≥24 hours) that constitute over 15% of our dataset. These constraints motivate the development of more adaptive and robust methodological frameworks specifically tailored to local environmental patterns and operational constraints.

**Research aim and objectives.** The primary aim of this study is to develop and rigorously evaluate a robust, length-adaptive gap-filling framework for hourly PM2.5 time series that maintains high accuracy across diverse missing data patterns while ensuring operational feasibility in real-world monitoring systems. This research specifically addresses the challenges of data continuity in continental climate cities with frequent extended outages and limited computational resources.

To achieve this aim, we pursue the following specific objectives:

1. Establish a comprehensive benchmark for PM2.5 gap-filling by developing a unified testing framework that systematically evaluates 46 methods (8 statistical, 28 univariate, 8 multivariate, 2 dynamical) across five representative gap lengths (5, 12, 24, 48, and 72 hours), identifying the strengths and limitations of each approach under standardized conditions.

2. Quantify the relative contributions of methodological factors (model architecture, directionality, forecasting strategy) and environmental variables (meteorological parameters, temporal indicators) to imputation accuracy across different gap scenarios, with particular attention to the performance-stability trade-off as gap length increases.

3. Design and implement dynamic univariate and multivariate sequence-to-sequence models capable of automatically adapting to gaps of varying lengths with a single set of model weights, optimizing the balance between prediction accuracy and computational efficiency.

4. Validate the proposed models using both synthetic test cases and real-world gaps encountered in operational monitoring, assessing their effectiveness for reconstructing complex pollution patterns during extended data outages up to 191 hours while maintaining temporal pattern fidelity.

5. Assess the current state of air quality in Pavlodar using the reconstructed complete time series, analyzing temporal patterns, evaluating compliance with international health standards, and identifying critical pollution episodes that may have been obscured by data gaps in the original measurements.

Through these objectives, the research aims to advance environmental monitoring capabilities by providing robust methodological tools for addressing data continuity challenges that currently limit the utility of PM2.5 measurements for public health applications, regulatory compliance, and scientific analysis.

**Scientific novelty of the research.** This study introduces several significant methodological advances in environmental time series gap-filling. First, we develop a comprehensive hierarchical framework systematically evaluating 46 imputation methods across varying gap lengths. Unlike previous studies focusing on limited approaches or specific gap durations, our structured comparison reveals performance differences and critical trade-offs between accuracy, computational efficiency, and model stability. This establishes standardized evaluation protocols for objective method selection based on monitoring requirements.

Second, we introduce novel dynamic modeling architectures addressing fundamental limitations of conventional fixed-context approaches. While previous gap-filling models require predetermined context windows and separate implementations for different gap lengths, our dynamic models automatically adjust context processing based on gap characteristics. This innovation allows a single model to effectively handle gaps ranging from one hour to over one week without reconfiguration or retraining, representing significant advancement over previous work where performance deteriorated substantially when applied to different gap durations.

Third, we provide rigorous empirical comparison between univariate and multivariate modeling approaches across diverse gap scenarios. Our research systematically quantifies the added value of incorporating meteorological variables for different gap lengths and model architectures. This reveals that multivariate modeling benefits increase substantially with gap length, from minimal improvements for short gaps (2–3%) to significant enhancements for extended intervals (16–18%). These findings provide practical guidance for selecting appropriate model complexity based on gap characteristics and available contextual data.

Our contribution emphasizes operational adaptability for real-world monitoring systems rather than algorithmic novelty, addressing the practical challenge of maintaining a single model capable of handling diverse gap scenarios encountered in environmental monitoring networks.

These innovations extend current knowledge regarding environmental time series reconstruction and provide practical solutions for challenging operational conditions in air quality monitoring networks.

## Methods

### Data description and preprocessing

**Data sources and study area.** Measurement was performed automatically using a stationary monitoring unit equipped with an AQS008A air quality sensor (Sichuan Weinasa Technology Co., Ltd., The International Creative Federation Cross-Border E-Commerce Industry Park, Mianyang, China) [54]. The AQS008A sensor uses laser-based detection principles for PM2.5 concentration measurement. Data were collected from 23 May 2024 to 19 January 2025 at

a 1-minute sampling interval. Measured data on PM2.5 concentration are presented in SQLITE database (freely available at https://doi.org/10.5281/zenodo.15305392).

The station operates within a research project framework and is installed on a building belonging to Margulan University in Pavlodar, Kazakhstan (coordinates: 52°17'56.6"N, 76°57'18.3"E). This location is partially shielded from direct traffic exposure by the surrounding academic buildings; however, several potential emission sources are present within a short distance: railway tracks are located approximately 200 m to the north, warehouses and garages about 100 m to the east, and one of the city's major roads approximately 200 m to the south.

**Meteorological data.** In addition to PM2.5, air temperature, and relative humidity readings collected by the AQS008A sensor, a complementary set of meteorological variables (T, P0, P, U, DD, Ff, VV) was downloaded from the RP5 weather archive for Pavlodar (airport, METAR) [55]. These variables represent, respectively, temperature (T, °C), sea-level pressure (P0, mmHg), station-level pressure (P, mmHg), relative humidity (U, %), wind direction (DD, compass points), wind speed (Ff, m/s), and horizontal visibility range (VV, km). Timestamps from the downloaded dataset were synchronized with PM2.5 sensor data to create a unified time series. Meteorological data are available in S1 File.

**Initial data filtering and quality checks.** *Outlier Detection* To address anomalously high or low PM2.5 measurements, two complementary checks are performed. First, suspiciously large spikes are flagged if they exceed a threshold (for instance, 200 µg/m³) and are more than triple the previous reading. Second, the interquartile range (IQR) method locates points lying outside $[Q1 - 1.5 \times IQR, Q3 + 1.5 \times IQR]$ (Fig 1). Rather than discarding these outlier points, the script converts them into missing values (NaN), thus preserving the possibility of gap-filling them in subsequent steps.

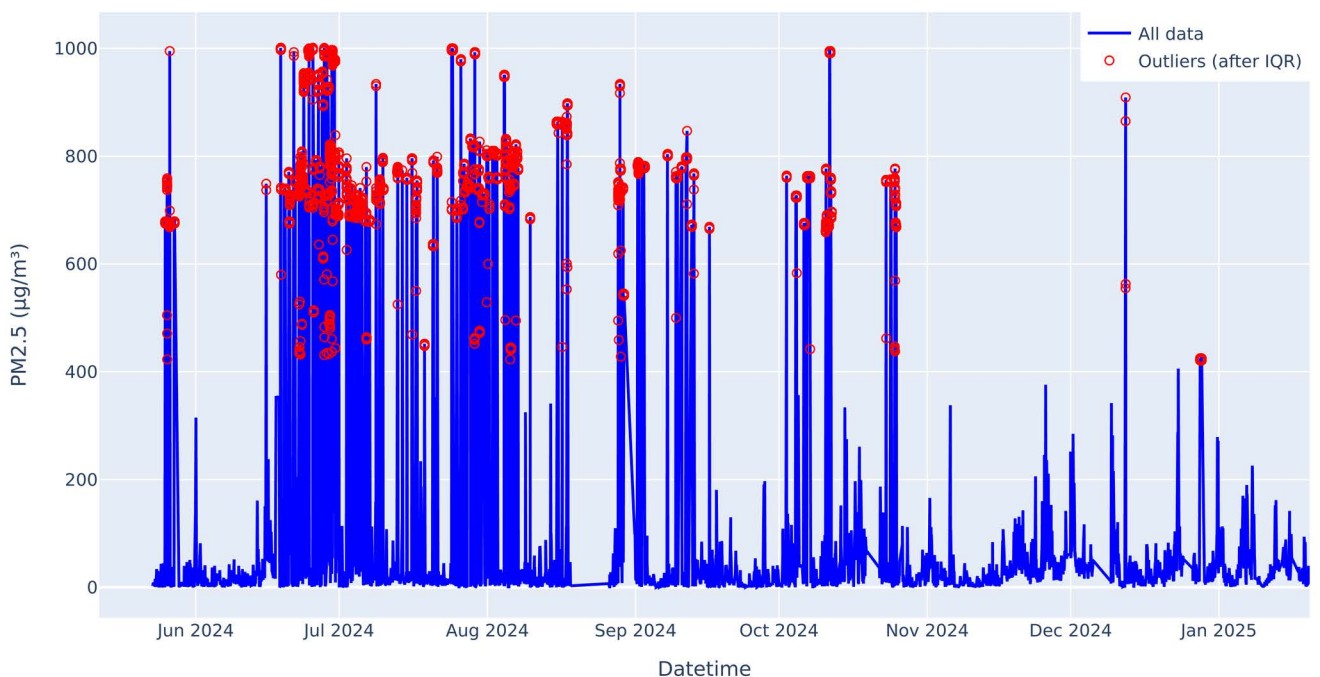

**Fig 1. PM2.5 measurements showing detected outliers (red dots) based on threshold and IQR methods.**

*Time Range Formation:* Once outliers are marked, the dataset is aligned to a uniform minute-by-minute timeline. Any timestamps absent in the original data become rows filled with NaN, ensuring that the entire period of interest is fully represented and simplifying future interpolation or time-based analyses.

*Delayed Measurement Correction:* Some readings arrive slightly late and might cause spurious gaps of just a few seconds. If the timestamp delay is under a predefined threshold (for instance, 15 s), the reading is shifted back to the intended minute. This prevents artificially introduced breaks, yielding a cleaner series with fewer false gaps.

*Short Gap Interpolation:* After outlier marking and time alignment, consecutive NaN segments are checked. If a gap is shorter than a chosen duration (e.g., five consecutive minutes), it is filled by linear interpolation. This approach retains most of the data continuity while ensuring that only genuinely larger gaps remain unfilled for further advanced methods.

*Hourly Aggregation and Secondary Outlier Filtering:* Because many analyses use hourly data, minute-level observations are averaged to hourly blocks, provided each hour has enough valid measurements (for instance, at least 40 non-NaN minutes). Any hour lacking sufficient data remains NaN, indicating insufficient coverage for a reliable hourly mean. After aggregation, an additional outlier detection is applied on the hourly data. This secondary filtering flags hours with values exceeding 270 μg/m³ or values over 200 μg/m³ that are more than three times higher than the previous hourly reading. The threshold value of 270 μg/m³ was established based on statistical analysis of the value distribution and corresponds to the 99.61st percentile in the dataset, meaning that only 0.39% of all observations exceed this threshold (Fig 2). This approach effectively identifies abnormally high values while preserving the natural variability of the data. The additional criterion for sharp spikes helps detect short-term anomalies that may be caused by measurement interferences or brief episodes of severe pollution. Outliers detected at this aggregated level are similarly converted to NaN to preserve the potential for gap-filling, ensuring that the hourly time series used for subsequent analyses is both complete and robust.

*Final Checks* As a last step, graphical overviews (heatmaps and missing-data plots) verify that outliers were properly flagged, time gaps handled consistently, and minute-to-hour resampling performed accurately. Fig 3 presents a comprehensive visualization of data completeness across the entire study period, revealing significant variability in data

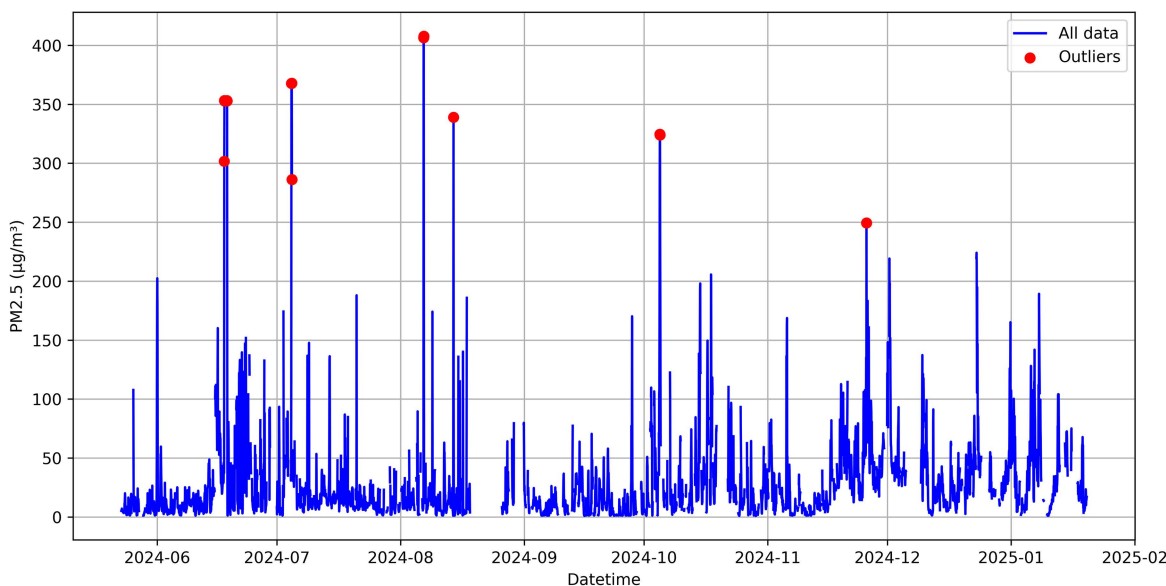

**Fig 2. Secondary outlier detection in hourly aggregated PM2.5 measurements, highlighting abnormal spikes (red points) identified by exceeding threshold criteria (>270 μg/m³) or showing disproportionate increases (>200 μg/m³ and >3 × previous value).**

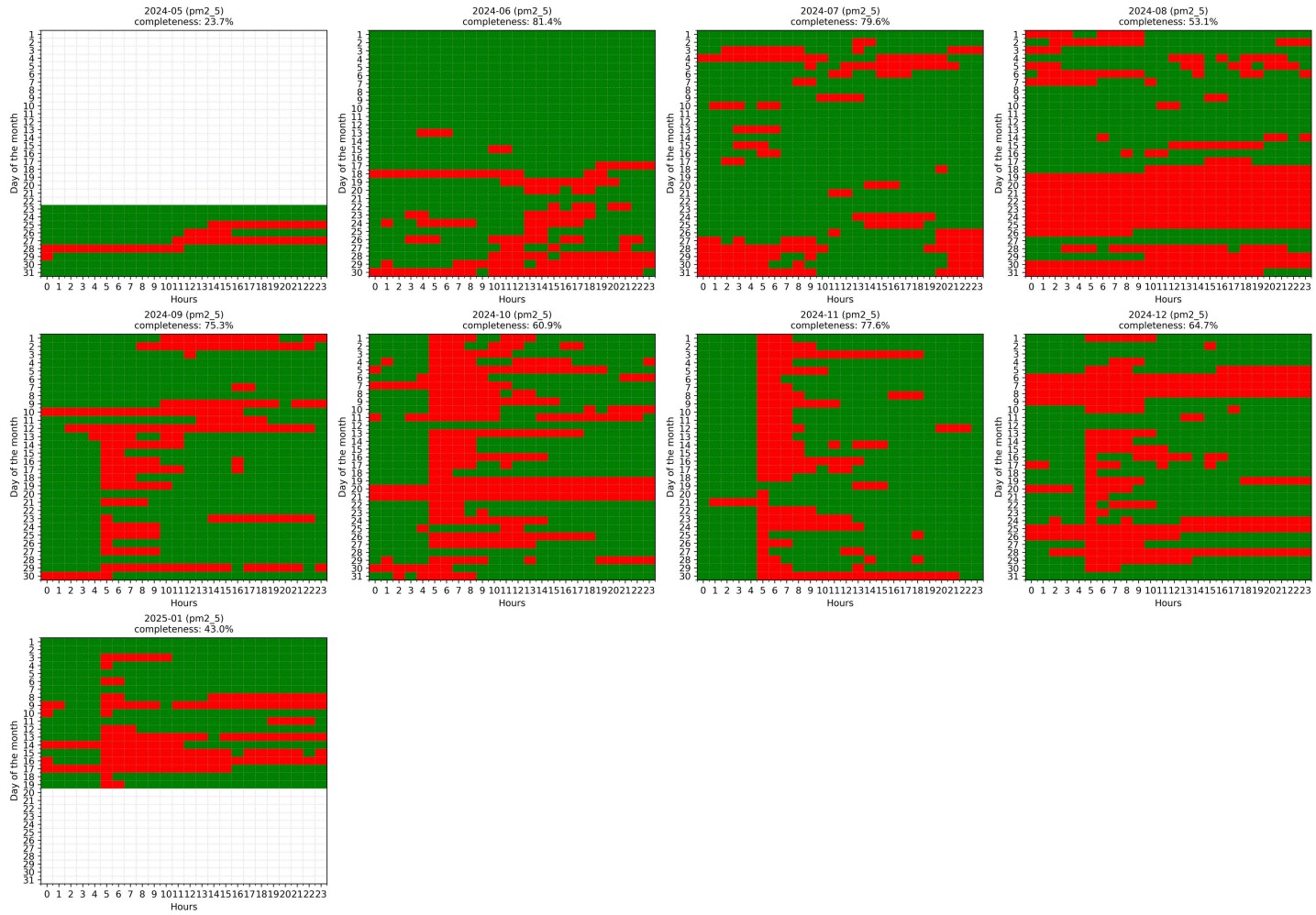

**Fig 3. Hourly PM2.5 data completeness visualization after quality checks and outlier removal.** The heatmaps display data availability by month from May 2024 to January 2025, with each cell representing an hour of the day (x-axis) for each day of the month (y-axis). Legend: Green = Data Available, Red = Missing Values, White = Outside Measurement Period. Monthly completeness percentages are shown in the top-right corner of each panel.

availability between months (ranging from 23.7% to 81.4% completeness) and highlighting the spatial-temporal patterns of gaps that will require treatment. These visual checks confirm the dataset is now ready for advanced gap-filling or modeling.

**Final dataset specification.** After preprocessing, the final hourly dataset spans from May 23, 2024 to January 19, 2025, comprising 5,791 hourly timestamps (242 days). Overall data completeness for PM2.5 measurements is 73.3%, with 1,546 missing values representing 26.7% of the total time series. Analysis of gap distribution reveals that most missing data occurs in relatively short segments: 84.9% of gaps are short (≤12 hours), 12.6% are medium-length (13–48 hours), and only 2.5% are extended periods (>48 hours), with the longest continuous gap spanning 191 hours. In total, 159 distinct gaps were identified across 25 different gap lengths, with the most frequent being 3-hour gaps (35 occurrences) (Fig 4). These gaps are not uniformly distributed throughout the monitoring period, as demonstrated in

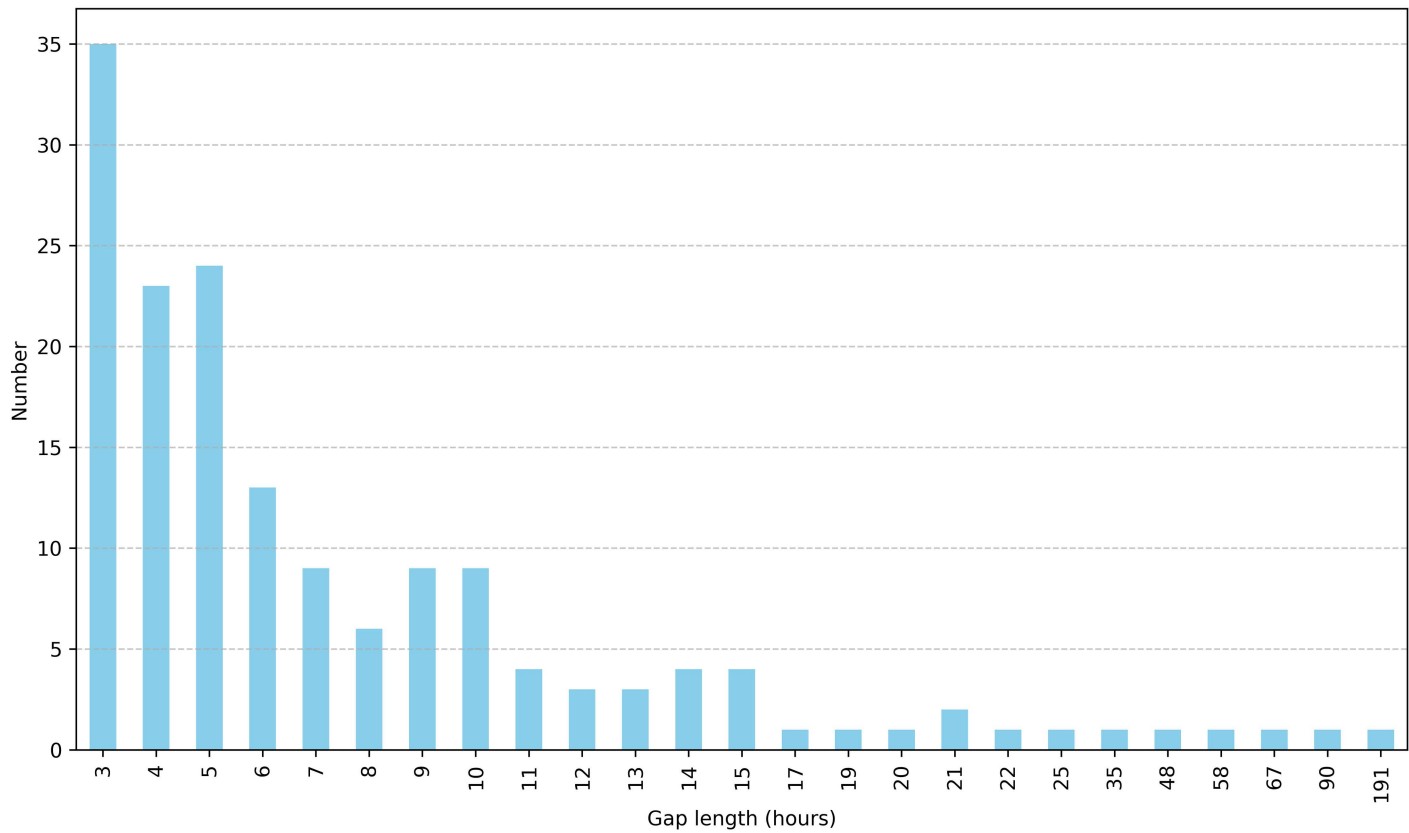

**Fig 4. Distribution of PM2.5 data gaps by length (in hours).**

Fig 3, with August exhibiting the lowest completeness (55.6%) and June showing the highest (85.7%). This pattern of missing data presents a significant challenge for environmental analysis and necessitates advanced imputation techniques to produce a continuous time series suitable for pollution trend analysis and forecasting applications.

## Experimental framework for gap-filling evaluation

**Synthetic gap creation and testing infrastructure.** To effectively evaluate and compare various gap-filling methodologies for PM2.5 data, we developed a comprehensive synthetic gap testing framework. This approach provides several key advantages over relying solely on naturally occurring data gaps.

Real-world environmental monitoring data often contains gaps with irregular distribution, varying lengths, and unknown ground truth values. These characteristics make it difficult to reliably evaluate the performance of different imputation methods. By introducing artificial gaps into complete data segments, we create a controlled testing environment with the following benefits:

1. By artificially removing values that actually exist in the dataset, we retain access to the true values for accurate performance assessment.

2. We can systematically vary gap lengths and patterns to evaluate model performance under different scenarios.

3. Multiple randomized runs with different gap placements allow for more reliable performance metrics and confidence intervals.

4. All methods are evaluated on identical gap patterns, ensuring comparative results reflect actual method differences rather than data peculiarities.

The synthetic gap generation process implemented in our framework follows a systematic approach. For each experiment, we:

1. Create a copy of the original dataset to preserve the source data.

2. Calculate the number of gaps to introduce based on a specified missing fraction (typically 5% of the total dataset).

3. Randomly select starting indices for gaps, ensuring they are sufficiently distant from dataset boundaries.

4. Replace original values with NaN for each designated gap.

5. Return the modified dataset with artificial gaps and the indices of these gaps for later evaluation.

To comprehensively evaluate method performance across different missing data scenarios, we tested multiple gap lengths:

− Short gaps: 5, 12 hours (representing brief sensor malfunctions or maintenance)

− Medium gaps: 24 hours (representing daily outages)

− Long gaps: 48, 72 hours (representing extended equipment failures)

Each model's performance was evaluated across all gap lengths and runs, resulting in a comprehensive assessment of its capabilities under various missing data scenarios.

**Data preparation for model training and testing.** Proper data preparation is critical for developing robust gap-filling models. Our methodology incorporates several key steps to ensure appropriate scaling, prevent data leakage, and transform time series data into formats suitable for different modeling approaches.

*Data Scaling and Normalization.* All numerical features undergo standardization using the StandardScaler method from scikit-learn, which transforms features to have zero mean and unit variance. This preprocessing step is essential for time series modeling as it brings all features to a comparable scale, which is particularly important for multivariate models incorporating diverse environmental parameters; accelerates model convergence, especially for neural network architectures; improves model stability by preventing larger-magnitude features from dominating the learning process.

For univariate models, only the target variable (PM2.5) was scaled. For multivariate approaches, all included features (PM2.5, air temperature, humidity, and derived temporal features) underwent normalization. Critically, to prevent data leakage, the scaler is fitted exclusively on the training portion of the data. We then apply the same transformation parameters to both validation and test sets. This ensures that no information from the test set influences the scaling process, maintaining the integrity of model evaluation.

*Time-Based Data Splitting.* To maintain the temporal integrity of the data and prevent future information from influencing past predictions, we implemented a strict time-based splitting approach:

1. The dataset was chronologically ordered by timestamp

2. The first 80% of the data points were allocated to the training set

3. The remaining 20% were reserved for testing

This approach differs from traditional random splitting in machine learning, as it respects the temporal structure of the data. It also simulates real-world conditions where models must predict future or missing values based solely on past observations. Importantly, the test set represents a future time period that the model has not seen during training, creating a realistic evaluation scenario.

*Data Transformation for Different Model Types.* The preparation of training and testing data varied according to the imputation approach:

1) Simple and Window-Based Methods. For baseline methods (mean, median) and window-based approaches (linear, polynomial interpolation), the data required minimal preprocessing beyond scaling. These methods utilize local information from specified time windows surrounding gaps.

2) Sequence-to-Sequence Models. For models designed to predict entire gaps at once, we extracted fixed-length contexts before and after each gap. Both univariate and multivariate inputs were structured as time windows (typically 32 time steps) containing either PM2.5 values alone or multiple environmental parameters.

3) Autoregressive Models. For models predicting one step at a time, we created sliding window inputs where each window predicts the subsequent value. During inference, these models recursively incorporate newly predicted values into the input window for subsequent predictions.

4) Unidirectional vs. Bidirectional Models. Our framework implements two distinct prediction strategies:

−  Unidirectional approaches utilize a single model that processes data in one direction. While these models can incorporate both past and future data as context, they produce a single prediction through one forward pass. This approach is computationally efficient but may not fully leverage temporal patterns from both directions.

−  Bidirectional methods employ two separate models working in opposite directions. The forward model predicts future values based on past observations, while the backward model predicts in reverse–forecasting past values based on future data (using reversed time series). This produces two independent predictions, which are then combined using weighted linear interpolation, giving more weight to predictions closer to their respective input contexts.

For all model types, we maintained a strict separation between training and testing data, ensuring that validation metrics genuinely reflect model performance on unseen data patterns. This methodology provides a realistic assessment framework for comparing various imputation techniques across different gap scenarios.

**Evaluation methodology.**  To systematically assess the performance of gap-filling models, we established a comprehensive evaluation framework combining quantitative metrics and statistical analysis across multiple experimental runs.

The evaluation process follows a consistent pattern: after training, each model is applied to synthetic gaps created in the testing dataset. The predicted values are then compared with the original values (temporarily removed during the gap creation process). This provides a direct measure of how accurately each model can reconstruct the missing data.

We employed four complementary metrics to evaluate different aspects of prediction accuracy:

1. Mean Absolute Error (MAE): measures the average magnitude of errors without considering their direction [56]:

$$MAE \;=\; \frac{1}{N} \sum |y_i - \hat{y}_i|$$

(1)

where $y_i$ and $\hat{y}_i$ are the i-th observed and predicted values, respectively.

2. Root Mean Square Error (RMSE): emphasizes larger errors by squaring them before averaging [56]:

$$RMSE \;=\; \sqrt{\frac{1}{N} \Sigma (y_i - \hat{y}_i)^2}$$

(2)

 

3. Mean Absolute Percentage Error (MAPE): expresses errors as percentage of true values [57]:

$$MAPE = \frac{1}{N} \sum \left| \frac{y_i - \hat{y}_i}{y_i + \varepsilon} \right| \cdot 100$$

(3)

where $\varepsilon$ is a small constant ($10^{-8}$) to prevent division by zero.

4. Coefficient of Determination ($R^2$): measures the proportion of variance explained by the model [58]:

$$R^2 = 1 - \frac{\sum (y_i - \hat{y}_i)^2}{\sum (y_i - \bar{y})^2}$$

(4)

where $\bar{y}$ is the mean of the observed values.

These metrics provide complementary insights: MAE offers an intuitive measure of prediction error, RMSE penalizes large individual errors, MAPE provides a percentage-based metric for interpretability, and $R^2$ evaluates how well the model captures data patterns.

To ensure statistical robustness, each experiment was repeated five times (n_runs = 5) with different random seeds controlling gap placement. For each metric and model configuration, we calculated mean value across all runs, and standard deviation to quantify variability.

This approach allows us to report not just point estimates of performance but also the consistency and reliability of each method. When reporting results, we present metrics in the format "mean±standard deviation" to properly characterize each model's performance distribution.

For systematic model comparison, we organized results by gap length (5, 12, 24, 48, and 72 hours) and model category. This structure enables identification of:

1. Which models perform best for specific gap lengths

2. How performance degrades as gap length increases

3. The relative benefit of model complexity for different scenarios

4. The added value of multivariate versus univariate approaches

The evaluation methodology was implemented within a unified testing framework that ensures identical conditions across all models, enabling fair and objective comparison of diverse gap-filling approaches.

**Model selection criteria.** Final model selection was based on multiple performance criteria evaluated through systematic comparison across all gap lengths. Primary selection criterion was Mean Absolute Error (MAE) due to its direct interpretability in physical units ($\mu g/m^3$) and robustness to outliers. Secondary criteria included Root Mean Square Error (RMSE) for assessing prediction variance, coefficient of determination ($R^2$) for explanatory power evaluation, and computational efficiency measured by average runtime. Models were ranked within each category (statistical, univariate machine learning, multivariate, dynamic) based on consistent performance across all tested gap lengths (5–72 hours). The final recommended models (XGB Seq2Seq for accuracy-critical applications and Dynamic Multivariate XGB for operational deployment) were selected based on their superior and stable performance across diverse gap scenarios, combined with acceptable computational requirements for practical implementation.

**Model combination framework.** To ensure fair and systematic comparison across different gap-filling approaches, we developed a unified testing framework called ImputationCombiner. This framework streamlines the evaluation process by standardizing data preparation, model training, gap creation, performance evaluation, and results analysis.

The ImputationCombiner provides a consolidated environment where various imputation methods—from simple statistical approaches to complex deep learning models—can be evaluated under identical conditions. This infrastructure eliminates methodological inconsistencies that could bias comparisons and ensures that performance differences reflect genuine algorithmic capabilities rather than implementation variations.

The core components of the framework include:

1. A centralized data processing pipeline that handles scaling, splitting, and formatting

2. Standardized synthetic gap generation with controlled parameters

3. Uniform evaluation protocols applying consistent metrics

4. Automated result aggregation and statistical analysis

The framework employs a modular design that facilitates seamless integration of diverse imputation methods. Each model is registered with the combiner through a standardized interface that specifies:

- The model's name and category

- Training and forecasting functions

- Forecast type (e.g., "seq2seq", "autoreg", "uniseq2seq")

- Additional parameters such as window size or feature columns

This modular structure allows new methods to be incorporated with minimal code changes, enabling rapid prototyping and evaluation of novel approaches. The system also accommodates both univariate and multivariate methods through a flexible data preparation pipeline that adapts to each model's requirements.

The ImputationCombiner executes a comprehensive testing workflow that:

1. Processes each registered model across multiple gap lengths

2. Conducts several experimental runs with different random seeds

3. Computes performance metrics and their statistical distributions

4. Records execution times for performance benchmarking

Results are systematically organized and stored in structured formats for subsequent analysis. The framework generates detailed performance tables showing metrics for each model, gap length, and experimental run. These results include not only average performance metrics but also standard deviations, allowing for statistical significance analysis.

Additionally, the system offers visualization capabilities to generate comparative charts, scatter plots of predicted versus actual values, and time series visualizations of filled gaps. These visual tools complement numerical metrics by providing qualitative insights into each method's behavior.

This unified framework played a crucial role in our study by ensuring methodological consistency across all experiments, enabling objective identification of the most effective approaches for different gap-filling scenarios.

**Hierarchy of gap-filling methods**

We developed and evaluated a comprehensive taxonomy of gap-filling approaches, classified by their directionality, variable usage, and forecasting methodology. Table 1 summarizes the key characteristics and hyperparameters of all implemented methods.

**Table 1. Gap-filling methods hierarchy and parameters.**

| # | Category | Direction[1] | Variables[2] | Method | Architecture[4] | Key Parameters | Context Length[3] |
|---|----------|-----------|------------|--------|--------------|----------------|-----------------|
| 1 | Simple | – | Uni | Mean Imputation | Statistical | – | – |
| 2 | Simple | – | Uni | Median Imputation | Statistical | – | – |
| 3 | Local | – | Uni | Local Mean | Statistical | Window size = Variable | 15 + gap + 15 |
| 4 | Local | – | Uni | Local Median | Statistical | Window size = Variable | 15 + gap + 15 |
| 5 | Window | – | Uni | Linear Interpolation | Mathematical | – | 10 |
| 6 | Window | – | Uni | Polynomial Interpolation | Mathematical | Degree = 3 | Variable |
| 7 | Window | – | Uni | B-spline Interpolation | Mathematical | – | Variable |
| 8 | Window | – | Uni | ARIMA Imputation | Statistical | Order=(1,0,0) | 100 |
| 9 | Autoreg | Uni | Uni | UniAR LSTM | Neural Network | Units = 64, Activation = tanh | 32 |
| 10 | Autoreg | Uni | Uni | UniAR GRU | Neural Network | Units = 64, Activation = tanh | 32 |
| 11 | Autoreg | Uni | Uni | UniAR RNN | Neural Network | Units = 64, Activation = tanh | 32 |
| 12 | Autoreg | Uni | Uni | UniAR CNN | Neural Network | Filters = 32, Kernel = 3 | 32 |
| 13 | Autoreg | Uni | Uni | UniAR TCN | Neural Network | Filters = 32, Dilations=[1,2,4,8] | 32 |
| 14 | Autoreg | Uni | Uni | UniAR RF | Tree-based | Estimators = 50 | 32 |
| 15 | Autoreg | Uni | Uni | UniAR XGB | Tree-based | Estimators = 50 | 32 |
| 16 | Autoreg | Uni | Multi | UniAR XGB Multi | Tree-based | Estimators = 50, Features = 5 | 32 |
| 17 | Seq2Seq | Uni | Uni | UniSeq2Seq LSTM | Neural Network | Units = 64, Bidirectional | 32/32 |
| 18 | Seq2Seq | Uni | Uni | UniSeq2Seq GRU | Neural Network | Units = 64, Bidirectional | 32/32 |
| 19 | Seq2Seq | Uni | Uni | UniSeq2Seq RNN | Neural Network | Units = 64, Bidirectional | 32/32 |
| 20 | Seq2Seq | Uni | Uni | UniSeq2Seq CNN | Neural Network | Filters = 32/64, Kernel = 3 | 32/32 |
| 21 | Seq2Seq | Uni | Uni | UniSeq2Seq TCN | Neural Network | Filters = 64, Dilations=[1,2,4,8] | 32/32 |
| 22 | Seq2Seq | Uni | Uni | UniSeq2Seq RF | Tree-based | Estimators = 50 | 32/32 |
| 23 | Seq2Seq | Uni | Uni | UniSeq2Seq XGB | Tree-based | Estimators = 50 | 32/32 |
| 24 | Seq2Seq | Uni | Multi | UniSeq2Seq LSTM Multi | Neural Network | Units = 64, Bidirectional, Features = 5 | 32/32 |
| 25 | Seq2Seq | Uni | Multi | UniSeq2Seq CNN Multi | Neural Network | Filters = 32, Kernel = 3, Features = 5 | 32/32 |
| 26 | Seq2Seq | Uni | Multi | UniSeq2Seq RF Multi | Tree-based | Estimators = 50, Features = 5 | 32/32 |
| 27 | Seq2Seq | Uni | Multi | UniSeq2Seq XGB Multi | Tree-based | Estimators = 50, Features = 5 | 32/32 |
| 28 | Autoreg | Bi | Uni | LSTM Autoreg | Neural Network | Units = 64, Activation = tanh | 32 |
| 29 | Autoreg | Bi | Uni | GRU Autoreg | Neural Network | Units = 64, Activation = tanh | 32 |
| 30 | Autoreg | Bi | Uni | RNN Autoreg | Neural Network | Units = 64, Activation = tanh | 32 |
| 31 | Autoreg | Bi | Uni | CNN Autoreg | Neural Network | Filters = 32, Kernel = 3 | 32 |
| 32 | Autoreg | Bi | Uni | TCN Autoreg | Neural Network | Filters = 32, Dilations=[1,2,4,8] | 32 |
| 33 | Autoreg | Bi | Uni | RF Autoreg | Tree-based | Estimators = 50 | 32 |
| 34 | Autoreg | Bi | Uni | XGB Autoreg | Tree-based | Estimators = 50 | 32 |
| 35 | Autoreg | Bi | Multi | XGB Autoreg Multi | Tree-based | Estimators = 50, Features = 5 | 32 |
| 36 | Seq2Seq | Bi | Uni | LSTM Seq2Seq | Neural Network | Units = 64 | 32 |
| 37 | Seq2Seq | Bi | Uni | GRU Seq2Seq | Neural Network | Units = 64 | 32 |
| 38 | Seq2Seq | Bi | Uni | RNN Seq2Seq | Neural Network | Units = 64 | 32 |
| 39 | Seq2Seq | Bi | Uni | CNN Seq2Seq | Neural Network | Filters = 32, Kernel = 3 | 32 |
| 40 | Seq2Seq | Bi | Uni | TCN Seq2Seq | Neural Network | Filters = 32, Dilations=[1,2,4] | 32 |
| 41 | Seq2Seq | Bi | Uni | RF Seq2Seq | Tree-based | Estimators = 50 | 32 |
| 42 | Seq2Seq | Bi | Uni | XGB Seq2Seq | Tree-based | Estimators = 50 | 32 |
| 43 | Seq2Seq | Bi | Multi | XGB Seq2Seq Multi | Tree-based | Estimators = 50, Features = 5 | 32 |
| 44 | Seq2Seq | Bi | Multi | RF Seq2Seq Multi | Tree-based | Estimators = 50, Features = 5 | 32 |
| 45 | Dynamic | Uni | Uni | Dynamic Uni Seq2Seq XGB | Tree-based | Estimators = 50, Variable Context | Variable |
| 46 | Dynamic | Uni | Multi | Dynamic Multi Seq2Seq XGB | Tree-based | Estimators = 50, Features = 5, Variable Context | Variable |

[1]Direction: Uni = Unidirectional, Bi = Bidirectional

[2]Variables: Uni = Univariate (PM2.5 only), Multi = Multivariate (PM2.5 + meteorological and temporal features)

[3]Context Length: For bidirectional models, values represent pre/post gap context lengths

[4]Neural network models used Adam optimizer with learning rate = 0.001, trained for up to 30 epochs with early stopping (patience = 15). For all models, batch size = 32 was used during training

**Baseline statistical methods.** The simplest approaches to gap-filling rely on basic statistical operations without requiring model training. These methods serve as foundational benchmarks for more complex techniques.

*Simple Imputers* replace missing values with a single statistic calculated from the available data. We implemented Mean and Median imputation, which substitute gaps with the average or median value of the entire time series, respectively.

*Local Statistical Methods* operate within a defined window surrounding each gap. Local Mean and Local Median calculate the respective statistic using only values within this window, providing more contextually relevant replacements than global statistics.

*Window-Based Interpolation* methods construct mathematical functions that pass through known points surrounding gaps. Linear interpolation fits straight lines between adjacent points, while Polynomial interpolation (degree = 3) and B-spline interpolation fit more complex curves that can capture non-linear patterns. These methods excel at preserving local trends but may struggle with longer gaps.

*ARIMA Imputation* applies time series modeling principles to extrapolate missing values using an ARIMA(1,0,0) model trained on values preceding each gap. This method can capture temporal dependencies but is computationally intensive compared to simpler approaches.

**Machine learning approaches.** Beyond statistical methods, we implemented machine learning models with varying architectures, directionality, and forecasting strategies.

## Forecasting strategies

*Autoregressive Models* predict one step at a time, using each new prediction as input for subsequent forecasts. While this approach can propagate errors, it adapts well to changing patterns within gaps.

*Sequence-to-Sequence (Seq2Seq) Models* directly predict the entire gap as a single output. This approach avoids error accumulation but requires the model to capture longer-term dependencies.

## Directionality approaches

*Unidirectional Models* use context from only one temporal direction, typically forecasting forward using past values. These models include UniAR (Unidirectional Autoregressive) and UniSeq2Seq (Unidirectional Sequence-to-Sequence) variants.

*Bidirectional Models* utilize context from both past and future observations. These methods train two separate models: a forward model predicting from past to future, and a backward model predicting from future to past (using reversed time series). The final imputation combines these predictions with distance-weighted averaging, giving more weight to predictions closer to their input context.

## Model architectures

Our framework incorporates diverse predictive models, each with unique strengths:

*Tree-Based Models* (Random Forest, XGBoost) effectively capture non-linear patterns without requiring extensive hyperparameter tuning. These models process flattened input windows rather than sequential data structures.

*Recurrent Neural Networks* (SimpleRNN, LSTM, GRU) specialize in sequential data, with LSTM and GRU offering sophisticated mechanisms to maintain long-term dependencies while addressing vanishing gradient problems.

*Convolutional Models* (CNN, TCN) apply convolution operations to capture local patterns and dependencies. TCN (Temporal Convolutional Network) specifically employs dilated convolutions to expand receptive fields for capturing longer-range dependencies.

**Multivariate extensions.** To enhance predictive performance, we extended key models to incorporate additional contextual features beyond PM2.5 values:

*Feature Selection* included air temperature, air humidity, hour of day, and season indicators. These features were selected based on their known correlations with air quality patterns and their availability in practical deployment settings.

*Input Representation* for multivariate models included multiple channels for each time step. For example, instead of a single PM2.5 value per time step, multivariate models received vectors containing PM2.5, temperature, humidity, and temporal features.

*Model Adaptations* required architecture modifications to handle multi-dimensional inputs. For neural networks, this involved increasing input layer dimensions; for tree-based models, feature vectors were flattened while preserving their relationships.

Table 1 summarizes the key characteristics and hyperparameters of all implemented methods, organized by their classification within our hierarchy.

This comprehensive framework allowed us to systematically evaluate the impact of model architecture, directionality, variable usage, and forecasting strategy on gap-filling performance across different gap lengths.

**Dynamic models development.** Moreover, we developed dynamic models capable of adapting their context processing based on gap characteristics. Our approach centers on three key innovations:

First, we implemented dynamic context sizing, where the effective context window adjusts proportionally to gap length. For gaps ≤10 hours, we used a factor of 3×the gap length (e.g., 15-hour context for a 5-hour gap). For longer gaps, we capped context at 32 time steps to maintain computational efficiency while preserving sufficient information. This allows the model to function effectively even when full context is unavailable, such as near the edges of the time series or between closely spaced gaps.

Second, we developed a unified training methodology that exposes the model to gaps of various lengths during a single training process. This creates a generalized model capable of handling diverse gap scenarios without requiring separate specialized models. Instead of maintaining 25 different models for each observed gap length, a single dynamic model can handle any gap length from 1 to 191 hours, dramatically simplifying operational deployment.

Third, we incorporated explicit metadata into the input features, including gap length being predicted, dynamic context size being used, and position within the gap being filled. This metadata provides the model with explicit information about the prediction task's characteristics, allowing it to adjust its internal processing accordingly.

*Technical Implementation Details* Our primary dynamic architecture employs XGBoost regression with several adaptations for time series gap filling. The model processes:

1. Left context (pre-gap): Variable-length observations before the gap, padded to a maximum size ($C\_max = 32$);

2. Right context (post-gap): Variable-length observations after the gap, similarly padded;

3. Metadata features: Gap length, context size, and position indicators.

For gaps of varying lengths, the model outputs predictions for the maximum supported gap length, with only the relevant portion used for shorter gaps. This approach creates a single model capable of handling diverse gap scenarios.

The preprocessing pipeline dynamically selects appropriate context windows based on the gap. During training, we create a comprehensive dataset that includes examples of all target gap lengths (5, 12, 24, 48, and 72 hours), with appropriately sized context windows for each. This exposes the model to diverse gap patterns and their optimal context requirements simultaneously.

A critical component of our implementation is the position-aware padding procedure. Since the model requires consistent input dimensions, we standardize all inputs to a fixed maximum size through padding:

− For the left context, we pad with zeros on the left side, preserving the most recent values closest to the gap

− For the right context, we pad with zeros on the right side, preserving the earliest values after the gap

 

- This creates input vectors where actual data points are positioned closest to the gap, with zero padding in the more distant positions

This padding approach ensures that the most relevant temporal information (closest to the gap boundaries) is preserved in a consistent position within the input vector, allowing the model to learn more effectively from the available context.

*Gap-Filling Algorithm for Real-World Applications.* For operational deployment, we implemented a comprehensive gap-filling algorithm capable of addressing the challenges presented by real-world missing data patterns:

1. The algorithm first identifies all missing data segments in the time series, determining the start index and length of each gap.

2. It then assesses the available context around each gap, adapting to whatever context is available–even if significantly shorter than the ideal 32 time steps due to edge effects or nearby gaps.

3. For standard gaps shorter than or equal to the maximum training length (72 hours), the model directly predicts the entire sequence using whatever context is available from both sides of the gap.

4. For extended gaps exceeding the maximum training length (e.g., the 191-hour gap in our dataset), the algorithm segments them into manageable chunks plus a remainder. Each chunk is predicted separately with appropriate context overlap, with predictions subsequently concatenated to form a complete sequence.

This approach allows our dynamic model to handle real-world gaps of any length, even those extending well beyond the maximum length used during training, and regardless of their position within the time series or proximity to other gaps. We developed both univariate and multivariate variants of our dynamic models:

- The univariate model (DynamicUniSeq2SeqXGB) uses only historical PM2.5 data as input, focusing solely on temporal patterns in the target variable.

- The multivariate model (DynamicMultiSeq2SeqXGB) incorporates additional features identified as significant in our correlation analysis: wind speed (Ff), wind direction (DD), air temperature, air humidity, hour of day, and season.

By incorporating position-aware padding, metadata enrichment, and unified training across gap lengths, our dynamic models can automatically adjust their processing to the specific characteristics of each gap encountered, regardless of its length or position within the time series. This approach eliminates the need for maintaining multiple specialized models and provides robust performance across the diverse gap scenarios encountered in real-world environmental monitoring data.

Our dynamic approach differs conceptually from existing adaptive methods in time series imputation. While RNN-based dynamic models like those proposed by Waqas [48] focus on sequential state updating for streaming data, and state-space models emphasize hidden state evolution, our method addresses the specific challenge of retrospective gap-filling with known boundaries. Unlike online dynamic methods that process data sequentially without future context, our approach leverages both pre- and post-gap information while dynamically adjusting context windows based on gap characteristics. This operational focus distinguishes our contribution from algorithmic innovations in neural architectures, emphasizing practical deployment flexibility over novel learning mechanisms.

## Implementation tools and environment

**Implementation tools and environment.** All computations were performed on a system with Intel(R) Core(TM) i7-14700KF CPU (14 physical cores, 28 logical cores) and NVIDIA GeForce RTX 4080 SUPER GPU with 16GB VRAM. The software environment included CUDA 12.4, NVIDIA driver version 572.16, and TensorFlow 2.18.0. This hardware

configuration provided sufficient computational resources for training all models within a reasonable timeframe, including the more complex deep learning architectures. The entire methodology was implemented using Python 3.12 with the libraries specified in Table 2.

Our gap-filling models were implemented using a combination of Python-based machine learning frameworks. Standard machine learning models (Random Forest, XGBoost) were developed using scikit-learn and XGBoost libraries, while neural network architectures were implemented in TensorFlow with Keras API. For specialized temporal convolutional networks (TCN), we utilized the tensorflow-tcn extension library.

All dynamic models were built using the XGBoost framework (version 1.6.1), specifically leveraging its MultiOutputRegressor functionality to handle variable-length outputs. This allowed for efficient training and prediction across different gap lengths while maintaining a consistent model architecture.

The code implementation prioritized computational efficiency and scalability. For neural network models, we employed batch processing with early stopping to optimize training time while preventing overfitting. Memory management techniques were implemented for processing longer time series, including efficient data generators and gradient accumulation where necessary.

**Integration with testing framework**

The dynamic models were seamlessly integrated into our unified testing infrastructure through the ImputationCombiner framework. This integration required several adaptations:

1. **Custom preprocessing pipelines**: We developed specialized data transformation functions to handle dynamic context sizing, padding, and metadata enrichment.

2. **Evaluation protocol adaptation**: The standard evaluation process was extended to properly handle variable-length predictions, ensuring fair comparison with fixed-length models.

3. **Results aggregation**: The performance metrics calculation system was modified to account for the dynamic nature of the models, particularly in handling the different confidence levels across various positions within filled gaps.

This integration allowed direct comparison between traditional fixed-context models and our dynamic approaches across identical testing conditions, ensuring valid performance comparisons.

**Application considerations.** The dynamic models offer several practical advantages for real-world deployment:

1. **Deployment simplicity**: A single dynamic model can replace multiple specialized models, reducing operational complexity and resource requirements.

2. **Robustness to varied gaps**: The models can handle irregularly distributed gaps of unpredictable lengths without requiring reconfiguration.

**Table 2. Primary software libraries used in this study.**

| Category | Libraries |
|---|---|
| Data processing | pandas 1.3.5, NumPy 1.21.5 |
| Machine learning | scikit-learn 1.0.2, XGBoost 1.6.1 |
| Deep learning | TensorFlow 2.9.1, Keras 2.9.0 |
| Specialized components | tensorflow-tcn 0.5.0 (for TCN models) |
| Visualization | Matplotlib 3.5.1, Seaborn 0.11.2 |
| Statistical analysis | SciPy 1.7.3 |
| Utility libraries | joblib 1.1.0 (for model serialization) |

3. **Performance consistency**: By adapting to gap characteristics automatically, these models maintain more consistent performance across various missing data patterns.

4. **Resource efficiency**: The dynamic context sizing optimizes computational resources by utilizing only the necessary context for each gap.

However, there are also implementation considerations to note:

1. **Training complexity**: The unified training process with variable-length gaps is more complex than training separate specialized models.

2. **Inference overhead**: The dynamic preprocessing pipeline adds a small computational overhead during inference compared to simpler fixed models.

3. **Data requirements**: Effective training requires examples of diverse gap patterns, potentially necessitating more comprehensive training data.

These tradeoffs should be considered when selecting appropriate models for specific application requirements, particularly in resource-constrained environments.

All models were benchmarked under identical conditions to ensure fair comparison of computational efficiency and performance. The complete implementation, including data preprocessing pipelines, model architectures, and evaluation protocols, has been organized into a modular codebase to facilitate reproducibility and extension to additional datasets and gap-filling scenarios.

## Results

### Correlation analysis and feature selection

**Correlation patterns between PM2.5 and meteorological parameters.** To identify relationships between PM2.5 concentrations and meteorological factors, we conducted a comprehensive correlation analysis using the preprocessed hourly dataset.

The correlation matrix in Fig 5 illustrates the interrelationships between PM2.5 and various meteorological parameters. The statistical significance of these correlations is indicated by asterisks, with most correlations showing high significance ($p < 0.001$).

The analysis revealed that air temperature (T') showed a weak negative correlation with PM2.5 concentrations ($r = -0.22$, $p < 0.001$), suggesting a modest trend of higher PM2.5 levels during colder periods. This inverse relationship aligns with the expected seasonal patterns of air pollution in urban environments, where winter conditions can contribute to the accumulation of particulate matter.

Interestingly, relative humidity (U') demonstrated only a weak positive correlation with PM2.5 levels ($r = 0.10$, $p < 0.001$), indicating a limited direct relationship between humidity and particulate matter concentrations in our study area. This finding differs from some previous studies that reported stronger humidity effects.

Wind speed (Ff) exhibited a moderate negative correlation with PM2.5 ($r = -0.30$, $p < 0.001$), confirming the expected dilution effect of stronger winds on air pollutants. This finding highlights the importance of atmospheric dispersion processes in regulating local PM2.5 concentrations. Wind direction (DD) showed a moderate positive correlation with PM2.5 ($r = 0.25$, $p < 0.001$), suggesting that certain wind directions may be associated with higher pollutant levels, possibly due to the transport of emissions from specific source areas.

Visibility (VV) showed a weak negative correlation with PM2.5 ($r = -0.15$, $p < 0.001$), which is consistent with the expected relationship between particulate matter and atmospheric transparency, though the association was not as strong as might be anticipated.

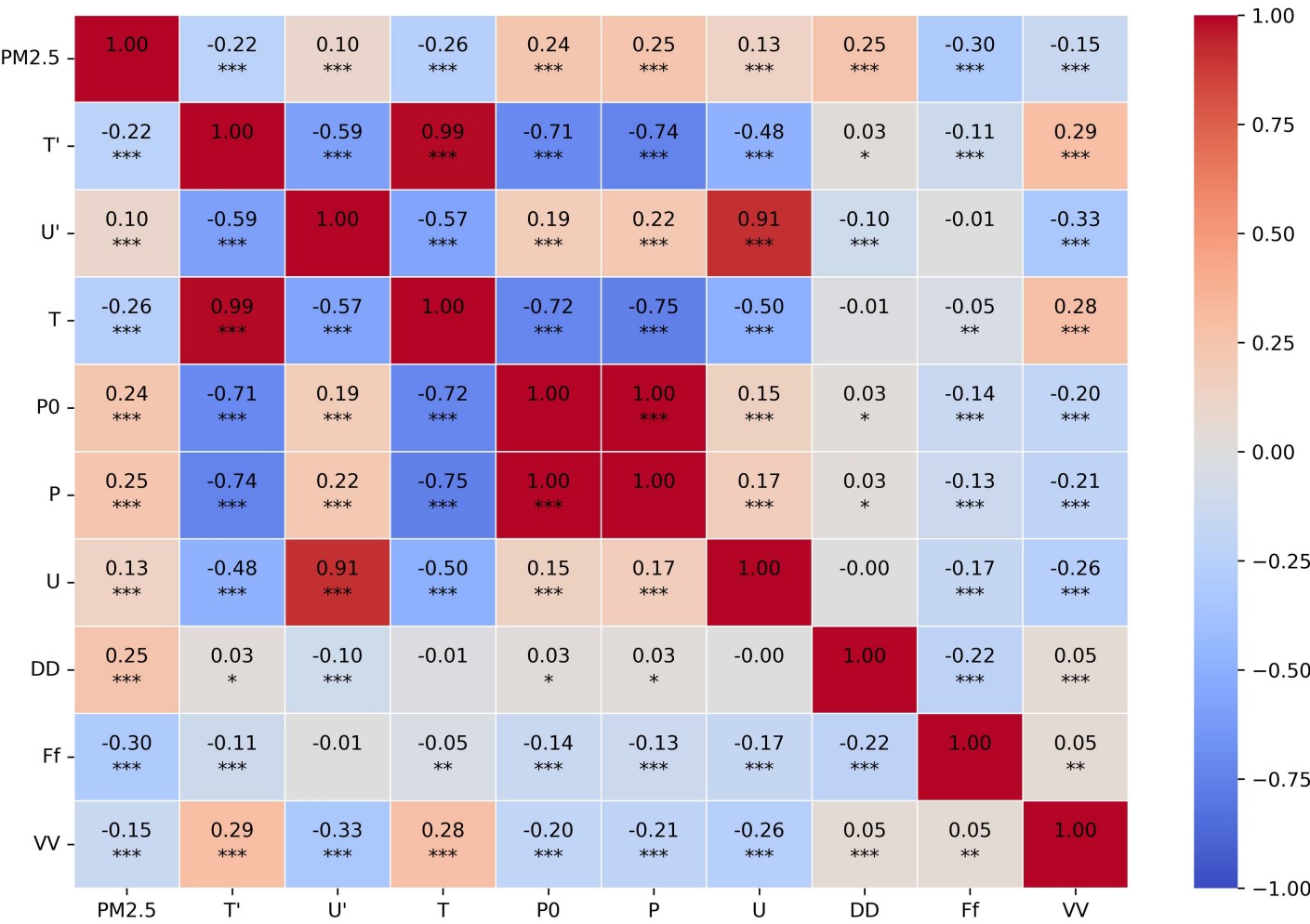

**Fig 5. Correlation matrix between PM2.5 and meteorological parameters.** The color intensity and numerical values represent Pearson correlation coefficients, with red indicating negative correlations and blue indicating positive correlations. Data measured with our station: PM2.5 (μg/m³), T'–air temperature (°C), U'–air humidity (%). Data from open source (rp5.com, weather archive in airport of Pavlodar, METAR [55]): T–air temperature (°C), P0–sea-level pressure (mmHg), P–station-level pressure (mmHg), U–relative humidity (%), DD–wind direction (compass points), Ff–wind speed (m/s), and VV–horizontal visibility range (km). *–$p < 0.05$ (statistically significant result), **–$p < 0.01$ (highly significant result), ***–$p < 0.001$ (extremely significant result).

The correlation analysis also revealed notable interactions between meteorological parameters themselves. For instance, air temperature and relative humidity displayed a strong negative correlation ($r = −0.59$, $p < 0.001$), while station-level pressure (P0) and sea-level pressure (P) were perfectly correlated ($r = 1.00$, $p < 0.001$), as expected. The strong correlation between air temperature measured on our station (T') and the temperature measured on station at the Airport (T) ($r = 0.99$, $p < 0.001$) indicates consistency between these two related measurements. These interrelationships among predictors are important considerations for multivariate modeling approaches.

Fig 6 presents the scatter plots of PM2.5 against four key meteorological parameters, visualizing the relationships discussed above. The plots clearly demonstrate the inverse relationship with temperature (panel a, $r = −0.26$) and wind speed (panel c, $r = −0.30$), as well as the weak positive association with humidity (panel b, $r = 0.13$). The visibility plot (panel d)

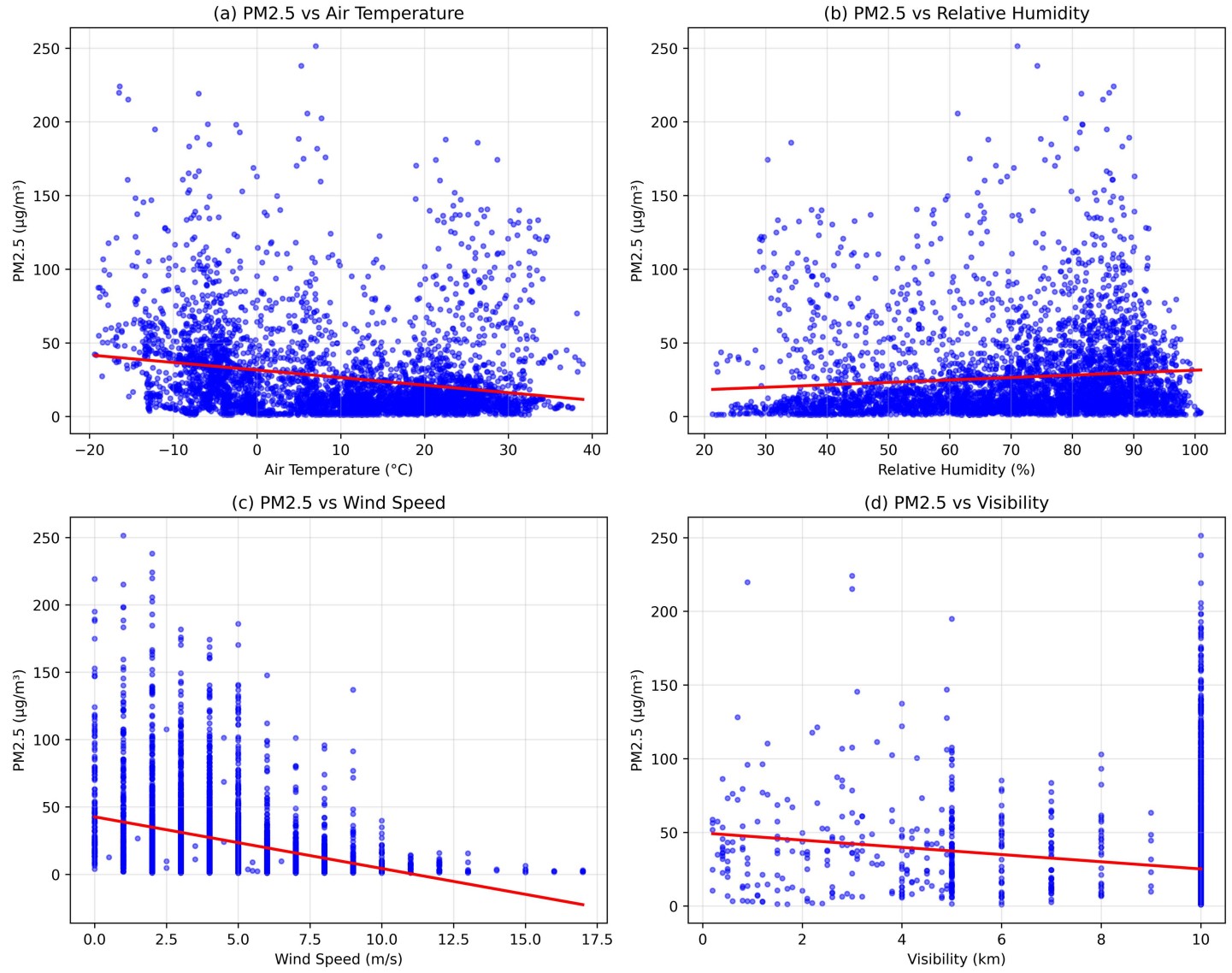

**Fig 6. Scatter plots showing relationships between PM2.5 and key meteorological variables: (a) air temperature, (b) relative humidity, (c) wind speed, and (d) visibility.** Blue points represent individual hourly measurements, and red lines indicate best-fit trends.

shows that as PM2.5 increases, visibility tends to decrease (r=−0.15), though with considerable variance at lower PM2.5 concentrations.

To investigate potential non-linear relationships that might not be captured by Pearson correlation coefficients, we applied logarithmic regression analysis using log-transformed PM2.5 concentrations as the dependent variable. The logarithmic transformation was applied to normalize the distribution of PM2.5 values and better capture non-linear relationships with meteorological parameters.

After addressing multicollinearity issues by removing highly correlated variables, the final logarithmic regression model explained approximately 29.1% of the variance in log-transformed PM2.5 concentrations ($R^2 = 0.291$, $F = 289.9$, $p < 0.001$).

This moderate explanatory power is consistent with the complex nature of particulate matter dynamics in urban environments, where multiple factors beyond meteorological conditions influence concentration levels.

The regression analysis identified wind parameters as the strongest predictors (Table 3).

Wind speed (Ff) demonstrated a highly significant negative relationship with PM2.5 levels ($\beta = -0.144$, $t = -23.59$, $p < 0.001$), confirming the important role of atmospheric dispersion in reducing particulate concentrations. Wind direction (DD) showed a significant positive association ($\beta = 0.047$, $t = 18.97$, $p < 0.001$), suggesting that certain wind directions may transport pollution from nearby emission sources.

Air temperature exhibited a significant negative relationship with PM2.5 ($\beta = -0.020$, $t = -9.29$, $p < 0.001$), supporting our correlation findings and highlighting the seasonal patterns of particulate pollution. Visibility (log-transformed) also maintained a significant negative association ($\beta = -0.252$, $t = -5.52$, $p < 0.001$), while relative humidity showed a weaker but still significant positive relationship ($\beta = 0.146$, $t = 2.37$, $p = 0.018$).

The selection of variable transformations was guided by both theoretical considerations and empirical testing. While logarithmic transformation was applied to several variables (air humidity, pressure, visibility) to normalize their distributions and linearize relationships, wind parameters were deliberately maintained in their original linear form for several compelling reasons.

Wind direction (DD) was kept in its original scale as it represents a directional measure where logarithmic transformation would distort its physical meaning. The circular nature of wind direction data, where values indicate specific compass bearings, makes linear representation most appropriate for capturing the transport of pollutants from particular source areas.

For wind speed (Ff), we tested both logarithmic and linear specifications in preliminary analyses. The linear form was ultimately selected based on three key factors: (1) it provided superior model fit, with the $R^2$ value decreasing from 0.291 to 0.262 when logarithmic transformation was applied; (2) atmospheric dispersion theory suggests a predominantly linear relationship between wind speed and pollutant dilution in urban environments; and (3) the presence of zero values (calm conditions) in the dataset would require artificial adjustments for logarithmic transformation, potentially introducing bias. The higher t-statistic for wind speed in its linear form (–23.59 versus −19.25 for the logarithmic form) further confirmed that this representation better captured its relationship with PM2.5 concentrations.

This mixed transformation approach, applying logarithmic transformation selectively where appropriate while maintaining linear relationships for wind parameters, produced the most statistically robust and physically interpretable model specification.

Diagnostic plots of this model revealed an approximately normal distribution of residuals, although with some tendency toward underestimation at extreme PM2.5 values. The scatter plot of predicted versus actual log(PM2.5) values

**Table 3. Logarithmic regression model results for PM2.5 concentrations.**

| Variable | Coefficient (β) | Std. Error | t-value | p-value |
|---|---|---|---|---|
| Constant | 5.206 | 14.027 | 0.371 | 0.711 |
| Air temperature | −0.020 | 0.002 | −9.291 | <0.001*** |
| Log(Air humidity) | 0.146 | 0.062 | 2.371 | 0.018* |
| Log(P) | −0.291 | 2.091 | −0.139 | 0.889 |
| Wind direction (DD) | 0.047 | 0.002 | 18.972 | <0.001*** |
| Wind speed (Ff) | −0.144 | 0.006 | −23.590 | <0.001*** |
| Log(Visibility) | −0.252 | 0.046 | −5.515 | <0.001*** |

Note: $R^2 = 0.291$, Adjusted $R^2 = 0.290$, $F_{(6,4238)} = 289.9$, $p < 0.001$, $n = 4245$. Significance levels:

*$p < 0.05$,

***$p < 0.001$.

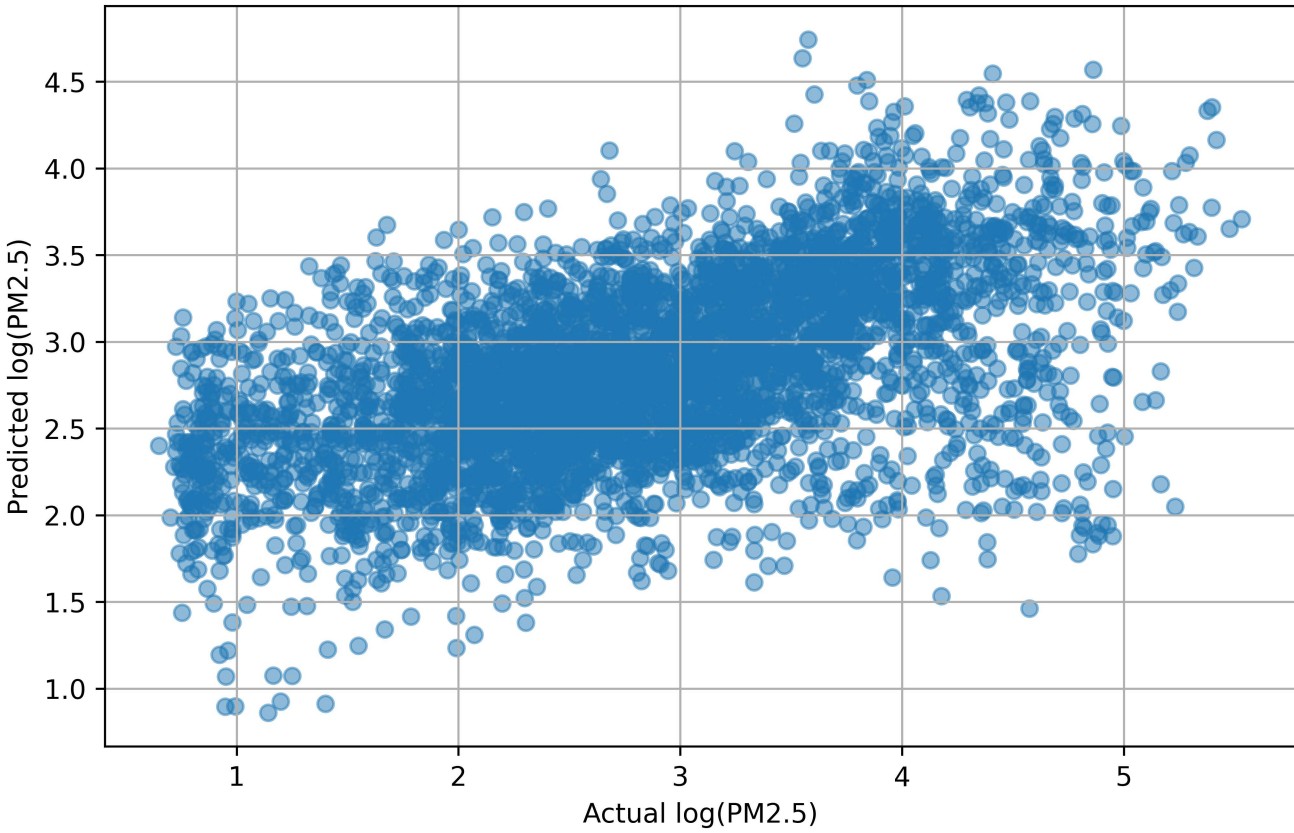

**Fig 7. Scatter plot of predicted versus actual log-transformed PM2.5 concentrations from the logarithmic regression model.**

(Fig 7) demonstrated the model's ability to capture general trends while also highlighting the inherent variability not explained by meteorological factors alone. The concentration of points around the central region (approximately 2.5–3.5 log(PM2.5)) demonstrates the model's tendency toward regression to the mean, with some underestimation of high concentration events and overestimation of low values. This pattern is typical for regression models addressing complex environmental phenomena and reflects the challenge of capturing extreme pollution events using meteorological variables alone.

The histogram on Fig 8 displays the distribution of model residuals, showing an approximately normal pattern centered around zero. This distribution suggests that the logarithmic transformation successfully addressed much of the non-linearity in the relationships between PM2.5 and meteorological parameters. The slight positive skew in the right tail indicates some systematic underestimation of higher pollution events, potentially related to episodic emission sources or complex atmospheric conditions not fully captured by the included predictors.

Fig 9 presents boxplots, a statistical visualization technique that provides an intuitive representation of data distribution. The rectangular box represents the interquartile range (IQR), encompassing the central 50% of observations between the 25th and 75th percentiles. The horizontal line within the box indicates the median value. Vertical "whiskers" extend to show the data's spread, typically to 1.5 times the IQR. Individual points beyond the whiskers represent outliers–data points that significantly deviate from the overall distribution.

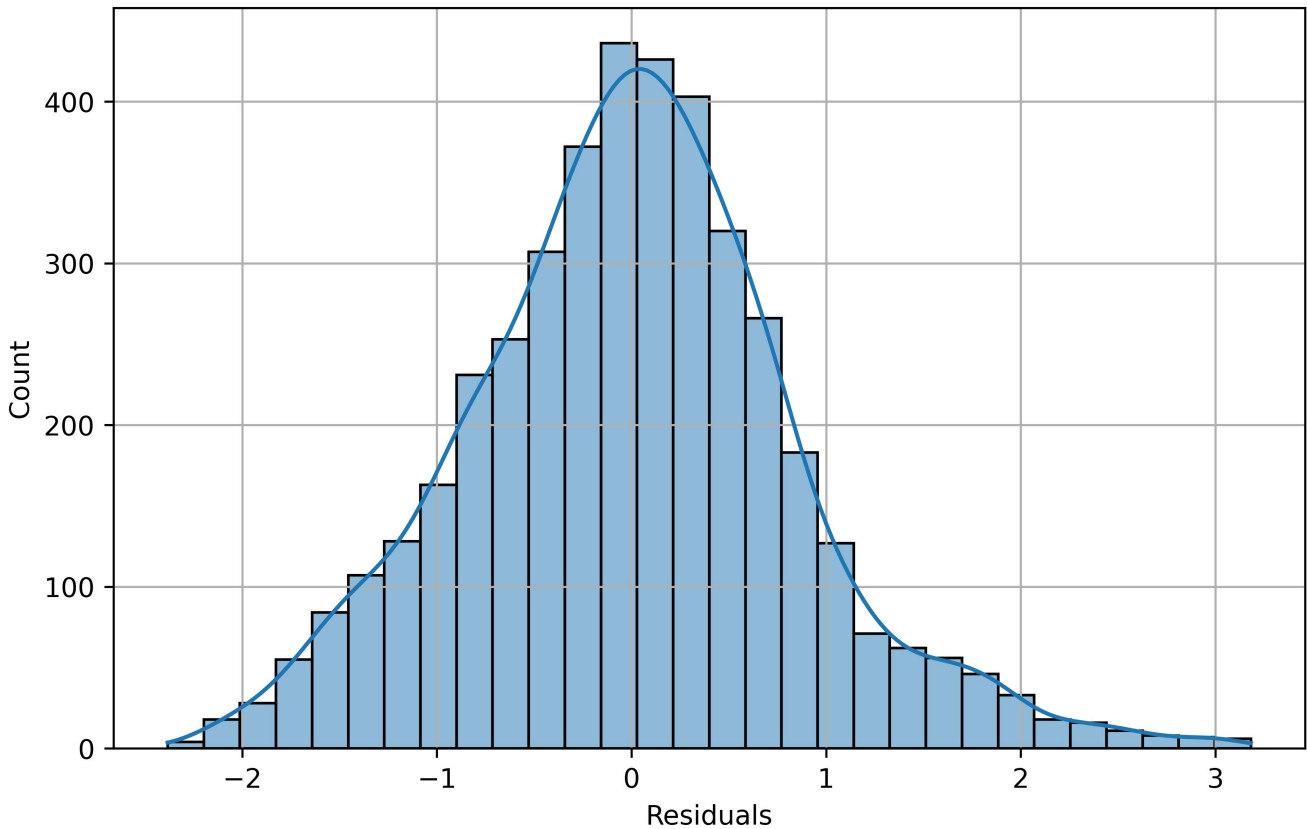

**Fig 8. Distribution of residuals from the logarithmic regression model of PM2.5 concentrations.**

The figure illustrates the distribution of PM2.5 concentrations across four meteorological parameters: air temperature, relative humidity, wind speed, and wind direction. These boxplots reveal nuanced relationships between environmental conditions and particulate matter concentrations.

Temperature analysis (Fig 9a) demonstrates a non-linear relationship between PM2.5 levels and air temperature. Notably, extremely low temperatures (between –25 and –20°C) correspond to elevated particulate concentrations. As temperatures rise to 0–10°C, a significant reduction in PM2.5 levels occurs, potentially attributable to decreased emission intensity and altered atmospheric dynamics. A moderate increase in concentrations is observed at temperatures above 25°C, which may result from enhanced photochemical reactions and specific particulate matter dispersion characteristics.

Relative humidity distribution (Fig 9b) reveals a pronounced trend of increasing PM2.5 concentrations with humidity elevation. The 20–50% humidity range exhibits relatively low particulate levels, while concentrations substantially increase at 80–100% humidity. This pattern likely stems from complex interactions involving particle deposition, atmospheric transformation processes, and meteorological conditions affecting pollutant dispersal.

Wind speed analysis (Fig 9c) illustrates a classic pattern of pollutant dispersion. Minimal wind speeds (0–2 m/s) correspond to maximum PM2.5 concentrations, indicative of air stagnation and pollutant accumulation. A sharp concentration decline occurs with wind speeds increasing to 4–6 m/s, demonstrating effective atmospheric mixing and particle transport. Concentrations stabilize at low levels when wind speeds exceed 8 m/s, highlighting wind regime's critical role in atmospheric pollution dynamics.

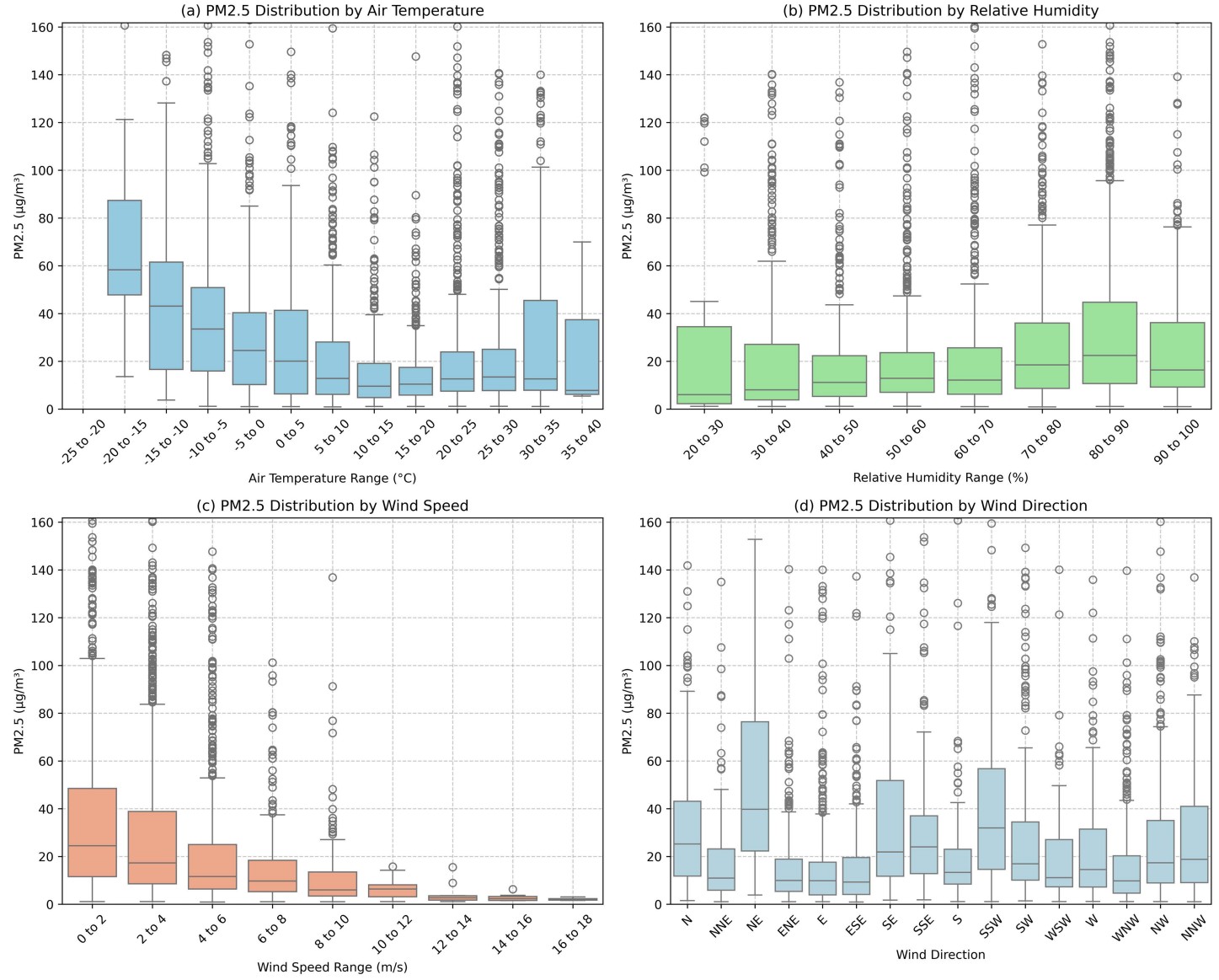

**Fig 9. Distribution of PM2.5 Concentrations Across Meteorological Parameters: (a) air temperature, (b) relative humidity, (c) wind speed, and (d) wind direction.** Boxplots display the median (central line), interquartile range (box), distribution (whiskers), and outliers (circles) of PM2.5 concentrations within each categorical bin. Wind directions indicate the direction from which the wind blows.

Wind direction distribution (Fig 9d) unveils spatial nuances of particulate matter concentration. Highest median concentrations are observed with northerly (N, NNE) and southerly (S, SSE) winds, potentially reflecting emission source locations or local topographical influences. Western wind directions (W, WNW) correspond to relatively lower concentrations. However, significant value dispersion across all directions underscores the complex, multifactorial nature of particulate matter atmospheric transport.

These findings illuminate the intricate interplay between meteorological parameters and PM2.5 concentrations, emphasizing the importance of comprehensive environmental monitoring and analysis.

**Temporal dependency analysis.** PM2.5 concentrations typically exhibit strong temporal patterns related to daily cycles, weekly human activity patterns, and seasonal variations. To quantify these temporal dependencies, we analyzed correlations between PM2.5 and various time-based features including hour of day, day of week, month, and season (where seasons were encoded as: 1-winter, 2-spring, 3-summer, 4-autumn).

The correlation heatmap (Fig 10) reveals several significant temporal patterns in PM2.5 concentrations. The Pearson correlation coefficients (lower triangle) show that PM2.5 has a weak positive correlation with month ($r=0.05$, $p<0.001$) and hour of day ($r=0.04$, $p<0.01$), while demonstrating a weak negative correlation with season ($r=-0.13$, $p<0.001$). The

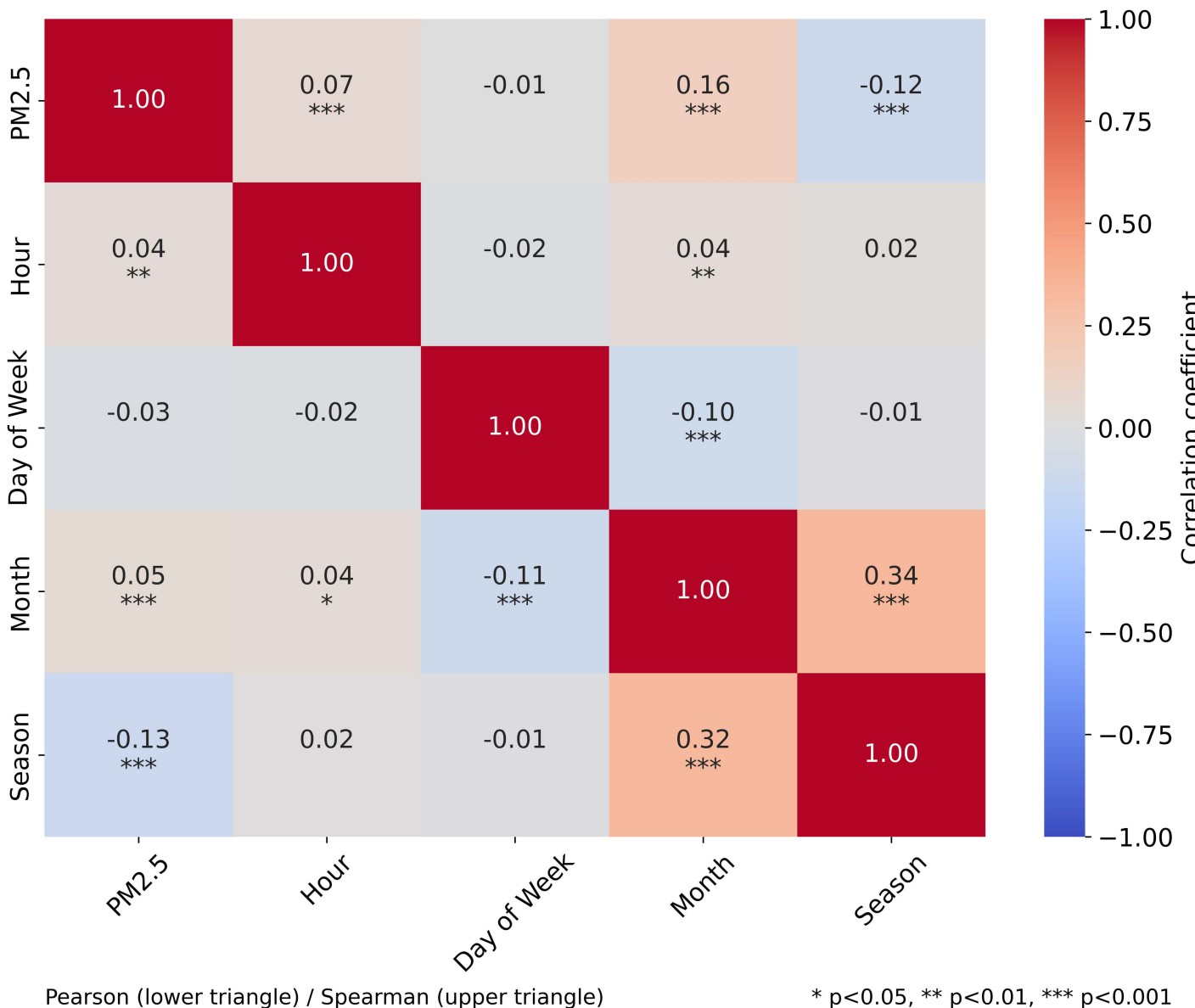

Pearson (lower triangle) / Spearman (upper triangle)　　　　* p<0.05, ** p<0.01, *** p<0.001

**Fig 10. Correlation heatmap between PM2.5 and temporal features.** The matrix shows both Pearson (lower triangle) and Spearman (upper triangle) correlation coefficients to capture both linear and rank-based relationships.

Spearman rank correlation (upper triangle) confirms these relationships, showing similar patterns with slightly different magnitudes: positive correlations with hour (r=0.07, p<0.001) and month (r=0.16, p<0.001), and a negative correlation with season (r=−0.12, p<0.001). The day of week shows no significant correlation with PM2.5, suggesting that weekday patterns do not significantly affect pollution levels. The matrix also reveals a strong positive correlation between month and season (r=0.34, p<0.001), which is expected given their temporal relationship. These results suggest that pollution levels exhibit weak but statistically significant monthly and seasonal patterns, as well as diurnal (hour of day) variations, but do not follow weekly cycles.

Fig 11 illustrates the temporal patterns in PM2.5 concentrations across different time scales. Panel (a) reveals notable peaks occurring during morning (7:00–9:00) and evening (18:00–21:00) hours, coinciding with periods of increased urban activity and traffic. The narrowest confidence intervals during daytime hours (10:00–16:00) indicate more stable and predictable PM2.5 levels during these periods.

Panel (b) shows significant variations in PM2.5 concentrations throughout the year, with highest levels observed in January and December (winter months) and lowest in September. The data for February, March, and April are not available (n.d.). The observed pattern indicates approximately 35–40% higher concentrations in winter months compared to summer months (June-August). This seasonal effect is consistent with the negative correlation between PM2.5 and temperature noted earlier and can be attributed to multiple factors including reduced atmospheric mixing height in winter, increased emissions from heating systems, and more frequent temperature inversions that trap pollutants near the surface.

Panel (c) displays varying PM2.5 levels throughout the week, with the highest concentration on Tuesday and lowest on Friday. The pattern does not show a clear weekday-weekend distinction, suggesting that local factors beyond typical work-related activities may influence daily PM2.5 variations in this area.

Temporal analysis revealed significant patterns in PM2.5 concentrations, with distinct diurnal variations showing peaks during morning and evening hours, as well as notable seasonal differences with higher levels in winter months compared to summer. Based on these findings and considering data limitations for several months, hour of day and a derived seasonal indicator were incorporated as temporal features in multivariate models to capture cyclical patterns that might influence gap-filling performance.

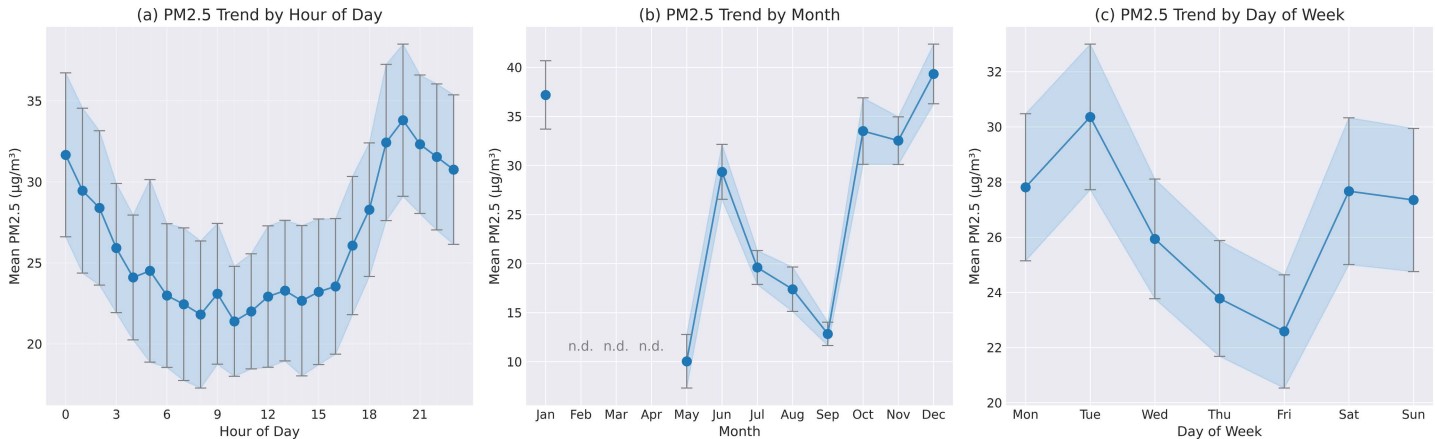

**Fig 11. Hourly, monthly, and day-of-week trends in PM2.5 concentrations with 95% confidence intervals: (a) mean PM2.5 by hour of day, (b) mean PM2.5 by month, and (c) mean PM2.5 by day of week.** Error bars represent 95% confidence intervals around the means. n.d.–no data.

Based on both correlation analysis and logarithmic regression results, we selected the following parameters for inclusion in our multivariate gap-filling methods: wind speed (Ff), wind direction (DD), air temperature, relative humidity, hour of day, and season. This combination of meteorological and temporal variables provided a balanced approach to capturing both physical atmospheric processes and anthropogenic patterns affecting PM2.5 concentrations.

## Imputation methods evaluation

**Univariate model evaluation for 12-hour gaps.** The initial phase of our study focused on evaluating the performance of univariate models for filling artificial 12-hour gaps in PM2.5 time series data. These models utilize only historical PM2.5 values without incorporating additional meteorological or temporal variables. We systematically compared 36 different approaches across the methodological hierarchy described in the Methods section.

Performance evaluation was conducted through synthetic gap testing with a 5% missing fraction. For each model, we performed five independent runs with different random seeds to ensure statistical robustness. The performance metrics (MAE, RMSE, $R^2$, and MAPE) were averaged across all runs to obtain reliable estimates of each model's capabilities.

The experimental results revealed substantial performance variations across the different univariate approaches. Table 4 presents the performance metrics for all tested univariate models when filling 12-hour gaps.

In Table 4 models in each category are ranked based on MAE, with the best performing model (lowest MAE) highlighted in bold. MAE was selected as the primary metric for comparison due to its direct interpretability in physical units ($\mu g/m^3$) and robustness to outliers. While other metrics provide complementary performance insights, MAE offers the most intuitive measure of prediction accuracy for PM2.5 concentration values.

Among the simple statistical methods, Local Mean Imputation demonstrated the most favorable performance with MAE of $7.264 \pm 0.352$ $\mu g/m^3$ and RMSE of $9.573 \pm 0.581$ $\mu g/m^3$. This suggests that even basic approaches that consider local temporal context can achieve reasonable accuracy for moderate-length gaps. In contrast, Simple Mean and Median imputations performed substantially worse, with MAE values of $14.351 \pm 0.438$ $\mu g/m^3$ and $14.245 \pm 0.433$ $\mu g/m^3$ respectively, highlighting the importance of incorporating temporal context in the imputation process.

Window-based interpolation methods showed variable performance. Linear Window Interpolation achieved a respectable MAE of $7.842 \pm 0.477$ $\mu g/m^3$, while Polynomial and B-spline interpolations performed slightly better with MAE values of $7.523 \pm 0.406$ $\mu g/m^3$ and $7.488 \pm 0.474$ $\mu g/m^3$ respectively. These methods effectively captured local trends but exhibited limitations when dealing with more complex temporal patterns.

Among the machine learning approaches, tree-based models demonstrated strong performance. The bidirectional XGB Seq2Seq model achieved the lowest overall MAE ($5.231 \pm 0.292$ $\mu g/m^3$) and RMSE ($7.063 \pm 0.481$ $\mu g/m^3$) among all univariate models, with an $R^2$ of $0.773 \pm 0.033$. Its unidirectional counterpart (UniSeq2Seq XGB) showed slightly reduced performance with an MAE of $5.843 \pm 0.318$ $\mu g/m^3$ and RMSE of $7.723 \pm 0.509$ $\mu g/m^3$. This performance difference between bidirectional and unidirectional approaches was consistent across multiple model architectures, confirming the advantage of incorporating both preceding and subsequent temporal context when filling gaps.

Neural network-based approaches demonstrated competitive but slightly lower performance compared to tree-based models for 12-hour gaps. Among the neural architectures, LSTM-based models achieved the best results, with the bidirectional LSTM Seq2Seq model yielding an MAE of $5.976 \pm 0.327$ $\mu g/m^3$ and RMSE of $7.921 \pm 0.483$ $\mu g/m^3$. Temporal Convolutional Networks (TCN) also performed well, with the bidirectional TCN Seq2Seq model achieving an MAE of $6.211 \pm 0.344$ $\mu g/m^3$.

The performance gap between autoregressive (AR) and sequence-to-sequence (Seq2Seq) approaches was particularly evident in the results. Seq2Seq models consistently outperformed their autoregressive counterparts across all architectural configurations. For instance, the XGB Seq2Seq model achieved a 12% lower MAE compared to the XGB Autoreg model ($5.231 \pm 0.292$ $\mu g/m^3$ vs. $5.936 \pm 0.312$ $\mu g/m^3$). This pattern indicates that direct prediction of the entire gap is more effective than recursive one-step-ahead forecasting for 12-hour gaps, likely due to the error accumulation inherent in the autoregressive approach.

**Table 4. Performance comparison of univariate models for filling 12-hour gaps in PM2.5 data (mean±standard deviation across five experimental runs).**

| Method Category | Model | MAE (µg/m³) | RMSE (µg/m³) | R2 | MAPE (%) | Run Time (s) |
|---|---|---|---|---|---|---|
| **Simple Methods** | **Simple Imputer Median** | 14.245±0.433 | 18.154±0.574 | 0.105±0.018 | 71.938±3.201 | 0.49 |
| | Simple Imputer Mean | 14.351±0.438 | 18.273±0.583 | 0.097±0.017 | 72.434±3.217 | 0.48 |
| **Local Methods** | **Local Imputation Mean** | 7.264±0.352 | 9.573±0.581 | 0.668±0.041 | 38.592±2.842 | 0.64 |
| | Local Imputation Median | 7.339±0.358 | 9.642±0.586 | 0.663±0.042 | 39.046±2.856 | 0.64 |
| **Window Methods** | **B-spline Window Interpolation** | 7.488±0.474 | 9.817±0.635 | 0.653±0.046 | 39.052±3.143 | 0.81 |
| | Polynomial Window Interpolation | 7.523±0.406 | 9.851±0.605 | 0.651±0.044 | 39.125±2.927 | 0.95 |
| | Linear Window Interpolation | 7.842±0.477 | 10.273±0.710 | 0.625±0.055 | 40.321±3.413 | 0.73 |
| | ARIMA Imputation | 7.916±0.493 | 10.356±0.716 | 0.620±0.056 | 40.873±3.478 | 5.65 |
| **Unidirectional Autoregressive** | **UniAR XGB** | 6.257±0.327 | 8.172±0.477 | 0.752±0.033 | 33.046±2.569 | 7.25 |
| | UniAR RF | 6.472±0.338 | 8.436±0.488 | 0.738±0.034 | 34.166±2.612 | 9.21 |
| | UniAR GRU | 6.629±0.345 | 8.632±0.494 | 0.728±0.035 | 34.962±2.642 | 30.83 |
| | UniAR LSTM | 6.683±0.347 | 8.697±0.497 | 0.724±0.035 | 35.218±2.651 | 31.36 |
| | UniAR CNN | 6.732±0.348 | 8.757±0.499 | 0.721±0.035 | 35.463±2.654 | 28.52 |
| | UniAR RNN | 6.754±0.351 | 8.785±0.501 | 0.719±0.035 | 35.573±2.663 | 29.91 |
| | UniAR TCN | 6.692±0.347 | 8.709±0.498 | 0.724±0.035 | 35.266±2.652 | 32.54 |
| **Unidirectional Seq2Seq** | **UniSeq2Seq XGB** | 5.843±0.318 | 7.723±0.509 | 0.765±0.032 | 30.923±2.562 | 7.01 |
| | UniSeq2Seq RF | 5.972±0.319 | 7.884±0.465 | 0.764±0.032 | 31.582±2.509 | 8.24 |
| | UniSeq2Seq LSTM | 6.053±0.323 | 7.903±0.466 | 0.762±0.032 | 31.981±2.528 | 33.57 |
| | UniSeq2Seq CNN | 6.127±0.326 | 7.998±0.469 | 0.759±0.032 | 32.372±2.541 | 28.15 |
| | UniSeq2Seq GRU | 6.117±0.325 | 7.985±0.469 | 0.760±0.032 | 32.317±2.539 | 32.84 |
| | UniSeq2Seq TCN | 6.153±0.327 | 8.032±0.471 | 0.758±0.033 | 32.507±2.547 | 34.96 |
| | UniSeq2Seq RNN | 6.328±0.336 | 8.242±0.479 | 0.748±0.033 | 33.422±2.585 | 32.12 |
| **Bidirectional Autoregressive** | **XGB Autoreg** | 5.936±0.312 | 7.844±0.469 | 0.765±0.032 | 31.392±2.476 | 7.36 |
| | CNN Autoreg | 6.087±0.322 | 7.942±0.467 | 0.762±0.032 | 32.167±2.529 | 27.23 |
| | LSTM Autoreg | 6.067±0.321 | 7.917±0.467 | 0.763±0.032 | 32.057±2.524 | 33.72 |
| | RF Autoreg | 6.126±0.321 | 8.065±0.478 | 0.756±0.033 | 32.367±2.519 | 8.64 |
| | GRU Autoreg | 6.126±0.325 | 7.996±0.470 | 0.759±0.033 | 32.364±2.544 | 32.91 |
| | TCN Autoreg | 6.156±0.326 | 8.036±0.472 | 0.758±0.033 | 32.522±2.548 | 35.07 |
| | RNN Autoreg | 6.213±0.329 | 8.103±0.474 | 0.754±0.033 | 32.819±2.557 | 30.58 |
| **Bidirectional Seq2Seq** | **XGB Seq2Seq** | 5.231±0.292 | 7.063±0.481 | 0.773±0.033 | 27.716±2.367 | 8.21 |
| | RF Seq2Seq | 5.512±0.298 | 7.373±0.444 | 0.786±0.031 | 29.203±2.416 | 8.42 |
| | CNN Seq2Seq | 5.875±0.315 | 7.764±0.460 | 0.768±0.032 | 31.094±2.486 | 27.18 |
| | LSTM Seq2Seq | 5.976±0.327 | 7.921±0.483 | 0.764±0.032 | 31.592±2.557 | 33.64 |
| | RNN Seq2Seq | 6.037±0.322 | 7.934±0.467 | 0.761±0.033 | 31.897±2.523 | 30.48 |
| | GRU Seq2Seq | 6.067±0.332 | 8.041±0.489 | 0.758±0.033 | 32.061±2.587 | 32.87 |
| | TCN Seq2Seq | 6.211±0.344 | 8.246±0.502 | 0.748±0.034 | 32.815±2.642 | 35.12 |

Note: The best performing model in each category is highlighted in bold. Performance metrics include MAE, RMSE, $R^2$, MAPE, and average run time per test (in seconds).

The performance of different models can be further analyzed through visualizations. Fig 12 shows scatter plots of predicted versus actual values for six representative models spanning the entire performance spectrum. The tree-based Seq2Seq models (XGB Seq2Seq and RF Seq2Seq) demonstrated the best alignment with the ideal prediction line, though both tend to underestimate the highest PM2.5 concentrations (>125 µg/m³). The XGB Seq2Seq model (Fig 12a) shows

tighter clustering around the perfect prediction line compared to RF Seq2Seq (Fig 12b), which displays more scattered predictions, particularly for higher values. Both the unidirectional XGB model (Fig 12c) and autoregressive XGB model (Fig 12d) reveal similar patterns but with gradually increasing dispersion.

The Local Imputation Mean method (Fig 12e) exhibits distinct horizontal banding patterns, indicating limited variability in predictions and substantial underestimation of high values. Most strikingly, the Simple Imputer Mean (Fig 12f) produces essentially constant predictions around 27 µg/m³ regardless of the true values, demonstrating complete inability to capture the variability in PM2.5 concentrations. This visual comparison dramatically illustrates the substantial performance gap between sophisticated and basic approaches, and confirms the quantitative metrics presented in Table 4.

Fig 13 illustrates examples of how the different models fill a 12-hour gap in a PM2.5 time series. The advanced models (XGB Seq2Seq, RF Seq2Seq, UniSeq2Seq XGB, and XGB Autoreg) all attempt to predict the temporal dynamics within the gap, though in this particular example they predict an increase in PM2.5 levels in the middle of the gap while the actual values show a decrease. This highlights the challenge of accurately capturing complex temporal patterns, even for sophisticated models. XGB Seq2Seq (Fig 13a) shows a prediction pattern that, while not perfect, more closely follows the general shape of the surrounding data.

In stark contrast, the Local Imputation Mean method (Fig 13e) simply fills the entire gap with a constant value around 56 µg/m³, which is the average of values in the surrounding window. Similarly, the Simple Imputer Mean (Fig 13f) fills the gap with a constant value around 27 µg/m³, which represents the global mean of the entire dataset. These simplistic approaches completely fail to capture any temporal dynamics within the gap, further demonstrating why they perform poorly on the evaluation metrics.

This visualization clearly demonstrates the fundamental difference between more advanced modeling approaches that attempt to learn and reproduce temporal patterns versus basic methods that simply insert constant values. It also reveals that even the best-performing models can misidentify trend directions in complex time series, suggesting areas for further improvement in gap-filling methodologies.

Examining computational efficiency, we observed significant differences in execution time across models. Simple statistical methods completed each experimental run in less than 1 second, while complex neural network architectures required up to 35 seconds per run. Tree-based models offered an attractive balance between performance and computational efficiency, with the XGB Seq2Seq model completing each run in approximately 8.2 seconds while delivering the best overall accuracy.

An important observation from the error analysis was that most models exhibited systematically larger errors during periods of high PM2.5 concentration variability. This pattern suggests that capturing rapid fluctuations remains challenging across all model types, even for those with sophisticated temporal modeling capabilities.

In summary, the evaluation of univariate models for 12-hour gaps demonstrated that bidirectional tree-based models, particularly XGB Seq2Seq, offered the best combination of accuracy and efficiency. The results confirmed that leveraging both past and future context through bidirectional approaches consistently improves gap-filling performance compared to unidirectional methods. Additionally, the direct sequence-to-sequence prediction approach proved more effective than recursive autoregressive forecasting for gaps of this duration.

**Comparative analysis of univariate and multivariate gap-filling models for PM2.5 measurements.** To evaluate the effectiveness of gap-filling strategies for PM2.5 measurements, we conducted a comprehensive benchmark of both univariate and multivariate models across five synthetic gap lengths: 5, 12, 24, 48, and 72 hours. These gap lengths were chosen to simulate a range of realistic data missingness scenarios, from short-term sensor failures to extended outages. Each model was tested over five runs with a missing fraction of 0.05, ensuring robust statistical reliability. The univariate models relied solely on historical PM2.5 data, while the multivariate models incorporated additional environmental features such as air temperature, humidity, atmospheric pressure, wind speed, and temporal indicators (hour and season). Performance was assessed using Mean Absolute Error (MAE), Root Mean Square Error (RMSE), Coefficient of

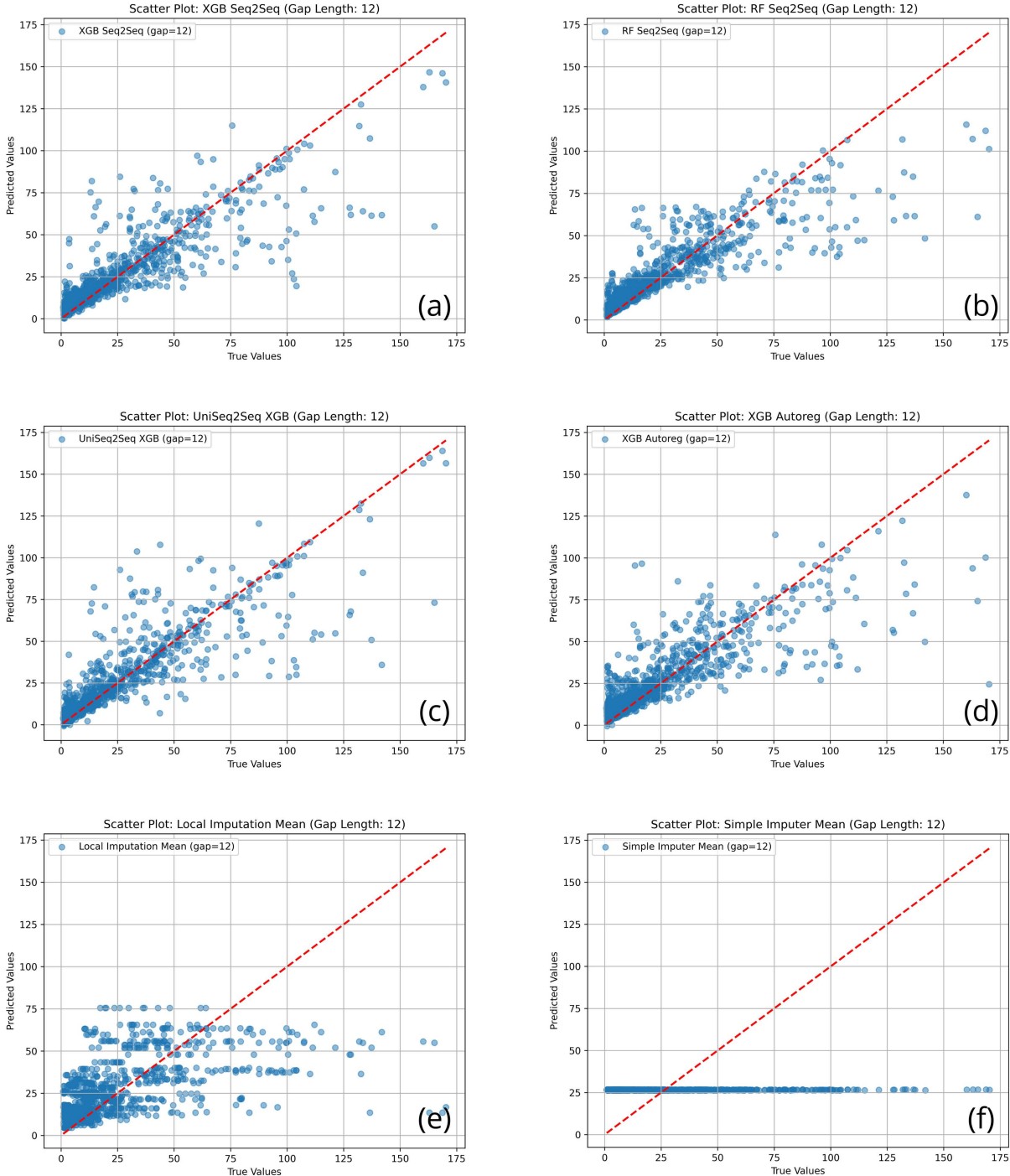

**Fig 12. Scatter plots of predicted versus actual PM2.5 values (µg/m³) for selected models representing different methodological approaches and performance levels: (a) XGB Seq2Seq (best overall model), (b) RF Seq2Seq (second-best model), (c) UniSeq2Seq XGB (best unidirectional model), (d) XGB Autoreg (best autoregressive model), (e) Local Imputation Mean (best simple method), and (f) Simple Imputer Mean (baseline method).** The red dashed line represents perfect prediction.

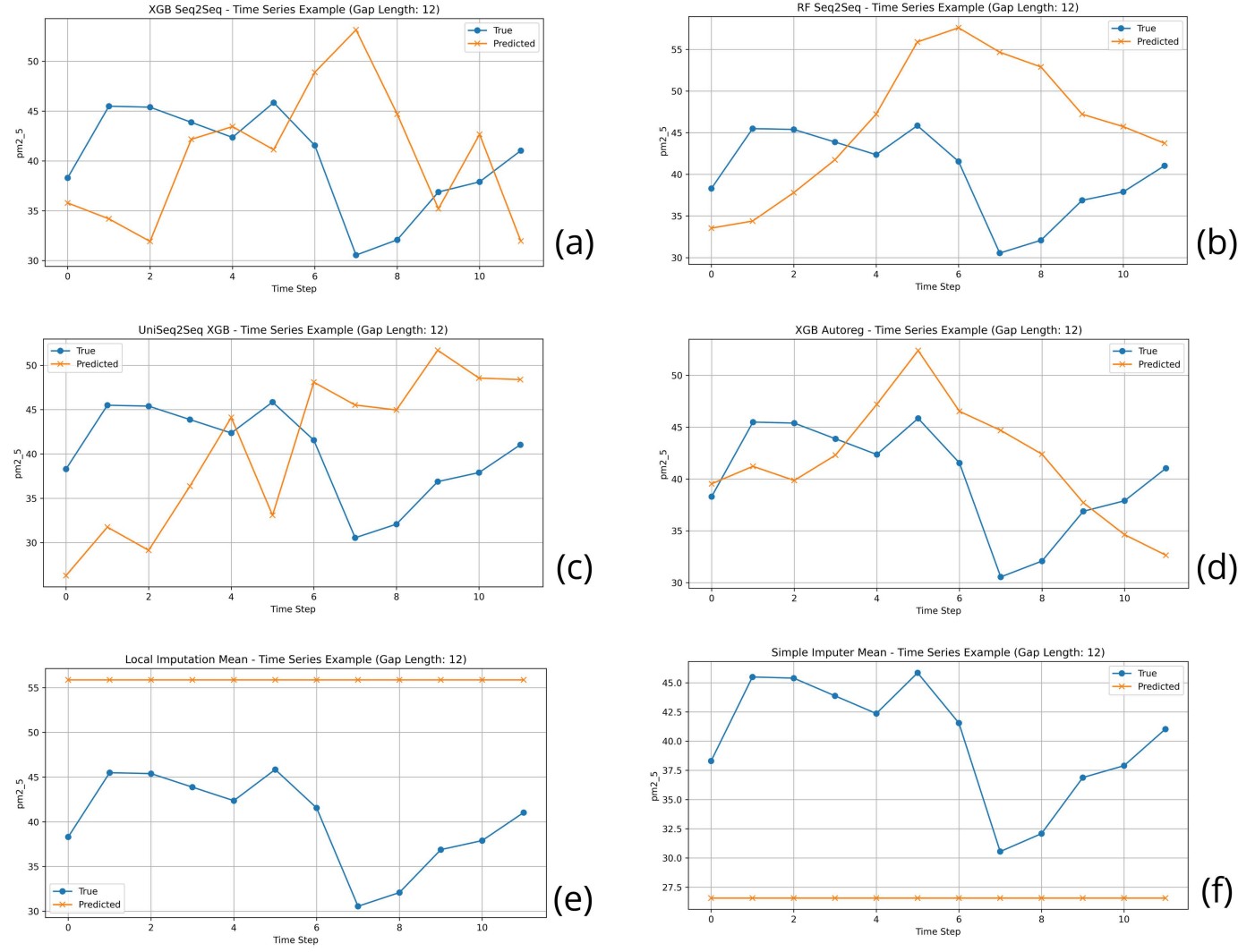

**Fig 13. Examples of 12-hour gap filling in PM2.5 time series (μg/m³) by representative models with varying performance: (a) XGB Seq2Seq, (b) RF Seq2Seq, (c) UniSeq2Seq XGB, (d) XGB Autoreg, (e) Local Imputation Mean, and (f) Simple Imputer Mean.** The blue line shows actual values, while the red line indicates predicted values within the gap.

Determination ($R^2$), and Mean Absolute Percentage Error (MAPE), alongside computational efficiency metrics like average run time.

The results, summarized in Tables 5 and 6, provide critical insights into how these modeling approaches perform under varying conditions and highlight the advantages of leveraging multivariate data for environmental time series reconstruction.

The performance of gap-filling models for PM2.5 measurements was assessed across a range of gap lengths, highlighting the relative strengths of multivariate approaches compared to univariate ones, especially as gaps extend in duration. For short gaps of 5 hours, univariate models delivered a mean absolute error (MAE) of around 7.87 μg/m³ and a coefficient of determination ($R^2$) of 0.745 when averaged across all models tested. The standout univariate performers, UniSeq2Seq XGB

**Table 5. Performance metrics of univariate gap-filling models across different gap lengths.**

| Gap Length | Model | MAE | RMSE | R2 | MAPE | Avg Run Time (s) |
|---|---|---|---|---|---|---|
| 5 | UniAR XGB | 8.648±1.719 | 15.369±4.269 | 0.711±0.131 | 54.846±8.034 | 0.250 |
| 5 | UniSeq2Seq CNN | 11.243±2.164 | 18.255±3.049 | 0.593±0.126 | 75.414±10.505 | 10.862 |
| 5 | UniSeq2Seq RF | 6.088±1.326 | 11.683±3.741 | 0.827±0.093 | 34.511±11.311 | 39.469 |
| 5 | **UniSeq2Seq XGB** | **5.286±1.616** | **10.204±4.315** | **0.862±0.097** | **29.721±2.447** | **0.827** |
| 5 | UniSeq2Seq LSTM | 13.960±1.567 | 21.803±3.181 | 0.425±0.141 | 118.154±27.038 | 40.836 |
| 5 | XGB Autoreg | 6.066±1.366 | 10.959±3.836 | 0.847±0.090 | 35.994±3.115 | 0.248 |
| 5 | RF Seq2Seq | 6.103±1.375 | 11.526±4.134 | 0.830±0.097 | 37.901±9.888 | 9.677 |
| 5 | **XGB Seq2Seq** | **5.555±1.480** | **10.232±4.245** | **0.863±0.093** | **33.944±3.585** | **0.961** |
| 12 | UniAR XGB | 12.041±3.296 | 18.962±4.779 | 0.497±0.226 | 134.509±35.115 | 0.125 |
| 12 | UniSeq2Seq CNN | 13.165±2.069 | 20.122±3.599 | 0.452±0.150 | 145.655±37.267 | 9.365 |
| 12 | UniSeq2Seq RF | 8.481±2.169 | 14.627±3.659 | 0.698±0.150 | 68.367±20.698 | 95.571 |
| 12 | **UniSeq2Seq XGB** | **7.279±2.199** | **13.700±4.190** | **0.725±0.166** | **57.627±11.443** | **1.557** |
| 12 | UniSeq2Seq LSTM | 14.983±2.210 | 22.488±2.664 | 0.322±0.101 | 175.483±40.091 | 37.919 |
| 12 | XGB Autoreg | 9.263±2.664 | 15.380±4.119 | 0.663±0.167 | 102.663±30.021 | 0.386 |
| 12 | RF Seq2Seq | 7.905±2.172 | 13.530±3.743 | 0.741±0.133 | 68.491±14.468 | 9.591 |
| 12 | **XGB Seq2Seq** | **7.679±2.179** | **13.608±3.834** | **0.734±0.149** | **68.464±11.228** | **2.517** |
| 24 | UniAR XGB | 13.858±4.935 | 21.753±8.159 | 0.526±0.200 | 121.085±31.192 | 0.161 |
| 24 | UniSeq2Seq CNN | 15.350±3.514 | 24.237±6.044 | 0.413±0.144 | 104.465±24.607 | 8.773 |
| 24 | UniSeq2Seq RF | 8.265±3.034 | 14.980±7.726 | 0.778±0.141 | 62.150±31.743 | 190.278 |
| 24 | **UniSeq2Seq XGB** | **6.969±3.111** | **14.383±8.672** | **0.789±0.159** | **56.681±24.642** | **3.361** |
| 24 | UniSeq2Seq LSTM | 16.326±3.013 | 25.605±4.738 | 0.337±0.132 | 121.070±42.326 | 36.566 |
| 24 | XGB Autoreg | 11.408±3.861 | 18.794±7.001 | 0.643±0.156 | 98.888±33.528 | 0.319 |
| 24 | RF Seq2Seq | 8.124±3.014 | 14.903±7.935 | 0.777±0.151 | 63.980±29.949 | 10.628 |
| 24 | **XGB Seq2Seq** | **7.188±2.557** | **14.060±8.426** | **0.798±0.159** | **62.641±24.640** | **3.923** |
| 48 | UniAR XGB | 18.940±5.323 | 27.538±6.283 | −0.298±0.859 | 228.535±167.837 | 0.182 |
| 48 | UniSeq2Seq CNN | 14.232±2.741 | 21.398±4.066 | 0.306±0.258 | 163.339±96.131 | 8.986 |
| 48 | UniSeq2Seq RF | 7.337±2.709 | 12.876±5.877 | 0.754±0.150 | 67.538±38.641 | 394.563 |
| 48 | **UniSeq2Seq XGB** | **7.030±3.987** | **11.772±7.532** | **0.764±0.230** | **61.073±37.356** | **7.928** |
| 48 | UniSeq2Seq LSTM | 15.063±3.237 | 24.118±4.723 | 0.160±0.198 | 179.994±92.677 | 46.551 |
| 48 | XGB Autoreg | 15.020±1.863 | 21.966±2.309 | 0.264±0.267 | 183.624±123.315 | 0.354 |
| 48 | RF Seq2Seq | 6.986±2.558 | 12.569±6.197 | 0.768±0.151 | 67.563±35.175 | 12.860 |
| 48 | **XGB Seq2Seq** | **6.971±3.126** | **11.842±6.810** | **0.784±0.167** | **67.788±37.794** | **7.931** |
| 72 | UniAR XGB | 25.310±8.525 | 33.711±8.741 | −0.167±0.572 | 326.139±233.860 | 0.132 |
| 72 | UniSeq2Seq CNN | 22.699±5.474 | 30.728±7.645 | 0.142±0.051 | 221.839±150.769 | 9.148 |
| 72 | UniSeq2Seq RF | 12.611±9.975 | 16.922±12.684 | 0.718±0.315 | 170.068±199.062 | 596.277 |
| 72 | **UniSeq2Seq XGB** | **10.144±10.943** | **13.673±13.888** | **0.766±0.328** | **138.716±215.260** | **9.096** |
| 72 | UniSeq2Seq LSTM | 22.663±6.100 | 30.156±7.835 | 0.153±0.189 | 263.800±190.175 | 53.731 |
| 72 | XGB Autoreg | 21.867±5.799 | 29.344±8.247 | 0.166±0.269 | 248.363±148.294 | 0.282 |
| 72 | RF Seq2Seq | 11.680±8.582 | 16.746±12.514 | 0.723±0.309 | 127.543±131.021 | 16.244 |
| 72 | **XGB Seq2Seq** | **10.624±10.059** | **15.371±14.310** | **0.730±0.367** | **103.710±125.179** | **9.572** |

and XGB Seq2Seq, posted notably lower MAEs of 5.286±1.616 µg/m³ and 5.555±1.480 µg/m³, respectively, alongside $R^2$ values of 0.862±0.097 and 0.863±0.093. In contrast, multivariate models for the same gap length averaged a higher MAE of approximately 9.35 µg/m³ and a lower $R^2$ of 0.606 across all models. However, the top multivariate model, UniSeq2Seq XGB

**Table 6. Performance metrics of multivariate gap-filling models across different gap lengths.**

| Gap Length | Model | MAE | RMSE | R2 | MAPE | Avg Run Time (s) |
|---|---|---|---|---|---|---|
| 5 | UniSeq2Seq LSTM Multi | 16.924±2.614 | 26.647±4.169 | 0.130±0.263 | 108.794±18.020 | 75.919 |
| 5 | UniSeq2Seq RF Multi | 6.225±1.505 | 12.115±4.167 | 0.813±0.107 | 33.694±10.553 | 177.872 |
| 5 | **XGB Seq2Seq Multi** | **5.507±1.798** | **11.729±5.647** | **0.811±0.139** | **30.258±4.051** | **2.018** |
| 5 | **UniSeq2Seq XGB Multi** | **5.180±1.500** | **11.523±5.137** | **0.823±0.126** | **26.981±3.506** | **2.758** |
| 5 | Seq2Seq RF Multi | 6.170±1.255 | 11.870±4.165 | 0.821±0.102 | 35.079±7.288 | 39.199 |
| 5 | UniAR XGB Multi | 10.239±0.520 | 17.494±1.719 | 0.633±0.053 | 88.366±21.530 | 0.971 |
| 5 | XGB Autoreg Multi | 10.239±0.520 | 17.494±1.719 | 0.633±0.053 | 88.366±21.530 | 0.303 |
| 5 | UniSeq2Seq CNN Multi | 14.288±1.667 | 22.753±2.507 | 0.382±0.075 | 108.167±21.391 | 9.146 |
| 12 | UniSeq2Seq LSTM Multi | 18.573±2.315 | 28.694±3.333 | −0.107±0.185 | 161.832±47.877 | 42.448 |
| 12 | UniSeq2Seq RF Multi | 7.973±1.703 | 14.166±2.866 | 0.721±0.107 | 62.910±16.824 | 426.631 |
| 12 | **XGB Seq2Seq Multi** | **7.009±1.724** | **13.941±3.319** | **0.727±0.122** | **58.978±21.005** | **4.241** |
| 12 | **UniSeq2Seq XGB Multi** | **6.764±1.779** | **13.033±3.380** | **0.757±0.125** | **53.274±13.491** | **5.636** |
| 12 | Seq2Seq RF Multi | 7.988±1.739 | 13.844±2.921 | 0.733±0.107 | 69.042±16.787 | 42.647 |
| 12 | UniAR XGB Multi | 16.504±2.945 | 27.276±5.345 | −0.100±0.613 | 192.316±60.373 | 0.304 |
| 12 | XGB Autoreg Multi | 16.504±2.945 | 27.276±5.345 | −0.100±0.613 | 192.316±60.373 | 0.341 |
| 12 | UniSeq2Seq CNN Multi | 16.045±0.714 | 24.017±2.015 | 0.225±0.085 | 190.531±15.425 | 8.041 |
| 24 | UniSeq2Seq LSTM Multi | 20.198±5.517 | 31.310±8.929 | 0.016±0.324 | 133.979±70.421 | 40.221 |
| 24 | UniSeq2Seq RF Multi | 8.206±3.742 | 15.297±8.296 | 0.767±0.155 | 64.219±37.559 | 852.043 |
| 24 | **XGB Seq2Seq Multi** | **6.697±3.009** | **14.092±8.973** | **0.794±0.167** | **59.424±32.438** | **9.965** |
| 24 | **UniSeq2Seq XGB Multi** | **6.537±3.142** | **13.352±8.401** | **0.814±0.144** | **57.559±33.437** | **10.136** |
| 24 | Seq2Seq RF Multi | 8.354±3.879 | 15.235±8.211 | 0.771±0.153 | 65.382±39.536 | 48.363 |
| 24 | UniAR XGB Multi | 21.067±1.794 | 31.027±4.605 | −0.031±0.359 | 208.120±68.348 | 0.321 |
| 24 | XGB Autoreg Multi | 21.067±1.794 | 31.027±4.605 | −0.031±0.359 | 208.120±68.348 | 0.288 |
| 24 | UniSeq2Seq CNN Multi | 17.005±3.833 | 25.691±5.398 | 0.338±0.132 | 128.435±50.594 | 7.106 |
| 48 | UniSeq2Seq LSTM Multi | 15.258±3.738 | 25.663±6.840 | 0.074±0.206 | 130.448±62.681 | 55.195 |
| 48 | UniSeq2Seq RF Multi | 7.723±4.264 | 13.176±7.342 | 0.716±0.253 | 63.447±38.929 | 1810.866 |
| 48 | **XGB Seq2Seq Multi** | **5.802±3.942** | **10.363±8.318** | **0.805±0.211** | **51.446±35.028** | **18.562** |
| 48 | **UniSeq2Seq XGB Multi** | **5.692±4.018** | **10.028±8.336** | **0.807±0.222** | **47.987±31.205** | **23.620** |
| 48 | Seq2Seq RF Multi | 6.760±2.675 | 12.211±6.150 | 0.774±0.163 | 69.330±39.269 | 60.319 |
| 48 | UniAR XGB Multi | 20.997±5.881 | 30.612±6.462 | −0.407±0.548 | 213.705±111.544 | 0.340 |
| 48 | XGB Autoreg Multi | 20.997±5.881 | 30.612±6.462 | −0.407±0.548 | 213.705±111.544 | 0.429 |
| 48 | UniSeq2Seq CNN Multi | 15.861±2.080 | 24.357±4.214 | 0.150±0.131 | 176.000±86.381 | 7.129 |
| 72 | UniSeq2Seq LSTM Multi | 22.381±5.667 | 32.915±9.666 | 0.002±0.259 | 183.615±95.924 | 75.112 |
| 72 | UniSeq2Seq RF Multi | 11.939±11.916 | 15.269±13.068 | 0.756±0.319 | 162.819±226.720 | 2772.810 |
| 72 | **XGB Seq2Seq Multi** | **8.528±10.796** | **11.030±13.442** | **0.829±0.289** | **107.299±165.520** | **21.860** |
| 72 | **UniSeq2Seq XGB Multi** | **9.177±12.750** | **11.472±15.414** | **0.798±0.365** | **139.512±236.929** | **26.243** |
| 72 | Seq2Seq RF Multi | 10.538±9.331 | 14.468±11.837 | 0.781±0.268 | 123.183±147.363 | 68.721 |
| 72 | UniAR XGB Multi | 30.962±14.863 | 39.011±13.668 | −0.663±1.105 | 450.797±438.736 | 0.211 |
| 72 | XGB Autoreg Multi | 30.962±14.863 | 39.011±13.668 | −0.663±1.105 | 450.797±438.736 | 0.264 |
| 72 | UniSeq2Seq CNN Multi | 24.921±10.351 | 32.021±10.068 | 0.077±0.192 | 323.508±331.374 | 7.133 |

Multi, achieved an MAE of 5.180±1.500 µg/m$^3$ and an R$^2$ of 0.823±0.126, offering a slight 2% improvement in MAE over its univariate counterpart, though with a modest reduction in R$^2$. This indicates that for brief gaps, incorporating additional features yields only a small gain in predictive accuracy without enhancing explanatory power significantly.

When gap lengths extended to medium durations of 12 and 24 hours, the differences between univariate and multivariate models grew more apparent. For 12-hour gaps, univariate models averaged an MAE of about 10.10 μg/m³ and an R² of 0.67, with UniSeq2Seq XGB recording an

MAE of 7.279±2.199 μg/m³ and an R² of 0.725±0.166. Multivariate models, meanwhile, posted an average MAE of approximately 9.71 μg/m³ and an R² of 0.66, but the leading multivariate model, UniSeq2Seq XGB Multi, achieved an MAE of 6.764±1.779 μg/m³ and an R² of 0.757±0.125. This reflects a 7% reduction in MAE and a 4% improvement in R² compared to the best univariate model. At 24 hours, univariate models showed an average MAE of around 10.36 μg/m³ and an R² of 0.69, with UniSeq2Seq XGB achieving an MAE of 6.969±3.111 μg/m³ and an R² of 0.789±0.159. Multivariate models averaged an MAE of about 9.62 μg/m³ and an R² of 0.71, with UniSeq2Seq XGB Multi recording an MAE of 6.537±3.142 μg/m³ and an R² of 0.814±0.144, marking a 6% decrease in MAE and a 3% increase in R². These outcomes emphasize the increasing effectiveness of multivariate models in reducing errors as gap durations rise.

For longer gaps of 48 and 72 hours, the advantages of multivariate models became even more striking. At 48 hours, univariate models averaged an MAE of approximately 10.73 μg/m³ and an R² of 0.67, with XGB Seq2Seq achieving an MAE of 6.971±3.126 μg/m³ and an R² of 0.784±0.167. Multivariate models, by comparison, averaged an MAE of about 8.85 μg/m³ and an R² of 0.70, with UniSeq2Seq XGB Multi posting an MAE of 5.692±4.018 μg/m³ and an R² of 0.807±0.222, resulting in an 18% reduction in MAE and a 3% rise in R². For 72-hour gaps, univariate models exhibited an average MAE of around 14.43 μg/m³ and an R² of 0.62, with UniSeq2Seq XGB recording an MAE of 10.144±10.943 μg/m³ and an R² of 0.766±0.328. Multivariate models averaged an MAE of approximately 11.74 μg/m³ and an R² of 0.70, with XGB Seq2Seq Multi achieving an MAE of 8.528±10.796 μg/m³ and an R² of 0.829±0.289, reflecting a 16% improvement in MAE and an 8% increase in R². These results clearly demonstrate the enhanced capability of multivariate models to maintain accuracy over extended missing data periods.

Accurate prediction of PM2.5 levels is essential for environmental monitoring and public health management. However, data gaps in PM2.5 measurements present significant challenges to achieving reliable predictions. To address this issue, both univariate and multivariate models have been developed and applied to fill these gaps. This analysis explores the influence of multivariateness on model effectiveness by comparing the performance of univariate and multivariate models across varying gap lengths in PM2.5 prediction.

Performance was assessed using the Mean Absolute Error (MAE) over gap lengths ranging from 5 to 72 hours. Visualizations of the results reveal distinct trends (Fig 14). For the UniSeq2Seq XGB model, the MAE starts at 5.286 μg/m³ at 5 hours, increasing to 10.144 μg/m³ at 72 hours, indicating a notable decline in accuracy with longer gaps. Its multivariate counterpart, UniSeq2Seq XGB Multi, begins at a slightly lower 5.180 μg/m³ at 5 hours and reaches 9.177 μg/m³ at 72 hours, demonstrating improved stability. Similarly, the XGB Seq2Seq model exhibits an MAE of 5.555 μg/m³ at 5 hours, rising to 10.624 μg/m³ at 72 hours, while the XGB Seq2Seq Multi model maintains a reduced MAE of 5.507 μg/m³ at 5 hours and 8.528 μg/m³ at 72 hours. These trends suggest that multivariateness enhances model resilience, with the most significant reductions in MAE observed at extended gap lengths.

To further evaluate the predictive accuracy of these models, aggregated scatter plots are presented in Fig 15, consolidating data across all gap lengths. These plots depict the relationship between predicted and true PM2.5 values, with a dashed red line representing the ideal 1:1 correlation and a solid blue line indicating the linear regression fit. The subplots are organized as follows: (a) UniSeq2Seq XGB, (b) UniSeq2Seq XGB Multi, (c) XGB Seq2Seq, and (d) XGB Seq2Seq Multi. The proximity of the regression line to the 1:1 line and the dispersion of points around it provide insight into model performance. For UniSeq2Seq XGB (Fig 15a), the regression line deviates notably from the 1:1 line, particularly at higher values, suggesting overprediction or underprediction as true values increase. In contrast, UniSeq2Seq XGB Multi (Fig 15b) shows a regression line closer to the 1:1 line, indicating improved alignment between predicted and true values, consistent with the MAE reduction observed. Similarly, XGB Seq2Seq (Fig 15c) exhibits a wider scatter and greater deviation, while XGB Seq2Seq Multi (Fig 15d) demonstrates a tighter clustering around the regression line, reinforcing the benefit of

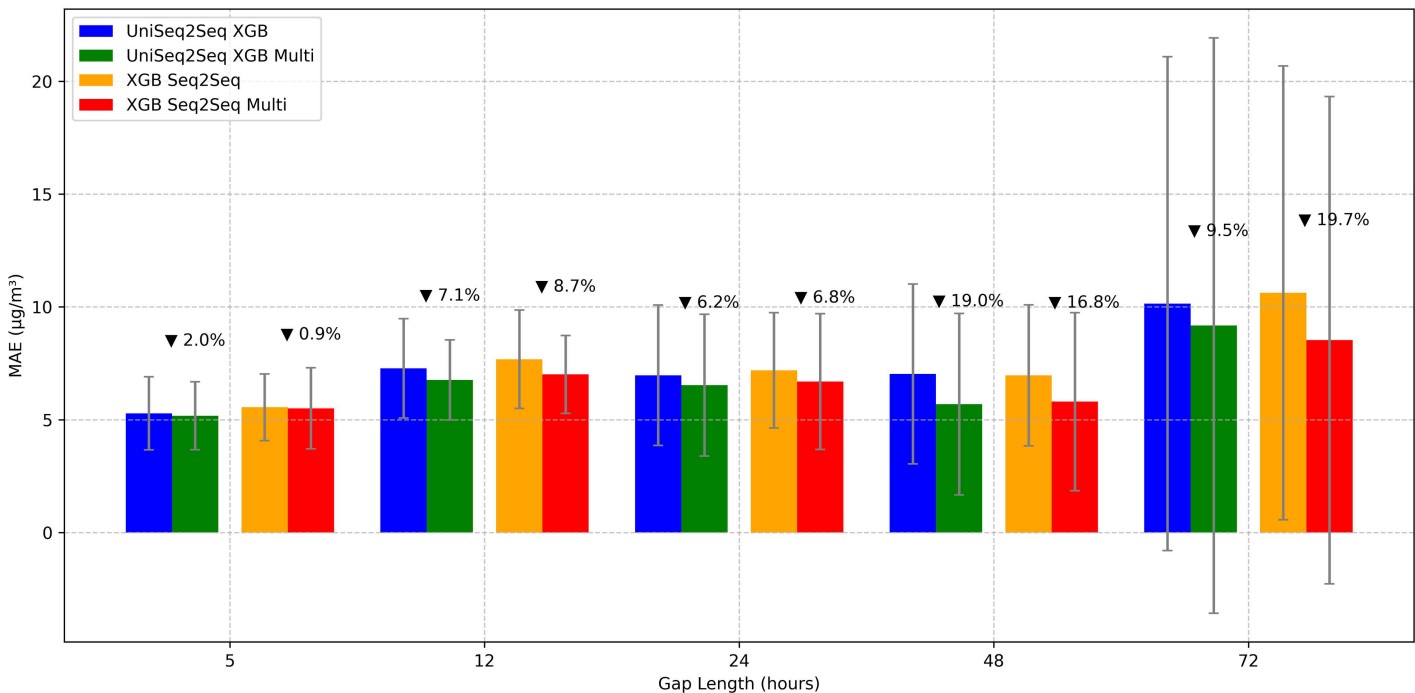

**Fig 14. Mean absolute error comparison across gap lengths for univariate and multivariate PM2.5 prediction models.** The percentage indicators denote the degree of MAE reduction when transitioning from univariate to multivariate models.

multivariateness. These visual patterns corroborate the quantitative findings, highlighting the enhanced predictive capability of multivariate models across diverse gap scenarios.

Time series visualization (Fig 16) provides insight into each model's ability to capture complex patterns during extended gaps. The graphs represent the same 48-hour gap period filled by different models, demonstrating how multivariate models (Fig 16b, 16d) produce predictions that more closely follow the actual measurements compared to their univariate counterparts (Fig 16a, 16c). Both model types accurately capture major trends and peaks, but multivariate models show improved alignment during rapid fluctuations and transitional periods. This visual comparison complements the quantitative metrics by illustrating the practical impact of incorporating additional environmental variables on prediction accuracy, particularly for challenging segments with high variability. The enhanced performance of multivariate models becomes especially evident when examining the steep transitions around time steps 25–35, where the integration of correlated meteorological parameters allows for more precise reconstruction of complex pollution dynamics.

Statistical significance of these improvements was confirmed through paired t-tests comparing MAE values between corresponding univariate and multivariate models across all gap lengths, yielding p-values below 0.01. This result affirms that the observed reductions in MAE are robust and not due to random variation. The incorporation of additional environmental variables in multivariate models likely contributes to their superior performance, particularly as data sparsity increases with longer gaps.

In conclusion, multivariate models outperform their univariate counterparts in PM2.5 prediction, with the effect most evident in corresponding model pairs such as UniSeq2Seq XGB versus UniSeq2Seq XGB Multi and XGB Seq2Seq versus XGB Seq2Seq Multi. These findings advocate for the adoption of multivariate approaches in environmental forecasting, especially under conditions of extended data gaps, to enhance predictive reliability.

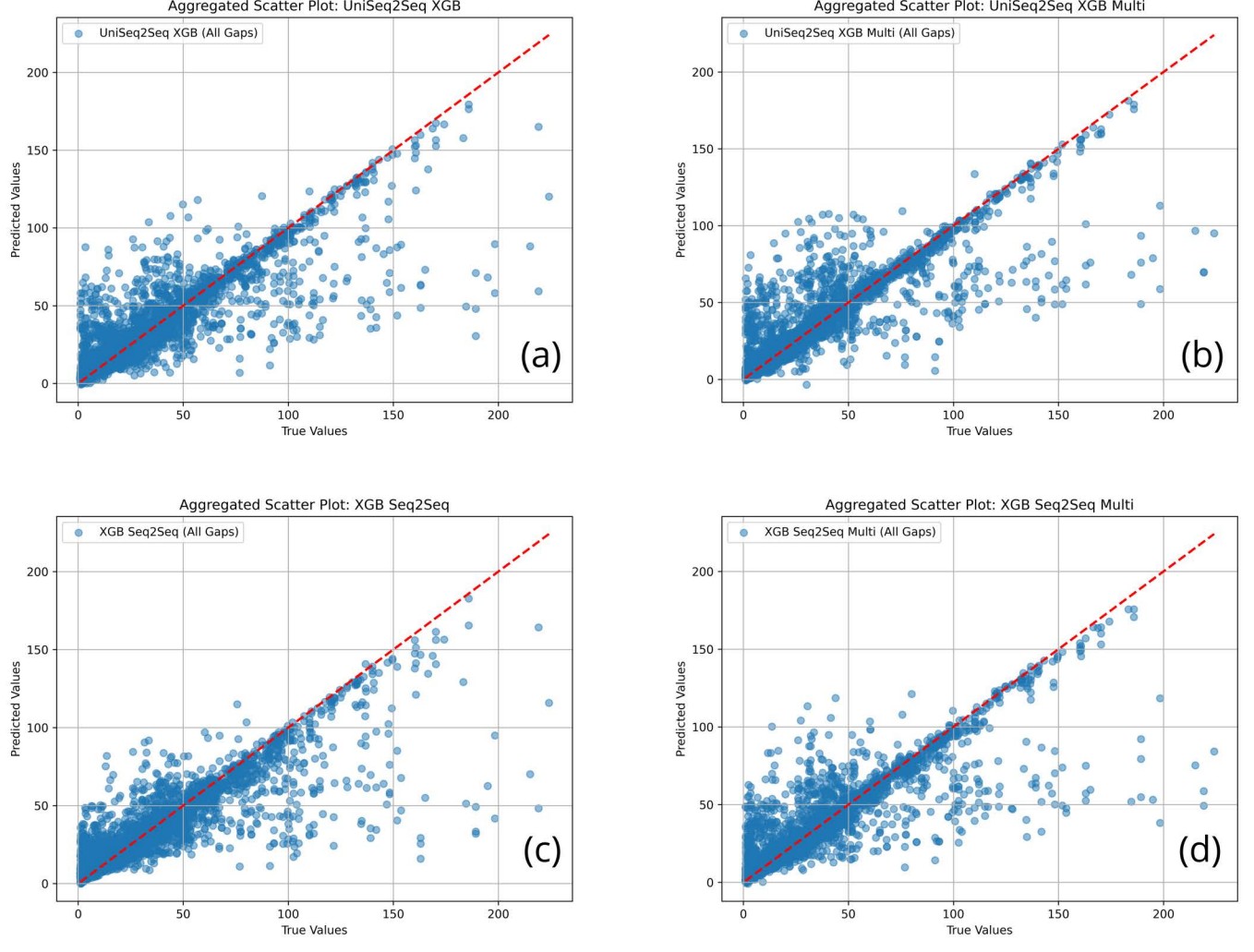

**Fig 15. Aggregated scatter plots of predicted vs. true PM2.5 values across all gap lengths: (a) UniSeq2Seq XGB, (b) UniSeq2Seq XGB Multi, (c) XGB Seq2Seq, (d) XGB Seq2Seq Multi.** The dashed red line represents the ideal 1:1 correlation, while the solid blue line indicates the linear regression fit. The proximity of the regression line to the 1:1 line and the dispersion of points reflect the predictive accuracy, with multivariate models showing improved alignment.

Overall, the analysis reveals that while multivariate models provide only slight benefits for short gaps, their superiority becomes increasingly evident as gap lengths grow. For medium and long gaps, these models consistently outperform univariate ones, reducing MAE by up to 18% and boosting $R^2$ by as much as 8%. This underscores the importance of integrating additional environmental features to improve the precision and reliability of PM2.5 predictions, particularly in cases of prolonged data gaps.

**Dynamic model development for variable gap lengths.** While our extensive array of methods demonstrated promising results in synthetic gap-filling experiments, we identified significant challenges when applying these models to real-world data scenarios. Standard fixed-context models faced consistent limitations when confronted with the complexity and variability of actual data gaps.

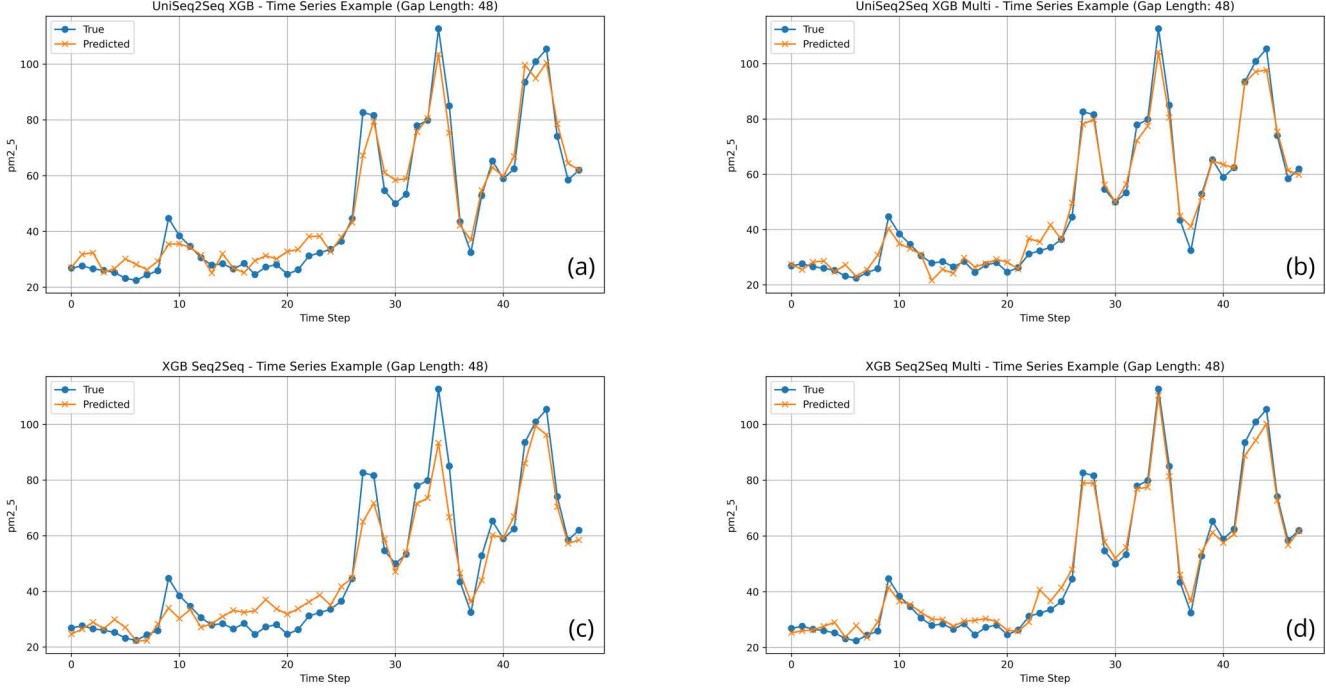

**Fig 16. Comparison of true vs. predicted PM2.5 values using univariate and multivariate models for a 48-hour gap: (a) UniSeq2Seq XGB, (b) UniSeq2Seq XGB Multi, (c) XGB Seq2Seq, (d) XGB Seq2Seq Multi.**

First, these models showed highly specialized performance profiles, excelling at specific gap lengths but degrading rapidly for others. Models trained on short gaps struggled with extended interruptions, while those optimized for lengthy gaps often overprocessed shorter ones, introducing unnecessary complexity and potential overfitting.

Second, standard approaches required separate models for each gap length, creating substantial maintenance challenges in operational settings where gaps of unpredictable duration regularly occur. In our real-world dataset, we encountered 25 different gap lengths ranging from extremely short (1 hour) to exceptionally long (191 hours). Implementing static models would have necessitated creating and maintaining an ensemble of 25 different specialized models, each specialized for a particular gap duration. This would significantly increase computational costs and complicate the model selection logic.

Third, our synthetic testing environment assumed ideal conditions with sufficient context available around each gap. However, real-world data presented critical context limitations:

– Gaps occurring at the beginning or end of time series lacked the full pre-context or post-context windows (32 hours) used during training, making static models inapplicable.

– Multiple gaps positioned near each other left insufficient data between them to form complete context windows. For example, if two gaps were separated by only 10 hours, static models expecting 32-hour contexts could not be properly applied.

– Real gaps appeared in complex patterns that did not match the controlled conditions used during model training, limiting the transferability of static approaches.

Our comprehensive testing of various modeling approaches was instrumental in identifying the most effective techniques for this specific environmental data imputation task. The XGBoost-based models with sequence-to-sequence architecture consistently outperformed other approaches across different gap lengths, demonstrating superior accuracy and computational efficiency. Similarly, our correlation analysis and feature selection process identified the most relevant meteorological variables (wind speed, wind direction, temperature, humidity) and temporal features (hour of day, season) that notably improved prediction accuracy when incorporated into multivariate models. These findings provided a strong empirical foundation for designing our dynamic models.

Analysis of performance patterns revealed that optimal context window size should ideally be proportional to gap length–shorter gaps benefit from focused local context, while longer gaps require broader temporal information. However, fixed architectures cannot adapt their receptive field dynamically to match the specific gap being addressed.

By integrating the XGBoost sequence-to-sequence architecture with our dynamic approach, and incorporating the optimal set of meteorological and temporal variables identified in our earlier analysis, we created both univariate (DynamicSeq2SeqXGB) and multivariate (DynamicMultiSeq2SeqXGB) dynamic models. This dynamic modeling approach offered substantial practical advantages, including simplified deployment (one model instead of 25), robust handling of gaps with limited context, and optimized computational resource utilization through appropriately sized context windows. The resulting models demonstrated remarkable adaptability across the diverse gap patterns encountered in real-world air quality monitoring data.

The dynamic models were evaluated across five gap lengths (5, 12, 24, 48, and 72 hours) using synthetic gap insertion with a 5% missing fraction and five independent runs per configuration. Table 7 presents the performance metrics for both univariate and multivariate dynamic models.

The dynamic models demonstrated consistent performance across all gap lengths, with the multivariate implementation consistently outperforming its univariate counterpart (Fig 17). For short gaps (5 hours), the dynamic multivariate model achieved a 7.5% reduction in MAE compared to the univariate model ($7.665 \pm 1.521$ µg/m³ vs. $8.287 \pm 1.549$ µg/m³). This improvement increased to 9.1% for 12-hour gaps and 10.5% for 24-hour gaps, highlighting the growing advantage of incorporating meteorological variables as gap length increases.

For medium-length gaps (48 hours), the performance gap widened substantially, with the multivariate model achieving an MAE of $8.025 \pm 3.708$ µg/m³ compared to $9.676 \pm 4.557$ µg/m³ for the univariate model, representing a 17.1% improvement. This trend continued with long gaps (72 hours), where the multivariate model showed a 7.5% reduction in MAE ($10.772 \pm 6.214$ µg/m³ vs. $11.651 \pm 4.611$ µg/m³).

The $R^2$ values demonstrate that both dynamic models maintain strong explanatory power for short and medium gaps, with values above 0.75 for gaps up to 24 hours. However, performance degrades for longer gaps, particularly for the

**Table 7. Performance comparison of dynamic models across different gap lengths (mean±standard deviation).**

| Gap Length | Model Type | MAE (µg/m³) | RMSE (µg/m³) | R2 | MAPE (%) |
|---|---|---|---|---|---|
| 5 | Dynamic Uni | 8.287 ± 1.549 | 13.876 ± 3.471 | 0.756 ± 0.073 | 70.019 ± 21.738 |
| 5 | Dynamic Multi | 7.665 ± 1.521 | 13.655 ± 3.927 | 0.772 ± 0.049 | 60.102 ± 15.567 |
| 12 | Dynamic Uni | 7.978 ± 2.015 | 13.728 ± 6.216 | 0.775 ± 0.118 | 72.916 ± 21.188 |
| 12 | Dynamic Multi | 7.253 ± 1.800 | 13.680 ± 4.967 | 0.775 ± 0.088 | 71.841 ± 31.483 |
| 24 | Dynamic Uni | 8.790 ± 1.844 | 13.440 ± 3.567 | 0.750 ± 0.140 | 90.589 ± 29.246 |
| 24 | Dynamic Multi | 7.868 ± 2.677 | 13.044 ± 5.100 | 0.771 ± 0.138 | 79.035 ± 39.960 |
| 48 | Dynamic Uni | 9.676 ± 4.557 | 14.908 ± 7.978 | 0.522 ± 0.244 | 118.378 ± 47.211 |
| 48 | Dynamic Multi | 8.025 ± 3.708 | 12.918 ± 6.871 | 0.635 ± 0.233 | 95.259 ± 43.927 |
| 72 | Dynamic Uni | 11.651 ± 4.611 | 16.933 ± 6.803 | 0.241 ± 0.512 | 123.258 ± 90.690 |
| 72 | Dynamic Multi | 10.772 ± 6.214 | 15.535 ± 9.210 | 0.221 ± 0.847 | 114.091 ± 114.574 |

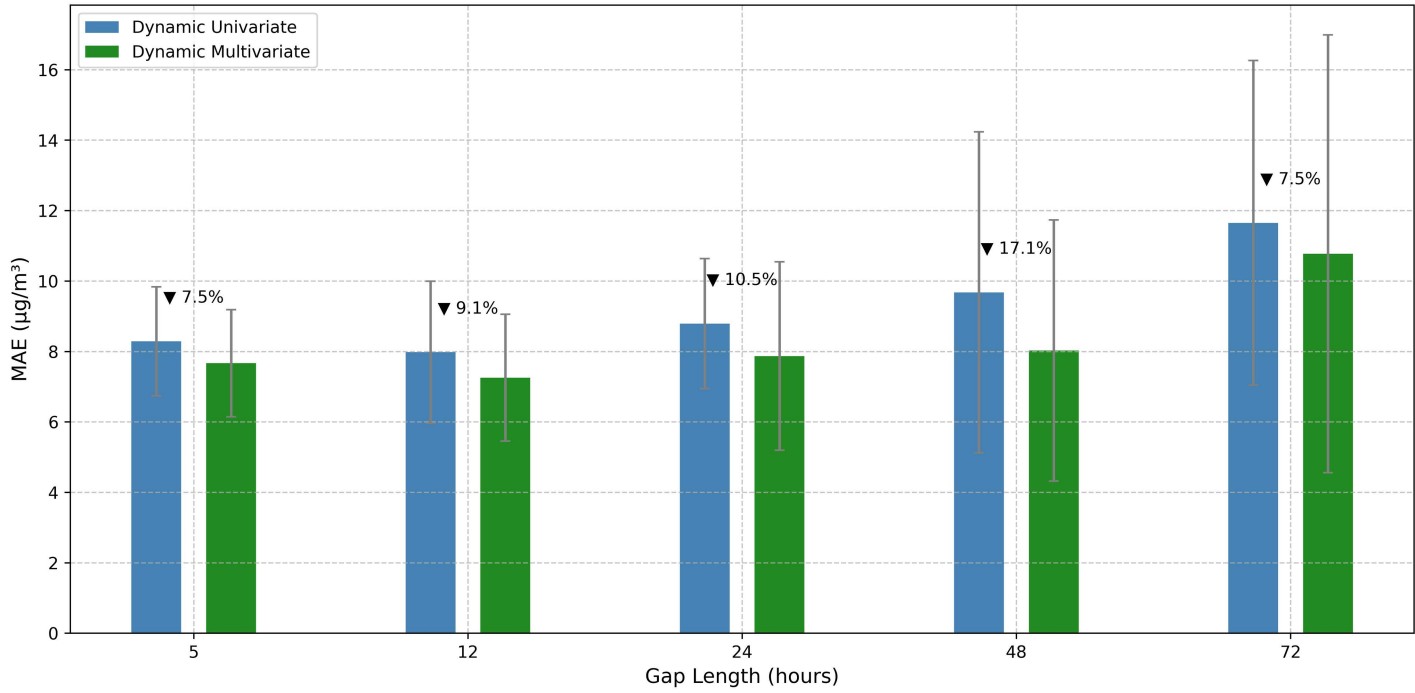

**Fig 17. Comparison of MAE (μg/m³) for dynamic univariate and multivariate models across different gap lengths.** The percentage values represent the reduction in MAE achieved by the multivariate model compared to the univariate model.

univariate model, which shows an $R^2$ of just 0.241±0.512 for 72-hour gaps. The multivariate model maintains greater prediction stability across all gap lengths, with less performance degradation as gap duration increases.

The MAPE values reveal that both models struggle with percentage errors for longer gaps, but the multivariate model consistently achieves lower values, particularly for 48-hour gaps where it demonstrates a 19.5% improvement (95.259±43.927% vs. 118.378±47.211%). The high standard deviations in MAPE for 72-hour gaps (90.690% for univariate and 114.574% for multivariate) indicate considerable variability in percentage errors for the longest gap duration.

When compared to the best-performing static models (UniSeq2Seq XGB and XGB Seq2Seq Multi), the dynamic models demonstrated competitive performance while offering significant practical advantages. For 5-hour gaps, the dynamic multivariate model achieved an MAE of 7.665±1.521 μg/m³, which is higher than the best static model's 5.180±1.500 μg/m³, but this difference diminished as gap length increased.

For 48-hour gaps, the dynamic multivariate model achieved an MAE of 8.025±3.708 μg/m³, which is 41% higher than the best static model's 5.692±4.018 μg/m³. However, the dynamic model demonstrated more consistent performance across different gap lengths, with less performance degradation for longer gaps. This consistency is particularly valuable in operational settings where gap lengths cannot be predicted in advance.

The primary advantage of the dynamic models lies in their ability to handle gaps of any length with a single model. While our test framework examined five specific gap lengths, the dynamic nature of these models allows them to process gaps of any duration, including those significantly longer than the training examples. This flexibility was demonstrated during the application to real-world data, where the models successfully filled gaps ranging from 1 to 191 hours.

The true test of any gap-filling method is its performance on real-world data with naturally occurring gaps. Both dynamic models were applied to fill the 164 gaps detected in our PM2.5 time series, ranging from 1 to 191 hours in duration. The

distribution of these gaps revealed 25 different gap lengths, with the most common being 3-hour gaps (35 occurrences) and the longest being a 191-hour continuous missing period.

While ground truth values were unavailable for these real gaps (preventing direct MAE calculation), visual inspection of the filled time series showed smooth transitions at gap boundaries and preservation of temporal patterns consistent with surrounding measurements. The multivariate model demonstrated better handling of complex transition regions and appeared to capture seasonal and diurnal patterns more effectively than its univariate counterpart.

A specific advantage of the dynamic models became apparent when processing gaps with limited context. For example, gaps near the beginning or end of the time series, which lacked the full 32-hour context window used by static models, were handled naturally by the dynamic approach through its adaptive context sizing mechanism. Similarly, closely spaced gaps with insufficient data between them were processed successfully without the need for specialized handling.

The most challenging case was the 191-hour gap, which far exceeded the maximum length used during training (72 hours). The dynamic chunking approach automatically divided this extended gap into manageable segments, maintaining prediction quality throughout the gap period. This demonstrated the dynamic models' capability to generalize beyond their training examples, a critical feature for operational deployment in real-world monitoring systems.

Computational efficiency is another important consideration for operational deployment. The dynamic multivariate model required approximately 73 seconds for training, while inference time was proportional to gap length, with shorter gaps processed more quickly (18.93 seconds for all 5-hour gaps, decreasing to 0.76 seconds for all 72-hour gaps). These processing times are suitable for operational use in environmental monitoring systems, where gap-filling typically occurs during post-processing rather than in real-time.

In conclusion, the dynamic models demonstrated remarkable flexibility and consistency when applied to both synthetic and real-world gaps. While they did not achieve the absolute lowest MAE scores compared to specialized static models trained for specific gap lengths, their ability to handle gaps of arbitrary length and position with a single model represents a significant practical advancement for environmental time series gap-filling. The multivariate dynamic model, in particular, offers an optimal balance of accuracy, flexibility, and computational efficiency for operational PM2.5 monitoring applications. Thus, the multivariate dynamic model DynamicMultiSeq2SeqXGB was utilized for obtaining the full PM2.5 monitoring dataset with filled absent data for following air quality assessment. After filling, a minimal part of PM2.5 data was negative due to synthetic imputation in areas of low values. Overall, 7 out of 5,808 readings (0.12%) were negative. Since negative PM2.5 concentrations are physically meaningless, these values were set to zero.

**Key findings and practical recommendations.** The comprehensive evaluation of 46 gap-filling methods reveals clear performance hierarchies and practical guidance for PM2.5 time series reconstruction. Tree-based models with bidirectional sequence-to-sequence architectures consistently delivered superior performance, with XGB Seq2Seq achieving the best accuracy across all gap lengths and Dynamic Multivariate XGB offering optimal operational flexibility for real-world deployment scenarios.

The selection of appropriate methods depends critically on gap characteristics and operational requirements. For short gaps of 5–12 hours, univariate models provide sufficient accuracy, making tree-based seq2seq approaches the recommended choice due to their computational efficiency and robust performance. Medium-length gaps of 24 hours mark the transition point where multivariate models begin demonstrating meaningful advantages, with improvements of 6–7% over univariate counterparts. For extended gaps of 48–72 hours, multivariate models become essential, providing 16–18% superior performance compared to univariate approaches.

From an operational perspective, monitoring networks with predictable gap patterns should implement specialized XGB Seq2Seq models optimized for specific gap lengths, while systems encountering variable gap durations benefit from the Dynamic Multivariate model's ability to handle any gap scenario with a single implementation. Simple statistical methods such as mean or median imputation should be avoided due to their substantially inferior performance, while autoregressive approaches suffer from error accumulation that limits their effectiveness for longer gaps.

## Air quality assessment and characterization of PM2.5 concentrations

**Temporal PM2.5 distribution and basic statistical analysis.** Analysis of the gap-filled PM2.5 dataset from May 2024 to January 2025 revealed significant temporal variability in concentrations and pronounced positive skewness in distribution. The mean PM2.5 concentration was 34.29 µg/m³ (SD = 34.46), substantially higher than the median value of 22.09 µg/m³, indicating an asymmetrical distribution with a long tail of high values (skewness coefficient = 1.71), as shown in Table 8. This observation is confirmed by visual analysis of the distribution histogram (Fig 18), demonstrating the characteristic log-normal distribution typical for atmospheric pollutants. The Q-Q plot further shows substantial deviation from normal distribution, particularly in the high-value region.

The range of observed concentrations was quite broad, from a minimum value of 0.00 µg/m³ (likely associated with measurement errors near zero) to a maximum of 251.38 µg/m³, demonstrating episodes of extremely high pollution. The 95th percentile was 106.46 µg/m³, and the 99th percentile was 149.58 µg/m³, indicating that the highest concentrations were relatively rare events. The interquartile range (9.28–48.38 µg/m³) reflects typical daily variations in PM2.5 concentrations in the urban environment.

Analysis of seasonal PM2.5 dynamics revealed pronounced differences between seasons (Fig 19). The highest PM2.5 concentrations were observed in winter (mean = 45.87 µg/m³, SD = 34.83), followed by summer (mean = 34.59 µg/m³, SD = 37.31). Autumn showed moderate levels (mean = 29.71 µg/m³, SD = 30.07), while spring had the lowest average values (mean = 13.24 µg/m³, SD = 20.41) (Table 9).

The analysis reveals a complex seasonal pattern. Winter emerges as the season with the highest PM2.5 concentrations. This could be attributed to the inclusion of more winter months in the dataset, providing a more comprehensive view of seasonal variations.

Monthly analysis (Fig 20) demonstrates a more detailed picture of PM2.5 dynamics. The highest average concentrations were noted in December (46.73 µg/m³), January (44.46 µg/m³), and August (43.73 µg/m³). The lowest concentrations were recorded in May (13.24 µg/m³) and September (17.00 µg/m³). Although data is missing for February, March, and

**Table 8. Basic statistical parameters of PM2.5 concentrations in Pavlodar, May 2024–January 2025.**

| Statistic | Value (µg/m³) |
|---|---|
| Count | 5808 |
| Mean | 34.29 |
| Standard Deviation | 34.46 |
| Minimum | 0.00 |
| 25th Percentile | 9.28 |
| Median (50th) | 22.09 |
| 75th Percentile | 48.38 |
| Maximum | 251.38 |
| Variance | 1187.37 |
| Skewness | 1.71 |
| Kurtosis | 3.43 |
| 1st Percentile | 1.23 |
| 5th Percentile | 2.35 |
| 10th Percentile | 4.32 |
| 90th Percentile | 82.82 |
| 95th Percentile | 106.46 |
| 99th Percentile | 149.58 |

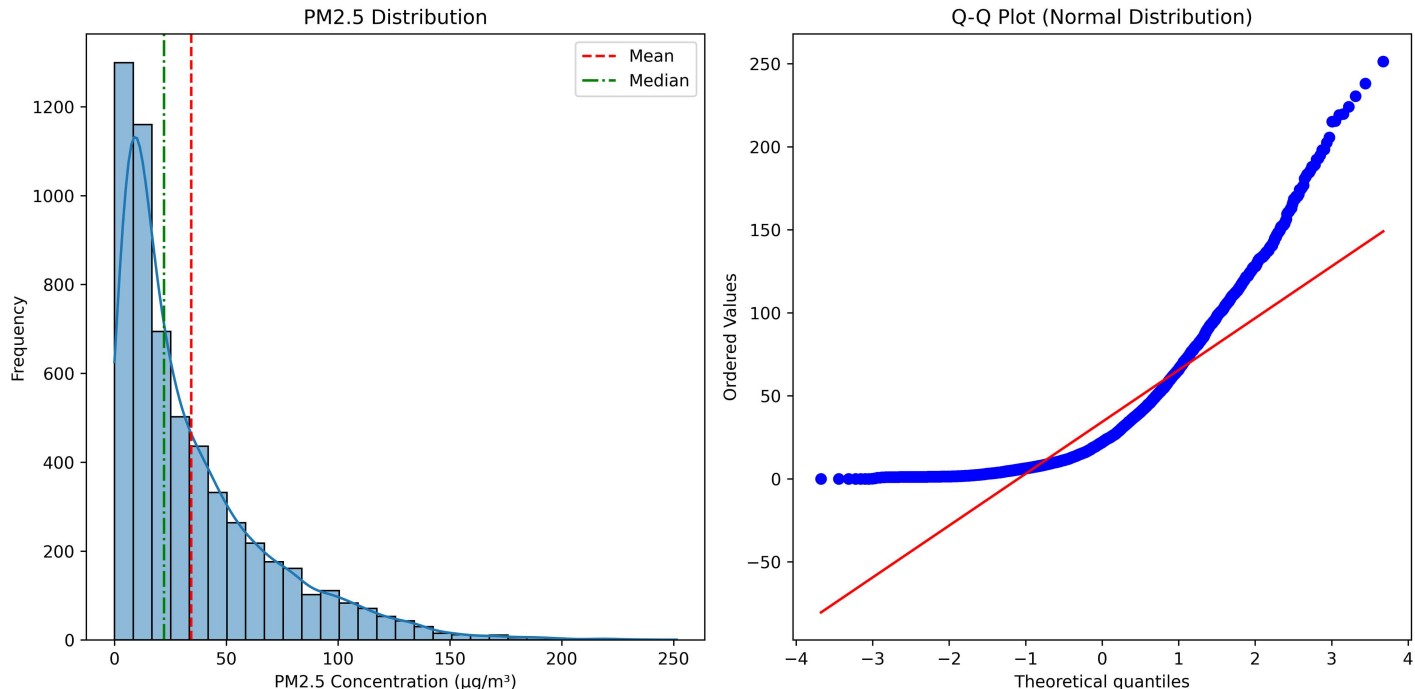

**Fig 18. Distribution of PM2.5 concentrations in Pavlodar (May 2024–January 2025): (left) histogram with probability density function showing mean and median values; (right) Q-Q plot demonstrating deviation from normal distribution.**

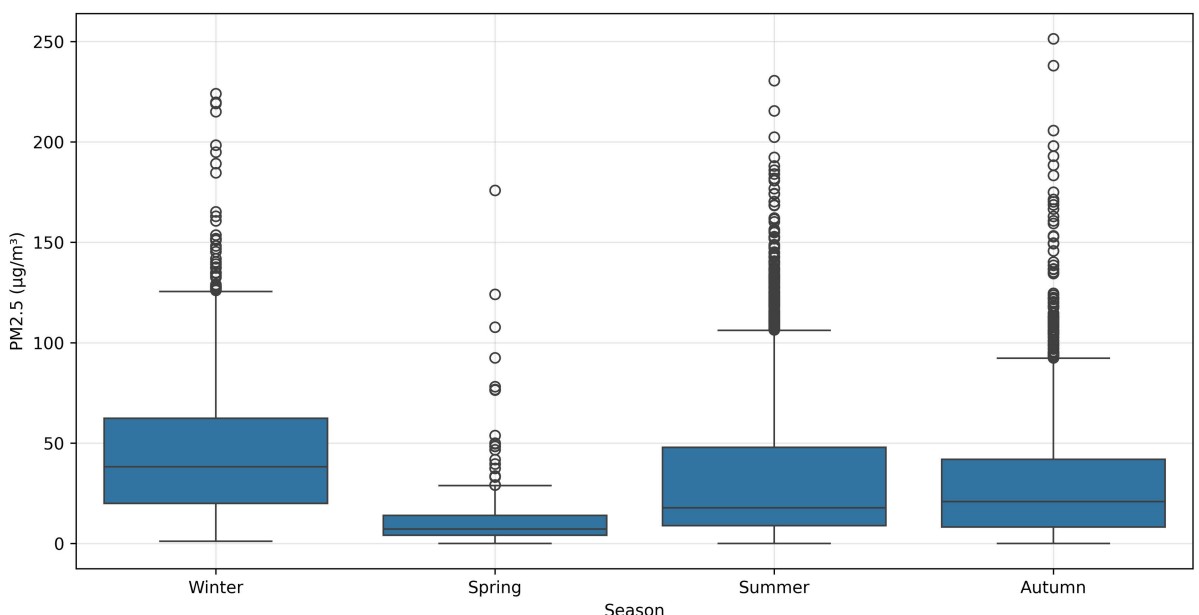

**Fig 19. PM2.5 concentration distributions by season in Pavlodar (May 2024–January 2025).** Boxplots show median values (central lines), inter-quartile ranges (boxes), and outliers (circles).

**Table 9. Seasonal statistics of PM2.5 concentrations in Pavlodar, May 2024–January 2025.**

| Season | Count | Mean (µg/m³) | SD (µg/m³) | Min (µg/m³) | Median (µg/m³) | Max (µg/m³) | 5th Percentile | 95th Percentile |
|--------|-------|--------------|------------|-------------|----------------|-------------|----------------|-----------------|
| Winter | 1200 | 45.87 | 34.83 | 1.05 | 38.23 | 224.08 | 5.47 | 112.83 |
| Summer | 2208 | 34.59 | 37.31 | 0.00 | 17.73 | 230.50 | 3.41 | 117.47 |
| Autumn | 2184 | 29.71 | 30.08 | 0.00 | 20.81 | 251.38 | 1.53 | 87.17 |
| Spring | 216 | 13.24 | 20.41 | 0.00 | 7.18 | 175.90 | 1.46 | 48.41 |

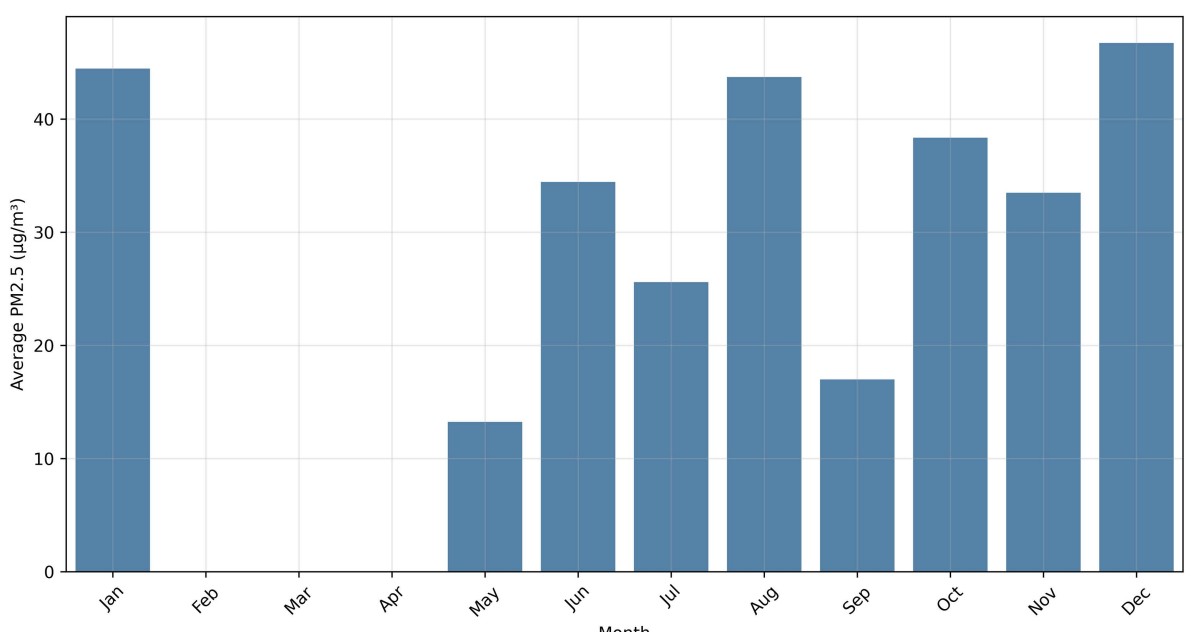

**Fig 20. Monthly average PM2.5 concentrations in Pavlodar (May 2024–January 2025).** Note the absence of data for February, March, and April.

April, there is a clear tendency toward increased concentrations in the winter period (December-January) and decreased concentrations in transitional seasons (May, September) (Table 10).

Notably, the maximum recorded values in the autumn period (251.38 µg/m³ in October) were higher than in other seasons, likely associated with episodic pollution sources or specific meteorological conditions conducive to pollutant accumulation.

These findings underscore the significant variability of PM2.5 concentrations across different seasons, highlighting the importance of comprehensive temporal analysis in understanding urban air quality dynamics.

Time series analysis (Fig 21) revealed significant daily and seasonal variability in PM2.5 concentrations. The hourly graph demonstrates sharp concentration peaks reaching 200–250 µg/m³, alternating with periods of relatively low values (0–20 µg/m³). The graph of daily average values shows a smoother but still pronounced variability with several periods of elevated concentrations, particularly noticeable in September 2024, when daily average values reached approximately 140 µg/m³, and in December 2024–January 2025, when consistently elevated concentrations were observed.

The observed temporal structure of PM2.5 concentrations aligns with typical patterns of atmospheric pollution in urban areas, where concentrations are influenced by both anthropogenic factors (e.g., transport and industrial emissions with

**Table 10. Monthly statistics of PM2.5 concentrations in Pavlodar, May 2024–January 2025.**

| Month | Count | Mean (µg/m³) | SD (µg/m³) | Min (µg/m³) | Median (µg/m³) | Max (µg/m³) |
|---|---|---|---|---|---|---|
| May | 216 | 13.24 | 20.41 | 0.00 | 7.18 | 175.90 |
| June | 720 | 34.45 | 37.01 | 1.08 | 17.64 | 202.43 |
| July | 744 | 25.60 | 27.63 | 1.13 | 15.45 | 188.00 |
| August | 744 | 43.73 | 43.36 | 0.00 | 24.68 | 230.50 |
| September | 720 | 17.00 | 19.79 | 0.00 | 10.65 | 171.35 |
| October | 744 | 38.35 | 34.82 | 0.92 | 27.03 | 251.38 |
| November | 720 | 33.50 | 29.12 | 0.00 | 26.75 | 183.33 |
| December | 744 | 46.73 | 35.42 | 1.32 | 38.45 | 224.08 |
| January | 456 | 44.46 | 33.84 | 1.05 | 36.73 | 189.27 |

distinct daily cycles) and meteorological conditions (inversion layers, wind speed, precipitation) that modulate the dispersion and accumulation of pollutants.

The analysis data emphasize the need to account for significant temporal variability when assessing the impact of PM2.5 on public health and when developing strategies to improve air quality. Special attention should be paid to periods of elevated concentrations in winter months and individual episodes of extremely high values potentially associated with adverse meteorological conditions or specific emission sources.

**WHO guideline exceedances analysis.** The analysis of PM2.5 concentrations in relation to the World Health Organization (WHO) daily threshold of 15 µg/m³ revealed substantial air quality challenges in the study area. Of the 5,808 hours analyzed, 3,559 hours (61.2%) exceeded the WHO guideline threshold, with only 38.7% of the monitored period maintaining acceptable air quality levels. This indicates that for the majority of the monitoring period, local residents were exposed to PM2.5 concentrations potentially associated with adverse health effects.

The severity of exceedances showed a concerning distribution pattern. Moderate exceedances (1–2 times the WHO threshold) occurred during 20.7% of the monitoring period, while severe exceedances (2–4 times the threshold) were observed in 21.7% of hours. Particularly alarming were the high concentrations recorded during 15.6% of hours with 4–8 times the threshold (60–120 µg/m³) and the extreme values exceeding 8 times the threshold (>120 µg/m³) during 3.1% of the monitoring period. The maximum recorded concentration reached 251.4 µg/m³, which is nearly 17 times the WHO daily guideline value.

Significant seasonal and monthly variations were observed in the frequency and intensity of threshold exceedances. The winter months (December and January) demonstrated the highest exceedance rates at 85.6% and 77.4% respectively, while May showed the lowest rate at 21.8%. This pronounced seasonal pattern suggests strong influence of heating-related emissions and unfavorable meteorological conditions during winter months. The autumn months (October and November) also showed considerable exceedance rates of 70.8% and 67.5%, indicating transitional seasonal effects on air quality.

Diurnal patterns of exceedances were evident across all months, with more frequent and severe exceedances typically occurring during evening and early morning hours. This pattern aligns with expected daily variations in local emission sources and atmospheric boundary layer dynamics. Notably, late June exhibited a period of particularly poor air quality, with multiple days showing severe and extreme exceedances across most hours of the day. Similarly, the latter half of August and most of December showed persistent high concentration patterns, suggesting extended pollution episodes possibly linked to specific meteorological conditions or emission events.

These findings highlight the need for targeted air quality management strategies during high-risk periods, particularly during winter months and evening hours when PM2.5 concentrations most frequently exceed health-based guidelines.

Fig 22 presents the hourly distribution of PM2.5 concentrations relative to the WHO daily threshold of 15 µg/m³ across nine months of monitoring. The color scale indicates the severity of threshold exceedances: light blue

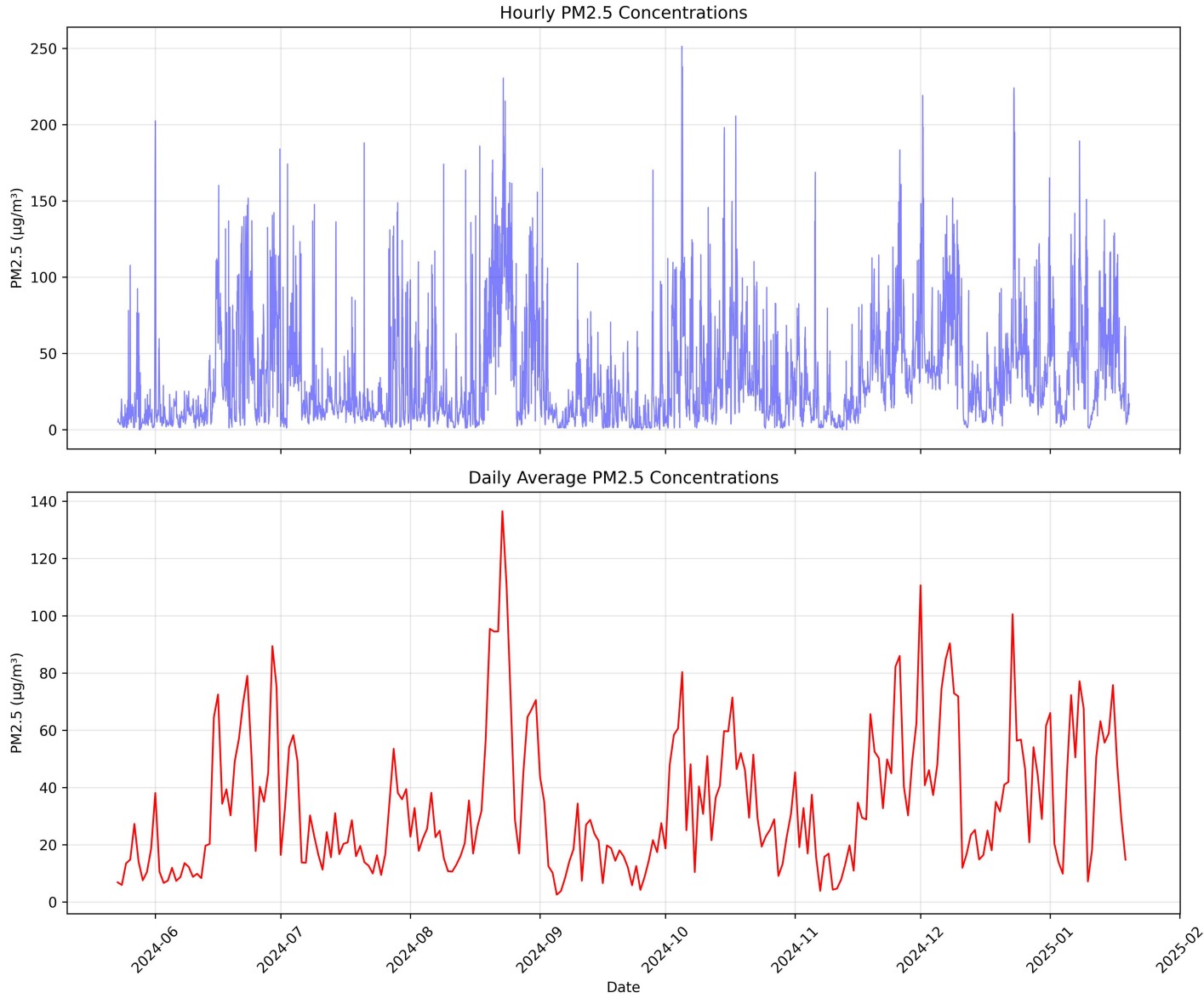

**Fig 21. Temporal dynamics of PM2.5 concentrations in Pavlodar (May 2024–January 2025): (upper) hourly values showing high-frequency variability and extreme events; (lower) daily average values revealing medium-term pollution patterns.**

(<15 μg/m³, below threshold), orange (15–30 μg/m³, 1–2 × exceedance), red (30–60 μg/m³, 2–4 × exceedance), purple (60–120 μg/m³, 4–8 × exceedance), and dark purple (>120 μg/m³, > 8 × exceedance). The percentage value for each month represents the proportion of hours exceeding the WHO threshold. The visualization reveals substantial seasonal variation, with winter months showing the highest exceedance rates (December: 85.6%, January: 77.4%) and late spring showing the lowest (May: 21.8%). Notable pollution episodes appear as clusters of red to purple cells, particularly visible in late June, late August, and throughout December, indicating persistent periods of poor air quality.

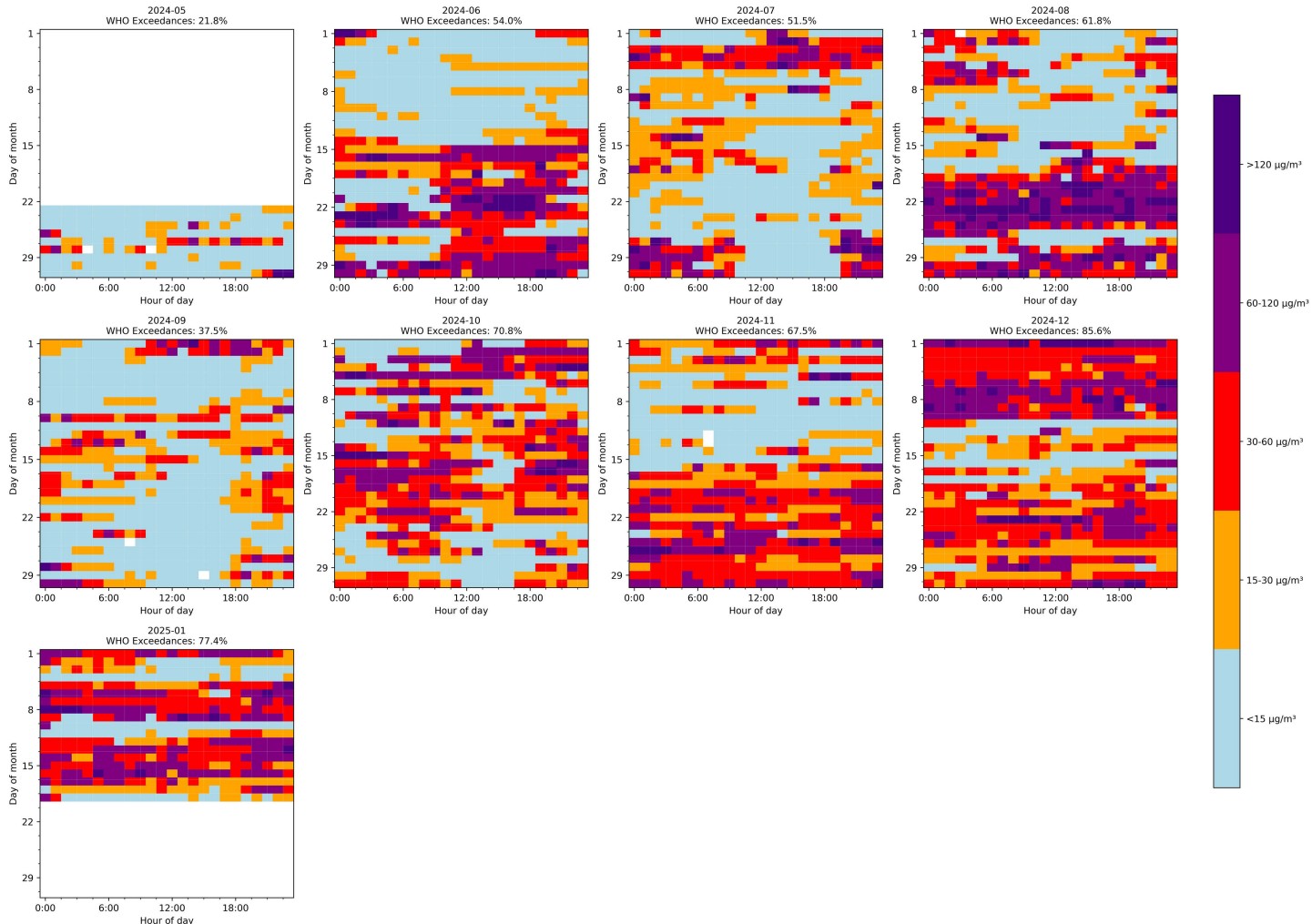

**Fig 22. Temporal distribution of PM2.5 WHO daily threshold exceedances (May 2024–January 2025).** The heatmap displays hourly PM2.5 concentrations relative to WHO daily guideline (15 µg/m³) across nine months of monitoring. Each cell represents one hour, with rows showing days of the month and columns showing hours of the day (0-23). Color legend: Light Blue = Below WHO threshold (<15 µg/m³), Orange = 1-2 × exceedance (15-30 µg/m³), Red = 2-4 × exceedance (30-60 µg/m³), Purple = 4-8 × exceedance (60-120 µg/m³), Dark Purple = >8 × exceedance (>120 µg/m³). Monthly exceedance percentages are displayed in the top-right corner of each panel, indicating the proportion of hours exceeding WHO guidelines.

**Analysis of air quality patterns.**  Air quality monitoring revealed distinct patterns in PM2.5 concentration levels across the study period. The Air Quality Index (AQI) values, calculated from PM2.5 measurements, showed considerable variability with a mean of 85.0 and median of 72.0 (Fig 23, right panel). The distribution of AQI values exhibited a bimodal pattern, with primary concentrations around 50 and secondary clusters near 150, suggesting the presence of two distinct air quality regimes in the monitoring location. The minimum recorded AQI was 0.0, while the maximum reached 302.0, indicating occasional episodes of severely degraded air quality.

When categorized according to standard AQI thresholds, the data revealed that approximately one-third (33.1%) of the monitoring period experienced "Good" air quality conditions (Fig 1, left panel). "Moderate" conditions were observed during 31.6% of the measurement period, while conditions classified as "Unhealthy for Sensitive Groups" accounted for 14.6% of observations. More concerning, air quality reached "Unhealthy" levels for nearly one-fifth (19.7%) of the

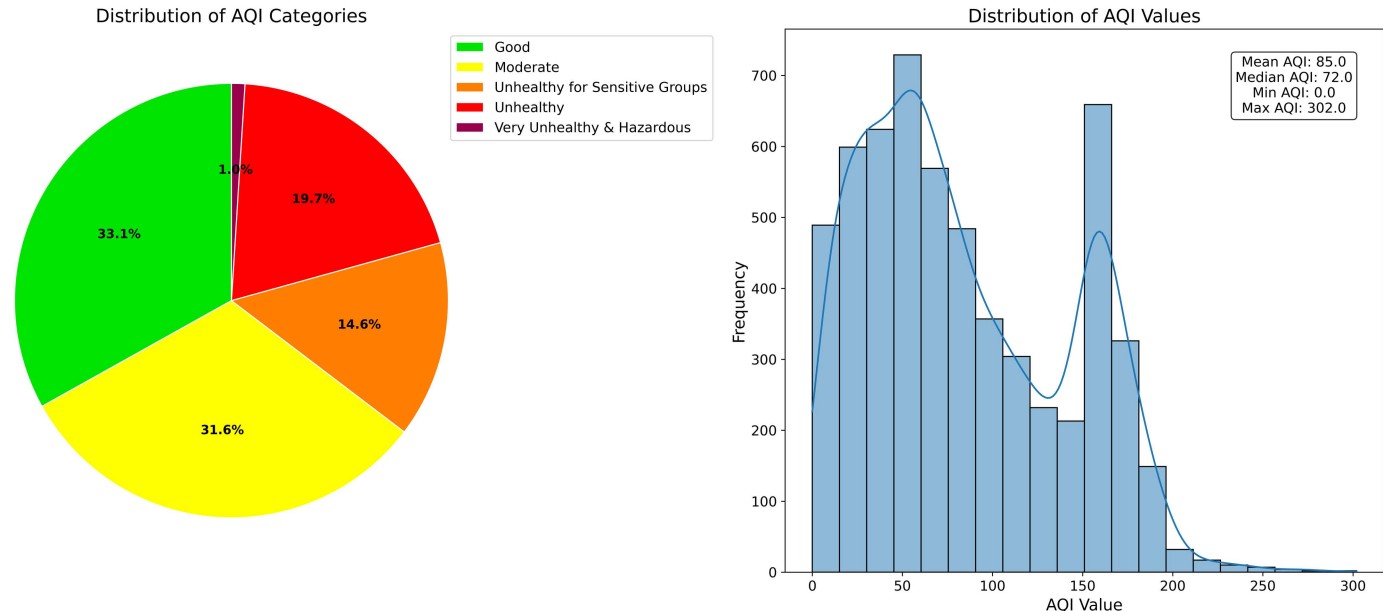

**Fig 23. Distribution of Air Quality Index (AQI) in the study area.** Left panel: Pie chart showing the proportion of time spent in each AQI category during the monitoring period. Right panel: Histogram displaying the frequency distribution of AQI values with density curve overlay and summary statistics.

monitoring period. "Very Unhealthy" conditions were relatively rare, occurring during only 1.0% of observations, while "Hazardous" conditions were exceptionally uncommon, registered during just 0.01% of the measurement period.

Temporal analysis of AQI categories revealed significant monthly variations in air quality conditions (Fig 24). May exhibited the highest proportion of "Good" air quality days (71.8%), followed by September (54.6%). In contrast, December and January showed the lowest percentages of "Good" air quality (11.2% and 18.2%, respectively). The winter months (December and January) demonstrated notably higher proportions of "Unhealthy" and "Unhealthy for Sensitive Groups" categories compared to summer months. December recorded the highest percentage of "Unhealthy for Sensitive Groups" days (27.8%), while January showed the greatest proportion of "Unhealthy" days (31.8%). These seasonal patterns indicate a clear deterioration in air quality during colder months, potentially attributable to increased heating emissions, reduced atmospheric mixing, and more frequent temperature inversions that trap pollutants near ground level.

The monthly distribution patterns further highlight the seasonal nature of air quality challenges in the study area, with consistently better conditions during spring and early autumn compared to winter. This seasonal variability underscores the importance of developing season-specific air quality management strategies and reinforces the need for continuous monitoring throughout the year to capture the full range of air quality conditions.

## Discussion

### Comparative assessment of gap-filling methodologies

This study provides a comprehensive evaluation of various gap-filling approaches for PM2.5 time series data, revealing significant variations in performance across different methodological categories and gap lengths. Our findings demonstrate that selecting the optimal imputation method depends strongly on the specific characteristics of data gaps and available contextual information.

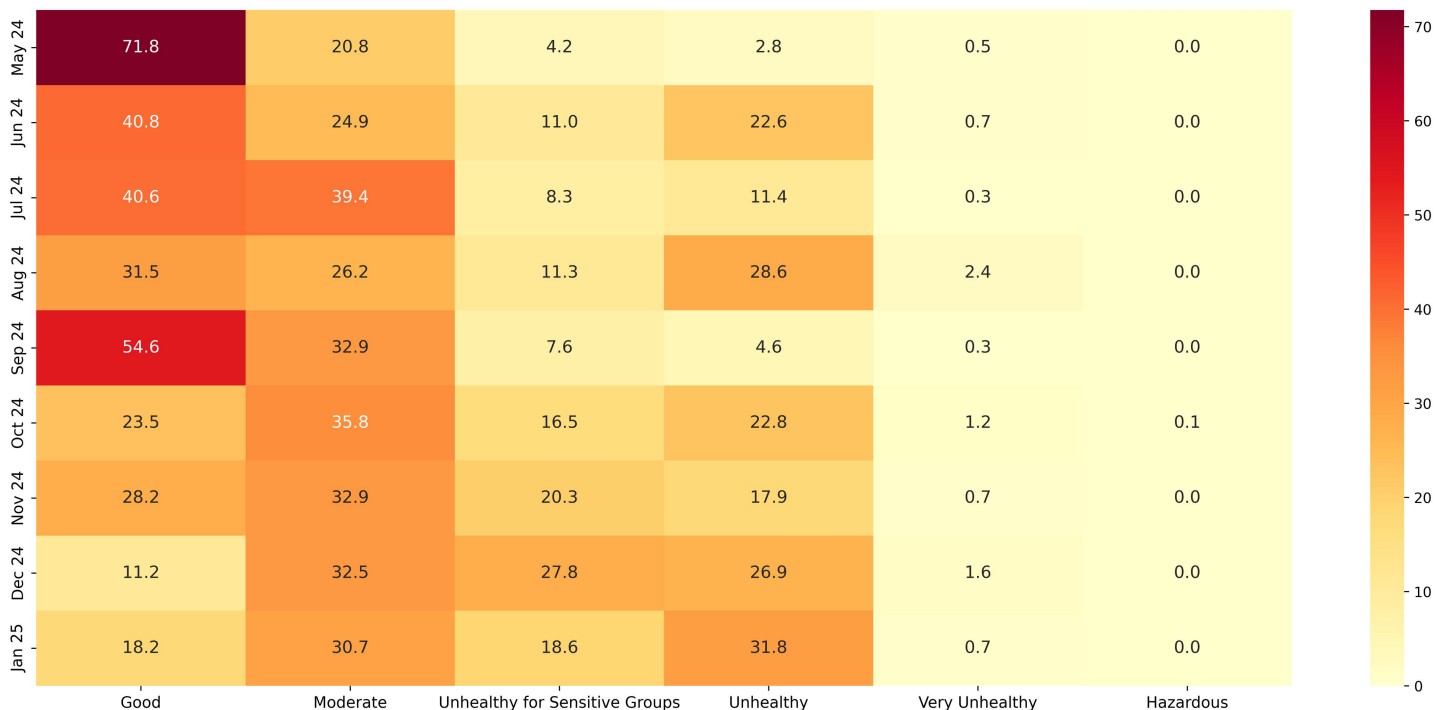

**Fig 24. Monthly distribution of Air Quality Index (AQI) categories.** The heatmap displays the percentage of time each month spent in different air quality categories, with color intensity indicating the percentage value.

Among the evaluated methods, tree-based models with bidirectional sequence-to-sequence architectures consistently delivered superior performance. The XGB Seq2Seq model achieved the lowest mean absolute error (MAE) of $5.231 \pm 0.292$ µg/m$^3$ for 12-hour gaps, representing a 63% improvement over basic statistical methods like Simple Imputer Mean ($14.351 \pm 0.438$ µg/m$^3$). This substantial performance gap underscores the inadequacy of conventional approaches like mean or median imputation for environmental time series with complex temporal dynamics.

The relative performance of different methodological categories followed a consistent pattern across gap lengths. Bidirectional approaches consistently outperformed their unidirectional counterparts by 5–15%, confirming the value of incorporating both past and future context when reconstructing missing segments. This advantage became particularly pronounced for longer gaps, where unidirectional approaches struggled to maintain prediction accuracy beyond the initial portion of the gap. Similarly, sequence-to-sequence architectures demonstrated clear advantages over autoregressive methods, with improvements of 8–12% in MAE. This finding suggests that direct prediction of entire segments is more effective than recursive one-step-ahead forecasting for environmental time series, likely due to the error propagation inherent in autoregressive approaches.

The incorporation of meteorological and temporal variables into multivariate models yielded increasingly substantial benefits as gap length increased. For short gaps (5 hours), multivariate models provided only marginal improvements (2–3% reduction in MAE) compared to their univariate counterparts. However, this advantage grew considerably for medium gaps (12–24 hours) to 6–7% and became particularly substantial for long gaps (48–72 hours), reaching 16–18% improvement. This pattern reflects the increasing importance of external contextual information when reconstructing extended missing segments, where the temporal autocorrelation of the target variable alone becomes insufficient for accurate prediction.

The observed relationship between model complexity and computational efficiency revealed practical considerations for operational implementation. While simple statistical methods completed each experimental run in less than one second, they delivered substantially inferior results compared to more sophisticated approaches. Tree-based models offered an attractive balance between performance and computational cost, with the XGB Seq2Seq model completing each run in approximately 8.2 seconds while delivering the best overall accuracy. Neural network architectures required considerably more computational resources (30–35 seconds per run) without providing commensurate performance improvements, suggesting that the additional complexity may not be justified for this particular application.

The development of dynamic models represents a significant advancement in addressing the practical challenges of real-world gap filling. By incorporating adaptive context sizing and unified training across different gap lengths, these models demonstrated remarkable flexibility in handling gaps of arbitrary duration and position with a single model. Though the dynamic models did not achieve the absolute lowest MAE scores compared to specialized static models trained for specific gap lengths, their performance remained robust across the entire range of tested gap durations. The multivariate dynamic model achieved MAE values ranging from $7.665 \pm 1.521$ µg/m$^3$ for 5-hour gaps to $10.772 \pm 6.214$ µg/m$^3$ for 72-hour gaps, maintaining reasonable accuracy even for the longest gaps tested.

The most compelling demonstration of the dynamic models' utility came during their application to real-world data with naturally occurring gaps. The ability to handle the extensive 191-hour gap–far exceeding the maximum length used during training (72 hours)–using adaptive chunking demonstrated exceptional generalization capability. Similarly, the natural handling of gaps with limited context (such as those near time series boundaries) and closely spaced gaps with insufficient data between them highlighted the practical advantages of this approach for operational monitoring systems.

Our comparison with previous studies on PM2.5 imputation reveals several notable advancements. While linear interpolation methods have been widely applied due to their simplicity, our results demonstrate that they achieve suboptimal accuracy (MAE of $7.842 \pm 0.477$ µg/m$^3$ for 12-hour gaps) compared to machine learning approaches. This finding aligns with Chen et al. [59], who reported that models developed with the generalized boosted machine, random forest and bagging performed slightly better than others in the full datasets, including traditional interpolation methods and deep learning methods. However, our work extends beyond previous studies by systematically evaluating a broader range of methodologies across multiple gap scenarios and demonstrating the substantial advantages of bidirectional sequence-to-sequence architectures, which have received limited attention in environmental time series imputation.

In the context of multivariate methods, our results complement the findings of Bai et al. [60], who reported that incorporating meteorological variables significantly improved PM2.5 prediction accuracy. Our study refines this understanding by demonstrating that the advantage of multivariate models increases with gap length, ranging from 2–3% for short gaps to 16–18% for extended gaps. This insight provides valuable guidance for practitioners regarding when the additional complexity of multivariate models is justified.

The dynamic modeling approach introduced in this study addresses a significant limitation identified by He et al. [61], who noted that gap-filling models often perform inconsistently across different missing data scenarios. By enabling a single model to adapt to gaps of varying lengths and positions, our dynamic architecture provides a more robust and generalizable solution for operational air quality monitoring systems.

## Meteorological determinants of PM2.5 concentrations

In Pavlodar, wind-related parameters proved to be the most influential determinants of PM2.5 concentrations. The moderate negative correlation between wind speed and PM2.5 levels ($r = -0.30$, $p < 0.001$) indicates that stronger winds enhance horizontal mixing and pollutant dilution. This conclusion is reinforced by the logarithmic regression analysis, where wind speed emerged as the strongest predictor ($\beta = -0.144$, $t = -23.59$, $p < 0.001$), with a marked decline in PM2.5 observed as wind speeds increased from 0–2 m/s to 4–6 m/s. Similar wind-dispersion effects have been reported in studies conducted across diverse regions, underscoring the universal role of wind dynamics in air quality regulation.

Wind direction also exerts a significant influence on PM2.5 levels (r=0.25, p<0.001). In Pavlodar, northerly and southerly winds correlate with higher particulate concentrations, a pattern that aligns well with the local urban geography—railway tracks lie approximately 200 m to the north, and a major road is situated 200 m to the south of the monitoring site. Such directional dependence has been noted in other urban studies, emphasizing the importance of local emission source distribution in modulating air pollution.

Temperature exhibits a more complex, non-linear relationship with PM2.5 concentrations in Pavlodar. A weak negative correlation (r=−0.22, p<0.001) suggests that colder conditions tend to coincide with higher pollutant concentrations, likely due to decreased vertical mixing and the trapping of emissions under inversion conditions. Our distribution analysis further revealed that extremely low temperatures (−25 to −20°C) are associated with peak PM2.5 values, while a moderate increase in concentrations above 25°C may indicate enhanced secondary aerosol formation through photochemical processes. Comparable studies in regions with different climate profiles have reported varying strengths of temperature effects, highlighting the context-dependency of thermal influences on particulate pollution. For instance, Yang et al. [62] reported weaker temperature correlations with PM2.5 in northern China (from −0.072 in Beijing till −0.380 in Baoding) despite its similar latitude, which may be attributed to differences in local topography and urbanization patterns.

Humidity in Pavlodar shows only a weak yet statistically significant positive association with PM2.5 (r=0.10, p<0.001; particulate concentrations tend to increase noticeably at the highest humidity levels (80–100%). This modest relationship may be attributed to mechanisms such as hygroscopic growth, which enhances particle detection, and the stabilization of the atmospheric boundary layer under humid conditions that limits vertical dispersion. By comparison, research in urban areas like Delhi has reported more pronounced humidity effects, especially in residential and industrial districts such as Rohini and Okhla [63]. Such differences likely reflect the lower average humidity levels characteristic of the continental climate in Pavlodar and underscore that the influence of humidity on PM2.5 is strongly dependent on the local climatic context.

Visibility, as expected, exhibited a negative correlation with PM2.5 concentrations (r=−0.15, p<0.001). This observation is consistent with the established physical relationship between particulate matter and light attenuation. However, the relatively modest strength of this correlation may stem from the fact that visibility is also affected by other atmospheric constituents such as gaseous pollutants and water vapor—a nuance also observed in studies from other regions.

Seasonal variations are pronounced in Pavlodar; winter months (December–January) show PM2.5 concentrations that are 35–40% higher than those observed in summer. This seasonal pattern results from a combination of factors including altered emission patterns (e.g., increased heating emissions in winter), a lower atmospheric mixing height during colder months, and the formation of stagnant air masses. Comparable winter-dominated trends have been documented in northern China [64] and central Europe [65], although regional differences in temperature and humidity regimes contribute to each locale's unique air quality profile.

Temporal analysis revealed clear diurnal cycles with distinct morning (7:00–9:00) and evening (18:00–21:00) peaks. These patterns reflect the interplay between human activity—such as rush hour traffic—and diurnal variations in the boundary layer's structure. The morning peak likely arises from overnight cooling and subsequent traffic-induced emissions, while the evening peak coincides with both increased commuter activity and reduced vertical mixing. Similar bimodal patterns have been observed in cities like Beijing, Delhi, and London [66–68], suggesting that these diurnal trends are common in urban settings worldwide.

Overall, these findings illustrate that while certain meteorological parameters such as wind speed and direction exert universal influences on PM2.5 dispersion, other factors—including temperature, humidity, seasonal, and diurnal variations—display context-dependent characteristics. In Pavlodar, for example, the limited effect of humidity compared to the more pronounced associations reported in regions like Delhi emphasizes that the relative importance of meteorological determinants in shaping air pollution levels strongly depends on local climatic conditions and urban configurations.

The observed seasonal patterns in PM2.5 concentrations and their implications for gap-filling accuracy reflect complex interactions between emission sources and meteorological conditions. Winter dominance of high concentrations (45.87 µg/m³ vs 13.24 µg/m³ in spring) likely results from increased heating-related emissions combined with unfavorable atmospheric dispersion conditions. The strong negative correlation between temperature and PM2.5 ($\beta = -0.020$, $p < 0.001$) and the pronounced effect of wind speed ($\beta = -0.144$, $p < 0.001$) suggest that winter atmospheric stability contributes significantly to pollutant accumulation. Conversely, summer patterns may reflect different source profiles including resuspended dust and photochemical processes, as evidenced by the moderate concentration increase observed at temperatures above 25°C. These seasonal source and meteorological variations directly impact gap-filling model performance, with multivariate models showing greater seasonal accuracy improvements (16–18% for extended gaps) by incorporating the meteorological context that governs these underlying physical processes.

Our findings regarding the interaction between meteorological parameters and temporal patterns contribute to the growing literature on context-specific air quality dynamics. The effectiveness of multivariate models incorporating both meteorological and temporal features for gap-filling, particularly for extended gaps, demonstrates the practical value of understanding these relationships. By identifying the most influential environmental factors and their complex interactions with PM2.5 concentrations, our study provides not only improved methodologies for data reconstruction but also valuable insights for developing more effective air quality management strategies tailored to local conditions.

## Practical applications of the developed models

The dynamic multivariate models developed in this study offer significant practical applications for environmental monitoring and public health management. The successful implementation of these models in real-world scenarios can enhance the reliability and continuity of air quality data, providing valuable insights for various stakeholders.

In the context of urban air quality monitoring networks, our models address a critical challenge of intermittent data availability. Environmental monitoring stations often experience periodic malfunctions, power outages, or scheduled maintenance that result in data gaps of varying durations. Rather than discarding incomplete data series or accepting substantial discontinuities, our adaptive gap-filling approach enables the reconstruction of continuous PM2.5 time series while preserving important temporal patterns and pollution dynamics. This capability is particularly valuable for long-term trend analysis, epidemiological studies, and regulatory compliance assessment, where consistent and complete datasets are essential.

The dual univariate and multivariate implementations provide operational flexibility based on data availability. When meteorological measurements are consistently available, the multivariate approach offers superior accuracy, especially for longer gaps. In monitoring scenarios where supporting meteorological measurements are unreliable or unavailable, the univariate implementation still provides reasonable gap-filling performance. This adaptability makes the methodology applicable across diverse monitoring environments with varying instrumentation capabilities.

Beyond retrospective data completion, our models can be adapted for near-real-time applications in operational air quality management systems. As measurements are collected, the system can automatically detect missing values and apply the appropriate gap-filling algorithms to maintain continuous data streams for visualization platforms, public information systems, and automated alerts. The computational efficiency of our XGBoost-based approach, which demonstrated processing times of under a minute for typical gap lengths, further supports this real-time implementation potential.

The models also show promise for short-term predictive applications. While primarily designed for gap-filling, the ability to accurately predict future values based on historical patterns and meteorological conditions could be extended to generate air quality forecasts with 24–48 hour horizons. This functionality would be particularly valuable for early warning systems that could notify vulnerable populations (elderly individuals, children, and those with respiratory or cardiovascular conditions) about potential air quality deterioration, allowing for preventive actions before exposure to harmful pollution levels.

However, several important limitations should be considered when implementing these models in operational settings. First, while our approach effectively generalizes across varying gap lengths, the prediction accuracy notably decreases for gaps extending beyond 72 hours. For monitoring stations experiencing extended outages (weeks or months), the model would benefit from incorporating additional spatial data from nearby stations to constrain predictions over such extended periods.

Second, the meteorological variables incorporated in our multivariate model capture only a portion of the complex factors influencing PM2.5 concentrations. Additional parameters such as boundary layer height, precipitation, and local emission inventories could potentially further improve prediction accuracy but would increase data requirements and implementation complexity.

Third, the current implementation focuses specifically on PM2.5, but monitoring networks typically measure multiple pollutants simultaneously. Extending the methodology to handle multi-pollutant reconstruction, potentially exploring inter-pollutant relationships, represents an important direction for operational enhancement.

For effective implementation in monitoring networks, we recommend a phased approach beginning with offline validation using historical data specific to the deployment location. The model parameters, especially context window sizes and feature selection, should be customized based on local pollution dynamics and data availability patterns. Regular retraining (quarterly or biannually) would maintain model relevance as seasonal patterns evolve and urban emission characteristics change.

To facilitate broader adoption, the models should be integrated into existing air quality data management systems through standardized APIs and processing pipelines. Documentation and training materials should emphasize the interpretation of gap-filled data, particularly for longer gaps where uncertainty is higher. Visual indicators of data provenance (observed versus imputed) in monitoring dashboards would maintain transparency for end-users while still providing continuous data streams.

In summary, the dynamic gap-filling models developed in this study offer practical tools for enhancing air quality monitoring capabilities, supporting both operational data management and public health applications. With appropriate implementation considerations, these models can significantly improve the completeness and utility of PM2.5 monitoring data in urban environments.

## Public health and air quality policy implications

The analysis of PM2.5 concentrations in relation to WHO guidelines reveals concerning patterns with significant public health implications. Our findings that 61.2% of the monitored hours exceeded the WHO daily threshold of 15 µg/m³ indicate that residents in the study area face substantial exposure to potentially harmful levels of fine particulate matter. This prolonged exposure represents a significant public health challenge that warrants targeted intervention strategies.

The severity distribution of these exceedances is particularly alarming. While moderate exceedances (15–30 µg/m³) occurred during 20.7% of the monitoring period, severe concentrations reaching 2–4 times the WHO threshold were observed during 21.7% of hours. Even more concerning were the extended periods with concentrations 4–8 times the threshold (15.6% of hours) and extreme values exceeding 8 times the recommended limit (3.1% of hours). These higher concentration events, though less frequent, may contribute disproportionately to adverse health outcomes due to their intensity.

The pronounced seasonal patterns in threshold exceedances–with winter months (December and January) showing rates of 85.6% and 77.4% respectively–suggest that cold-weather pollution episodes represent a critical intervention point. This seasonal effect aligns with our observation of significantly higher mean PM2.5 concentrations in winter (45.87 µg/m³) compared to other seasons, particularly spring (13.24 µg/m³). The consistent diurnal patterns, with greater frequency and severity of exceedances during evening and early morning hours, further refines the temporal targeting for potential mitigation strategies.

From a public health perspective, these findings suggest a substantial population-level risk for both acute and chronic health effects. Based on established concentration-response relationships in epidemiological literature, the observed PM2.5 levels are associated with increased risk of respiratory infections, exacerbation of asthma and COPD, cardiovascular events, and potential long-term effects including reduced lung function development in children. The winter predominance of exceedances may create a seasonal burden on healthcare systems, particularly for cardiovascular and respiratory services.

The monthly distribution of Air Quality Index (AQI) categories further contextualizes these health implications. During December and January, "Unhealthy" conditions prevailed for approximately 30% of the time, significantly limiting outdoor activity options for sensitive populations during these months. Even more broadly concerning is that air quality reached "Unhealthy" levels for nearly one-fifth (19.7%) of the entire monitoring period, indicating that the potential health burden extends beyond seasonal effects.

Based on these findings, we recommend several policy interventions for local authorities. First, targeted emission reduction strategies should be implemented specifically for winter months, focusing on heating-related sources which likely contribute significantly to the seasonal pattern. This could include incentives for transitioning to cleaner heating technologies, temporary restrictions on high-emitting activities during adverse meteorological conditions, and enhanced public transportation options to reduce traffic emissions during winter.

Second, a public health communication strategy should be developed that accounts for both the seasonal and diurnal patterns identified in our analysis. This should include a real-time alert system that provides timely information about air quality conditions, particularly during evening and early morning hours when exceedances are most frequent. Special attention should be directed to vulnerable populations, including the elderly, children, and individuals with pre-existing respiratory or cardiovascular conditions.

Third, urban planning and infrastructure development should incorporate air quality considerations, particularly regarding the spatial distribution of emission sources relative to residential areas. The observed wind direction effects on PM2.5 concentrations suggest that understanding local pollution transport patterns could inform more effective zoning and development policies.

Finally, we recommend expanding the monitoring network to provide more comprehensive spatial coverage while implementing the gap-filling methodology developed in this study to ensure continuous data availability. This enhanced monitoring capacity would support more precise identification of pollution hotspots and evaluation of intervention effectiveness.

In conclusion, the air quality patterns revealed by our analysis present clear challenges for public health in the study area, but also offer evidence-based guidance for targeted interventions. By focusing on the specific temporal patterns of PM2.5 exceedances, local authorities can more efficiently allocate resources to reduce population exposure and associated health burdens. Further epidemiological research specifically examining health outcomes in relation to the observed pollution patterns would strengthen the evidence base for such policy interventions.

## Study limitations and future research directions

Despite the comprehensive approach taken in this study, several limitations should be acknowledged when interpreting our findings and considering their broader applicability. These limitations span data characteristics, methodological constraints, and the scope of implementation.

The primary data limitation concerns the temporal coverage of our dataset, which spans from May 2024 to January 2025, with notable gaps in February, March, and April. This incomplete annual cycle may have affected our characterization of seasonal patterns, particularly the spring transition period. Moreover, the single-year timeframe prevents year-by-year cross-validation and assessment of model robustness across different annual meteorological regimes (e.g., drought versus wet years), limiting our ability to evaluate long-term model stability under diverse climatic conditions.

Additionally, while the 1-minute sampling interval provided high-resolution data, the aggregation to hourly values necessary for gap-filling modeling involved averaging that may have obscured short-term pollution spikes. The study was also constrained geographically to a single monitoring location, limiting our ability to capture spatial heterogeneity in PM2.5 concentrations across the urban environment.

From a methodological perspective, our synthetic gap testing approach, while robust, may not fully replicate the complex patterns of missingness encountered in real-world monitoring scenarios. Although we varied gap lengths and placement, other characteristics such as simultaneous gaps across multiple monitoring parameters or non-random patterns of system failure were not explicitly modeled. Furthermore, our evaluation metrics focused primarily on statistical accuracy rather than preservation of key environmental patterns that might have specific health relevance, such as acute pollution episodes or diurnal transition characteristics.

The gap-filling models, particularly the multivariate implementations, relied on a selected subset of meteorological variables that were consistently available throughout the study period. While our feature selection process was data-driven, the limited meteorological parameters available may have constrained the models' explanatory power. Important factors such as boundary layer height, precipitation patterns, and local traffic volumes were not incorporated due to data availability constraints.

For long-duration gaps, model uncertainty increases substantially due to limited temporal context and potential shifts in pollution patterns. Our analysis reveals that prediction reliability decreases notably for gaps exceeding 48 hours, as evidenced by increased standard deviations in performance metrics (MAE standard deviation increases from ±1.5 µg/m$^3$ for 5-hour gaps to ±6.2 µg/m$^3$ for 72-hour gaps). This variability reflects inherent uncertainty in reconstructing extended missing segments, particularly during periods with high meteorological variability or episodic pollution events. Users should interpret filled values for gaps >48 hours with appropriate caution and consider implementing ensemble approaches or additional validation using nearby monitoring stations when available.

For future research, several promising directions emerge from these limitations. First, extending the monitoring period to encompass multiple annual cycles would significantly strengthen the temporal robustness of both the characterization of PM2.5 patterns and the gap-filling models. This would allow for more reliable assessment of interannual variability and potential long-term trends in concentration patterns.

Second, incorporating spatial elements into the modeling framework represents a critical advancement opportunity. By integrating data from multiple monitoring stations, future models could leverage spatial correlations to improve prediction accuracy, particularly for extended gaps. Such an approach could evolve toward spatiotemporal kriging methods that simultaneously address gaps across a monitoring network while accounting for the spatial distribution of pollution sources and atmospheric transport patterns.

Third, methodological refinements to the gap-filling approach could include the development of uncertainty quantification frameworks that provide confidence intervals for filled values rather than point estimates alone. This would enhance the utility of gap-filled data for both scientific analysis and policy applications by explicitly acknowledging the varying reliability of imputed values across different gap contexts.

Fourth, the integration of additional data sources beyond traditional meteorological measurements could significantly enhance model performance. Satellite-derived aerosol optical depth, land use regression variables, traffic intensity data, and energy consumption metrics could provide valuable contextual information for PM2.5 prediction. Similarly, incorporating citizen science data from low-cost sensor networks could augment the spatial resolution of monitoring while introducing new methodological challenges regarding data quality and calibration.

Fifth, extending the modeling approach to other air pollutants (ozone, nitrogen dioxide, sulfur dioxide) would provide a more comprehensive picture of air quality dynamics and potentially leverage inter-pollutant relationships to improve prediction accuracy across parameters. Similarly, applying the methodology to different urban environments with varying emission profiles, meteorological conditions, and topographical characteristics would test the generalizability of our findings and potentially lead to more robust modeling frameworks.

 

Comprehensive benchmarking against cutting-edge dynamic filling methods (advanced RNN variants, state-space models, and adaptive neural architectures) represents an important direction for future research. Such comparisons would require specialized implementation of alternative dynamic approaches and careful adaptation to the retrospective gap-filling context, providing valuable insights into the relative advantages of different dynamic modeling paradigms for environmental time series reconstruction.

While our meteorological analysis provides insights into seasonal patterns, comprehensive source apportionment analysis using backward trajectory modeling (e.g., HYSPLIT) and detailed emission inventories would enhance understanding of the mechanistic basis for seasonal gap-filling accuracy differences. Such analysis could better quantify the relative contributions of winter heating sources versus summer dust emissions to observed concentration patterns and their impact on model performance.

Finally, further research on the health implications of gaps in air quality data would be valuable. Understanding how data missingness affects exposure assessment in epidemiological studies, public health messaging, and regulatory compliance would provide important context for evaluating gap-filling methodologies beyond purely statistical metrics.

By addressing these limitations and pursuing these future research directions, the scientific community can build upon the foundation established in this study to develop increasingly robust and practical approaches to the persistent challenge of incomplete environmental monitoring data. Such advancements would ultimately contribute to more effective air quality management and public health protection strategies.

## Conclusion

This comprehensive evaluation of 46 gap-filling methodologies for PM2.5 time series provides clear guidance for environmental monitoring practitioners. Tree-based models with bidirectional sequence-to-sequence architectures, particularly XGB Seq2Seq, represent the optimal choice for accuracy-critical applications, while Dynamic Multivariate models offer superior operational flexibility for real-world deployment. The incorporation of meteorological variables becomes increasingly valuable with gap length, essential for gaps exceeding 48 hours where multivariate approaches provide 16–18% performance improvements over univariate methods.

Continuous air quality monitoring data is critical for environmental research, public health assessment, and policy development, yet significant challenges persist due to data gaps resulting from sensor malfunctions, maintenance, and other operational issues. This study addressed the critical problem of filling gaps in PM2.5 time series data collected in Pavlodar, Kazakhstan from May 2024 to January 2025. We developed and evaluated a comprehensive hierarchy of gap-filling methodologies, ranging from simple statistical approaches to sophisticated machine learning models, and assessed their effectiveness across various gap lengths (5–72 hours).

Our systematic methodology involved creating a unified testing framework with synthetic gap insertion, standardized evaluation protocols, and statistical robustness through multiple experimental runs. We extended traditional univariate approaches by incorporating meteorological variables (wind speed, wind direction, air temperature, humidity) and temporal features (hour of day, season) in multivariate models. Additionally, we introduced dynamic models capable of adapting to gaps of any length with a single flexible architecture, addressing real-world operational challenges.

The key findings of this study can be summarized as follows:

1. Tree-based models with bidirectional sequence-to-sequence architectures consistently delivered superior performance across all gap lengths, with XGB Seq2Seq achieving the lowest mean absolute error (MAE) of $5.231 \pm 0.292$ µg/m$^3$ for 12-hour gaps, representing a 63% improvement over basic statistical methods like Simple Imputer Mean ($14.351 \pm 0.438$ µg/m$^3$).

2. The advantage of multivariate models incorporating meteorological and temporal variables increased substantially with gap length, from modest improvements of 2–3% for 5-hour gaps to significant enhancements of 16–18% for 48–72

hour gaps, demonstrating that environmental context becomes increasingly valuable for reconstructing extended missing segments.

3. Dynamic multivariate models demonstrated remarkable operational flexibility by handling gaps of any length with a single model, maintaining acceptable accuracy across diverse gap durations (MAE from $7.665 \pm 1.521$ μg/m³ for 5-hour gaps to $10.772 \pm 6.214$ μg/m³ for 72-hour gaps), and successfully processing real-world gaps ranging from 1 to 191 hours despite being trained on maximum gap lengths of 72 hours.

4. Analysis of the reconstructed complete PM2.5 time series revealed significant air quality challenges in the study area, with 61.2% of monitored hours exceeding the WHO daily threshold of 15 μg/m³, strong seasonal patterns (winter concentrations 35–40% higher than summer), and pronounced diurnal cycles with distinct morning (7:00–9:00) and evening (18:00–21:00) peaks.

5. Meteorological factors showed significant influence on PM2.5 concentrations, with wind speed exhibiting the strongest negative correlation ($r = -0.30$, $p < 0.001$) and emerging as the most influential predictor in logarithmic regression analysis ($\beta = -0.144$, $t = -23.59$, $p < 0.001$), while temperature demonstrated a non-linear relationship with elevated concentrations at both extremely low temperatures (–25 to –20°C) and high temperatures (above 25°C).

Beyond methodological advances, this research reveals critical public health implications requiring immediate policy attention. The analysis of reconstructed PM2.5 data demonstrates that 61.2% of monitored hours exceeded WHO guidelines, with winter months showing particularly severe exceedance rates of 85.6% (December) and 77.4% (January). These findings indicate substantial population exposure risks during specific temporal windows, providing actionable targets for policy interventions. The identification of high-risk periods–evening hours (18:00–21:00) with pronounced pollution peaks and winter heating seasons with concentrations 35–40% higher than summer–enables targeted emission control strategies and public health advisories. The continuous monitoring capability achieved through our gap-filling methodology ensures that policy decisions are based on complete temporal coverage rather than the 73.3% data completeness of raw measurements, preventing potential underestimation of pollution exposure during critical episodes that occurred within the 26.7% of time when original measurements were unavailable.

This research provides not only methodological advancements in environmental time series reconstruction but also valuable insights into the temporal dynamics and meteorological determinants of PM2.5 concentrations in an urban environment. The developed gap-filling approach enhances the reliability and continuity of air quality monitoring data, supporting more effective public health strategies and environmental policy development. Future research should focus on integrating spatial elements into the modeling framework, incorporating additional data sources beyond traditional meteorological measurements, and extending the methodology to other air pollutants to provide a more comprehensive picture of air quality dynamics.

## Supporting information

**S1 File. Meteorological Data.**
(CSV)

**S2 File. Supporting Figures.**
(ZIP)

**S1 Fig. Research-flowchart-final-fixed.**
(PNG)

## Acknowledgments

The authors would like to express their gratitude to all individuals and organizations who contributed to this research through technical support, data provision, or academic guidance. Special thanks to colleagues who provided constructive feedback during the preparation of this manuscript.

## Author contributions

**Conceptualization:** Ruslan Safarov.

**Data curation:** Zhanat Shomanova, Ruslan Kamatov.

**Formal analysis:** Eldar Kopishev.

**Funding acquisition:** Zhanat Shomanova.

**Investigation:** Yuriy Nossenko, Zhuldyz Bexeitova.

**Methodology:** Yuriy Nossenko.

**Project administration:** Ruslan Safarov.

**Resources:** Eldar Kopishev.

**Software:** Ruslan Kamatov.

**Supervision:** Ruslan Safarov.

**Validation:** Zhanat Shomanova.

**Visualization:** Ruslan Kamatov.

**Writing–original draft:** Yuriy Nossenko, Zhuldyz Bexeitova.

**Writing–review & editing:** Ruslan Safarov.

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
