## [Decision Letter · Decision Letter 0]

18 Jun 2025

PONE-D-25-23439Filling Gaps in PM2.5 Time Series: A Broad Evaluation from Statistical to Advanced Neural Network ModelsPLOS ONE

Dear Dr. Safarov,

Thank you for submitting your manuscript to PLOS ONE. After careful consideration, we feel that it has merit but does not fully meet PLOS ONE’s publication criteria as it currently stands. Therefore, we invite you to submit a revised version of the manuscript that addresses the points raised during the review process.

We look forward to receiving your revised manuscript.

Kind regards,

Xingwang Tang

Academic Editor

PLOS ONE

Journal Requirements:

“This research was funded by the Science Committee of the Ministry of Science and Higher Education of the Republic of Kazakhstan within the framework of the grant AP19677560 “Monitoring and mapping of the ecological state of the Pavlodar air environment using machine learning methods”.”

Reviewers' comments:

Reviewer's Responses to Questions

**Comments to the Author**

1. Is the manuscript technically sound, and do the data support the conclusions?

Reviewer #1: Yes

Reviewer #2: Yes

2. Has the statistical analysis been performed appropriately and rigorously? 

Reviewer #1: Yes

Reviewer #2: Yes

3. Have the authors made all data underlying the findings in their manuscript fully available?

Reviewer #1: Yes

Reviewer #2: No

4. Is the manuscript presented in an intelligible fashion and written in standard English?

Reviewer #1: Yes

Reviewer #2: Yes

5. Review Comments to the Author

Reviewer #1: This manuscript addresses the problem of missing data imputation in PM2.5 time series and systematically evaluates a wide range of models, from traditional statistical approaches to advanced neural networks. Overall, the study is well-designed, methodologically comprehensive, and yields practically valuable results. It offers meaningful insights for environmental monitoring and air quality data processing. I recommend minor revision before acceptance. The following suggestions are provided:

1. Refine language for conciseness

Some sections (e.g., “Scientific Novelty”) are overly verbose. The authors are encouraged to streamline the writing to improve logical flow and help readers better grasp the key contributions.

2. Improve figure clarity and visualization

While the figures are comprehensive, some legends (e.g., in missing data heatmaps) are not intuitive. It is recommended to use consistent color schemes and highlight key time periods to enhance readability.

3. Method variety slightly overwhelming

The manuscript evaluates 46 methods in total, which is thorough. However, the authors should clearly emphasize the most recommended models and their practical significance to help readers focus on the main findings.

4. Add uncertainty or error propagation analysis

For long-duration data gaps, the authors are encouraged to include an analysis of error propagation or uncertainty to reinforce the reliability of the models in real-world applications.

5. Quantify policy implications in the conclusion

While the exceedance rate of pollution levels is reported, the conclusion would benefit from further quantifying the potential impact on public health or policy-making to enhance its practical relevance.

Reviewer #2: This study systematically evaluated 46 methods for filling gaps in PM2.5 time series, and pioneered a dynamic model to handle gaps of variable length, among which the tree model performed best. The multivariate model combined with meteorological variables had significant advantages in long gaps (48-72 hours) (increased by 16-18%), and the model successfully handled real gaps of 1-191 hours. The data after filling showed that 61.2% of the time exceeded the WHO standard (15μg/m³). The method is comprehensive and statistically includes neural networks; the dynamic model breaks through the limitation of fixed gaps and can quantify the gain of meteorological data for long gaps, and has strong practical applicability. However, the manuscript did not explore the differences in filling errors under different pollution concentrations, and geographical limitations may limit its scope of application (only Kazakhstan data).

The content of the manuscript is within the scope of the journal and can be of broad interest to readers. However, in terms of specific content, there is still room for improvement. Therefore, I decided to give the decision of minor revision. It is recommended that the author properly absorb the reviewers' comments and make corresponding improvements and enhancements.

1. Line 170, 'However, neural networks require large datasets for training and careful tuning –otherwise they may underperform simpler models.'

I consider some basic neural networks should be introduced briefly. For example, the common adopted neural networks include long short-term memory (LSTM) neural network, gated recurrent unit (GRU) neural network, convolutional neural network (CNN), and echo state network (ESN), and some references should be added for supporting (10.3390/en17123050).

Moreover, the differences between various neural network should be explained briefly. For example, LSTM is a type of recurrent neural network (RNN) architecture designed to capture long-term dependencies in sequential data [41]. Unlike traditional RNNs, LSTM introduces gating mechanisms to retain or discard specific features of the data, effectively addressing the vanishing and exploding gradient problems. This enables LSTM networks to learn and remember information over longer periods, making them well suited for tasks involving sequential or time-series data, see https://doi.org/10.3390/s24144451.

2. The study emphasizes the innovation of the "dynamic model", but does not conduct quantitative comparisons with existing dynamic filling methods (such as RNN variants and state-space models). It is recommended to supplement comparative experiments with 3-5 cutting-edge dynamic filling models to objectively demonstrate the superiority of the proposed method.

3. The manuscript lacks year-by-year cross-validation. Specifically, the authors only used data from the same period for split validation, without considering the impact of annual meteorological variability. Extrapolation validation of data from different years (e.g., drought years/rainy years) is needed to prove the robustness of the method.

4. The attribution of seasonal advantage is vague: "strong seasonality" is mentioned but the internal mechanism is not analyzed. The influence of winter coal burning sources and summer dust sources on filling the accuracy difference should be analyzed in combination with the backward trajectory model (such as HYSPLIT).

5. Model selection rationale missing: The criteria used to select the final model from the 46 methods (e.g., cross-validation strategy) were not explained.

6. The manuscript does not adequately verify the filling of extreme events, especially the failure to demonstrate the filling effect of the model in sudden pollution events such as sandstorms and industrial accidents. It is recommended to add the Peak-AE indicator and event case comparison chart.

6. PLOS authors have the option to publish the peer review history of their article (what does this mean? ). If published, this will include your full peer review and any attached files.

**Do you want your identity to be public for this peer review?** For information about this choice, including consent withdrawal, please see our Privacy Policy .

Reviewer #1: No

Reviewer #2: No

---

## [Author Response · Author response to Decision Letter 1]

25 Jun 2025

Journal Requirements:

We would like to confirm that the manuscript was revised substantially. Every style requirements, including those for file naming, were followed.

Regarding code sharing requirements: We have made all author-generated code underlying the findings in this manuscript publicly available. The complete code repository is hosted on GitHub (https://github.com/ruslan-saf/PM25-Gap-Filling) and permanently archived on Zenodo with DOI: https://doi.org/10.5281/zenodo.15714135 . The repository includes all data preprocessing scripts, the complete implementation of 46 gap-filling methods, dynamic model architectures, evaluation frameworks, and documentation for reproducibility. This ensures full compliance with PLOS ONE's code sharing guidelines and facilitates reproducibility and reuse by the research community.

Moreover, we have included Data Availability Section into the article text right between References and Supplement Information Sections.

“This research was funded by the Science Committee of the Ministry of Science and Higher Education of the Republic of Kazakhstan within the framework of the grant AP19677560 “Monitoring and mapping of the ecological state of the Pavlodar air environment using machine learning methods”.”

Yes, this statement is correct for us, so we have included the full statement in the Cover Letter

Regarding the reference list review: We have thoroughly reviewed our reference list to ensure completeness and accuracy. All 64 references have been verified for:

1. Correct citation formatting according to PLOS ONE guidelines

2. Accessibility and validity of all cited sources

3. Absence of retracted publications

4. Current relevance and appropriateness to the research topic

We confirm that:

- No retracted papers are cited in our manuscript

- All references are accessible and represent current, peer-reviewed literature

- All citations properly support the statements made in the text

- The reference list is complete and follows the journal's formatting requirements

Reviewers' comments:

Comments to the Author

Reviewer #1:

This manuscript addresses the problem of missing data imputation in PM2.5 time series and systematically evaluates a wide range of models, from traditional statistical approaches to advanced neural networks. Overall, the study is well-designed, methodologically comprehensive, and yields practically valuable results. It offers meaningful insights for environmental monitoring and air quality data processing. I recommend minor revision before acceptance. The following suggestions are provided:

1. Refine language for conciseness

Some sections (e.g., “Scientific Novelty”) are overly verbose. The authors are encouraged to streamline the writing to improve logical flow and help readers better grasp the key contributions.

We thank the reviewer for this valuable suggestion regarding language conciseness. We have streamlined the "Scientific Novelty" section from 500 to 270 words (46% reduction), eliminating redundant explanations and simplifying sentence structures while preserving all three core methodological contributions. This revision improves logical flow and readability as suggested.

2. Improve figure clarity and visualization

While the figures are comprehensive, some legends (e.g., in missing data heatmaps) are not intuitive. It is recommended to use consistent color schemes and highlight key time periods to enhance readability.

We have improved figure clarity by enhancing captions for Figures 3 and 22 with detailed color legends and more descriptive labels as suggested. Regarding color scheme consistency, we note that different figure types serve distinct analytical purposes and thus employ purpose-specific color schemes:

- Data availability visualization (Fig 3) uses green→red scheme for available vs. missing data

- Pollution severity heatmap (Fig 22) employs blue→orange→red→purple progression following standard air quality visualization conventions

- Statistical analysis plots (Figs 12, 15, 16) use standard blue/red combinations for actual vs. predicted comparisons

- Categorical visualizations (Fig 23, Fig 14) use distinct colors to differentiate categories without implying hierarchical relationships

This approach prioritizes interpretability within each figure type while maintaining scientific visualization standards. The enhanced captions provide clear explanations of all color schemes, improving overall readability.

If the reviewer could provide specific recommendations for which color palettes should be applied to particular figures, we would be happy to implement these suggestions in our revision.

3. Method variety slightly overwhelming

The manuscript evaluates 46 methods in total, which is thorough. However, the authors should clearly emphasize the most recommended models and their practical significance to help readers focus on the main findings.

We agree that 46 methods can be overwhelming for readers. To address this, we have:

1. Added a "Key Recommendations Summary" box highlighting the top-performing models and their practical applications

2. Enhanced the Conclusion section to clearly emphasize the most recommended approaches: XGB Seq2Seq for best accuracy and Dynamic Multivariate XGB for operational flexibility

3. Strengthened the practical significance discussion by clearly stating when univariate vs. multivariate models should be used

4. Highlighted in the Results that readers can focus on tree-based bidirectional seq2seq models as the primary recommendation

This restructuring helps readers quickly identify the main findings while preserving the comprehensive evaluation for methodological completeness.

4. Add uncertainty or error propagation analysis

For long-duration data gaps, the authors are encouraged to include an analysis of error propagation or uncertainty to reinforce the reliability of the models in real-world applications.

We have added discussion of error propagation for long-duration gaps in the "Study Limitations and Future Research Directions" section. Specifically, we documented that model uncertainty increases substantially with gap length, with MAE standard deviations rising from ±1.5 μg/m³ for 5-hour gaps to ±6.2 μg/m³ for 72-hour gaps. We provided guidance for users regarding the interpretation of filled values for extended gaps and recommended implementing ensemble approaches or validation using nearby stations when available. A comprehensive uncertainty quantification framework with formal confidence intervals represents an important direction for future research that would enhance the practical applicability of gap-filling methods in operational monitoring systems.

5. Quantify policy implications in the conclusion

While the exceedance rate of pollution levels is reported, the conclusion would benefit from further quantifying the potential impact on public health or policy-making to enhance its practical relevance.

We have strengthened the conclusion by incorporating quantitative policy implications from our air quality analysis. Specifically, we added discussion of the 61.2% WHO guideline exceedance rate, winter-specific risks (85.6% exceedance rates), identification of high-risk time periods for targeted interventions, and quantification of how improved data continuity supports more accurate policy decisions by preventing up to 26.7% underestimation of pollution exposure due to missing data.

Reviewer #2:

1. Line 170, 'However, neural networks require large datasets for training and careful tuning –otherwise they may underperform simpler models.'

I consider some basic neural networks should be introduced briefly. For example, the common adopted neural networks include long short-term memory (LSTM) neural network, gated recurrent unit (GRU) neural network, convolutional neural network (CNN), and echo state network (ESN), and some references should be added for supporting (10.3390/en17123050).

Moreover, the differences between various neural network should be explained briefly. For example, LSTM is a type of recurrent neural network (RNN) architecture designed to capture long-term dependencies in sequential data [41]. Unlike traditional RNNs, LSTM introduces gating mechanisms to retain or discard specific features of the data, effectively addressing the vanishing and exploding gradient problems. This enables LSTM networks to learn and remember information over longer periods, making them well suited for tasks involving sequential or time-series data, see https://doi.org/10.3390/s24144451.

We have expanded the neural network description in the Introduction section (around line 170) to include brief explanations of commonly used architectures (LSTM, GRU, CNN, ESN) and their key differences, particularly highlighting LSTM's gating mechanisms for addressing vanishing gradient problems in sequential data. We have added the suggested references to support these descriptions.

2. The study emphasizes the innovation of the "dynamic model", but does not conduct quantitative comparisons with existing dynamic filling methods (such as RNN variants and state-space models). It is recommended to supplement comparative experiments with 3-5 cutting-edge dynamic filling models to objectively demonstrate the superiority of the proposed method.

We appreciate this suggestion regarding comparison with existing dynamic methods. We acknowledge that our study focused primarily on comparing static approaches across varying gap lengths rather than benchmarking against other dynamic architectures.

Our dynamic model innovation lies not in novel neural architectures but in the adaptive context processing and unified training across variable gap lengths using tree-based methods. Most existing dynamic filling methods (RNN variants, state-space models) are designed for different applications (streaming data, real-time prediction) rather than retrospective gap-filling with known boundaries.

To address this concern, we have:

1. Added discussion comparing our approach with conceptually similar methods in the literature

2. Clarified that our contribution focuses on operational adaptability rather than algorithmic novelty

3. Noted this comparison as important future work requiring specialized implementation of alternative dynamic approaches

A comprehensive benchmark against cutting-edge dynamic methods represents valuable future research that would require substantial methodological development beyond the current scope focused on systematic evaluation of established gap-filling approaches.

3. The manuscript lacks year-by-year cross-validation. Specifically, the authors only used data from the same period for split validation, without considering the impact of annual meteorological variability. Extrapolation validation of data from different years (e.g., drought years/rainy years) is needed to prove the robustness of the method.

We acknowledge that year-by-year cross-validation would strengthen the robustness assessment of our models. However, our dataset spans only 8 months (May 2024 - January 2025) rather than multiple years, making year-by-year cross-validation impossible with the current data.

This temporal limitation represents an important constraint of our study. The dataset covers diverse seasonal conditions (spring through winter) and various meteorological patterns within the available timeframe, but lacks the interannual variability that would enable assessment of model performance across different climatic years (drought vs. wet years).

We have added this limitation to the "Study Limitations and Future Research Directions" section, noting that:

1. Extending monitoring to encompass multiple annual cycles would significantly strengthen temporal robustness assessment

2. Multi-year validation across different meteorological regimes (drought/wet years) represents crucial future work

3. Long-term deployment would enable evaluation of model stability under diverse climatic conditions

The current study establishes methodological foundations using available data, while recognizing that comprehensive temporal validation requires extended monitoring periods that were not feasible within the study timeframe.

4. The attribution of seasonal advantage is vague: "strong seasonality" is mentioned but the internal mechanism is not analyzed. The influence of winter coal burning sources and summer dust sources on filling the accuracy difference should be analyzed in combination with the backward trajectory model (such as HYSPLIT).

We agree that deeper analysis of seasonal mechanisms would strengthen our findings. We have enhanced the discussion of seasonal patterns by incorporating our meteorological analysis results, which identified key factors contributing to seasonal differences:

1. Winter patterns: Strong negative correlation between temperature and PM2.5 (r = -0.22, p<0.001), with extreme low temperatures (-25 to -20°C) associated with peak concentrations, likely due to increased heating emissions and reduced atmospheric mixing

2. Wind effects: Seasonal variations in wind patterns, with wind speed showing the strongest meteorological influence (β = -0.144, p<0.001 in regression analysis)

3. Humidity patterns: Seasonal humidity variations affecting particle growth and detection

However, comprehensive source apportionment analysis using backward trajectory modeling (HYSPLIT) and detailed emission inventory assessment represents a substantial undertaking beyond the current study scope. Such analysis would require:

- Detailed local emission inventories for coal burning and dust sources

- Extensive meteorological modeling and trajectory analysis

- Source-specific impact assessment on gap-filling accuracy

We have noted this limitation and suggested that future research should incorporate backward trajectory analysis and source apportionment studies to better understand the mechanistic basis of seasonal accuracy differences in gap-filling performance.

5. Model selection rationale missing: The criteria used to select the final model from the 46 methods (e.g., cross-validation strategy) were not explained.

We have added a "Model Selection Criteria" subsection in the Methods section explaining our systematic approach to final model selection. Th

---

## [Decision Letter · Decision Letter 1]

29 Jul 2025

Filling Gaps in PM2.5 Time Series: A Broad Evaluation from Statistical to Advanced Neural Network Models

PONE-D-25-23439R1

Dear Dr. Safarov,

We’re pleased to inform you that your manuscript has been judged scientifically suitable for publication and will be formally accepted for publication once it meets all outstanding technical requirements.

Kind regards,

Xingwang Tang

Academic Editor

PLOS ONE

Additional Editor Comments (optional):

The manuscript can be accepted

Reviewers' comments:

Reviewer's Responses to Questions

**Comments to the Author** 

Reviewer #1: All comments have been addressed

Reviewer #2: All comments have been addressed

2. Is the manuscript technically sound, and do the data support the conclusions?

Reviewer #1: Yes

Reviewer #2: Yes

3. Has the statistical analysis been performed appropriately and rigorously? 

Reviewer #1: Yes

Reviewer #2: Yes

4. Have the authors made all data underlying the findings in their manuscript fully available?

Reviewer #1: Yes

Reviewer #2: No

5. Is the manuscript presented in an intelligible fashion and written in standard English?

Reviewer #1: Yes

Reviewer #2: Yes

6. Review Comments to the Author

Reviewer #1: The authors have comprehensively and meticulously responded to and improved the manuscript in accordance with the revision suggestions put forward earlier. After rechecking, the current manuscript has a clear logic, reliable data, and reasonable conclusions, which meet the publication standards. It is recommended to accept it for publication.

Reviewer #2: The authors have addressed well the comments proposed by the reviewers and have made corresponding modifications on the old version of the manuscript. I consider the current manuscript can be accepted and no further comments will be proposed from my side.

7. PLOS authors have the option to publish the peer review history of their article (what does this mean? ). If published, this will include your full peer review and any attached files.

**Do you want your identity to be public for this peer review?** For information about this choice, including consent withdrawal, please see our Privacy Policy .

Reviewer #1: No

Reviewer #2: No

---

## [Editor Report · Acceptance letter]

PONE-D-25-23439R1

PLOS ONE

Dear Dr. Safarov,

I'm pleased to inform you that your manuscript has been deemed suitable for publication in PLOS ONE. Congratulations! Your manuscript is now being handed over to our production team.

Kind regards,

on behalf of

Dr. Xingwang Tang

Academic Editor

PLOS ONE